# Rethinking Multimodal Time-Series Forecasting Evaluation

Haoxin Liu [1 2]  Yichen Zhou [2]  Rajat Sen [2]  B. Aditya Prakash [1]  Abhimanyu Das [2]

**Project Page:** https://haoxin1998.github.io/TimesX-project/

## Abstract

We introduce a new context-enriched, multimodal time series forecasting benchmark, TimesX. TimesX contains a wide selection of high-quality real-world time series with diverse domains and textual contexts obtained from an automated data generation pipeline, which helps address three main issues of existing multimodal forecasting benchmarks: (1) poor generalization due to the small scale and synthetic nature of benchmark data, (2) very limited types of textual contexts in the benchmarks, and (3) an inability to mitigate data leakage in evaluation. We conduct a thorough empirical study of zero-shot multimodal forecasting approaches on TimesX. Our results suggest that many approaches that perform well on existing benchmarks may fail on TimesX. In contrast, simple ensemble methods that leverage rich textual context accompanying time-series can outperform strong baselines on the TimesX.

## 1. Introduction

Time-series forecasting (TSF) (Liu et al., 2024a) is a ubiquitous task across numerous domains and is essential for informed decision-making. Motivated by success in NLP, there has been significant work in recent years on time-series foundation models (TFM) for forecasting, ranging from re-purposing LLMs directly for forecasting (Gruver et al., 2023; Tan et al., 2024) to fine-tuning pretrained LLMs on time-series data (Zhou et al., 2023; Chang et al., 2025) to pretraining time-series foundation models from scratch (Das et al., 2024; Goswami et al., 2024; Woo et al., 2024; Ansari et al., 2024). These models have demonstrated promising zero-shot TSF capabilities, often outperforming traditional statistical and supervised methods, using only the historical context of the time-series at inference time. Although historical numerical data provide a fundamental basis for prediction, they often lack the complete context necessary for reliable and accurate forecasts. Human forecasters often integrate additional information such as background knowledge, events and constraints, which can usually be captured through natural language (Liu et al., 2024c) - we call these *textual contexts*. Improving the ability of time-series foundation models to effectively leverage and integrate diverse textual contexts remains an ongoing challenge. This highlights a crucial need for high-quality and comprehensive multimodal datasets and benchmarks to propel research in this area.

At the same time, existing context-enriched multimodal TSF evaluation benchmarks and their construction techniques present several limitations when benchmarking pretrained models on them. The first limitation is the unknown generalization gap between the benchmarking metrics and the real world performance. This is often caused by either the benchmark being restricted to a *small scale* with a narrow selection of domains, or it being fully *synthetic*.

The second significant challenge across existing benchmarks is *data leakage*. Pretrained TFMs and LLMs may have already ingested evaluation data, leading to unknown data contamination and unfair comparison between methods. A benchmark itself is subject to short-lived validity as pretrained models will continuously update their pretraining datasets, which results in their knowledge cut-off becoming more recent than the benchmark's data.

The third limitation comes from our observation that the types and granularities of textual contexts in previous benchmarks are often limited and vary wildly. Textual contexts derived from real data include metadata information about the time-series, timestamp-related information such as important dates and holidays, textual event data related to the time-series, and textual data capturing statistics and trends of related exogenous variables correlated with the target time-series. The ability of multimodal models to process these different types of contexts, and the availability and quality of these different contexts in the datasets have a

This work was done while Haoxin Liu was a Student Researcher at Google Research. [1] Georgia Institute of Technology [2] Google Research. Correspondence to: Haoxin Liu <hliu763@gatech.edu>.

significant effect on the model's benchmark performance. When a multimodal model falls short on existing benchmarks, it is difficult to conclude whether the shortcoming is due to the fact that the method does not effectively utilize the provided information, or it is because the contexts provided in the benchmark are of a specific type, too vague, or lacking relevant details.

To address these fundamental limitations and establish an unbiased benchmark for context-enriched time series forecasting, we introduce **TimesX**, a novel multimodal benchmark designed to focus on (1) real, large-scale, cross-domain data, (2) a dataset generation pipeline providing continuous mitigation of data leakage, and (3) a collection of comprehensive, fine-grained textual contexts. We further introduce several promising baselines that combine TFMs and LLMs to achieve non-trivial multimodal, context-enriched forecasting performance on our benchmark.

The primary contributions of this paper are threefold:

1. TimesX is, as far as we are aware, the first real-world, large-scale, and cross-domain context-enriched time series forecasting benchmark. It encompasses 19 diverse domains with a total of 190 variables, covers diverse global geographical regions and includes daily and weekly frequencies. It also links each target numeric series to multiple forms of detailed textual contexts. Unlike existing multimodal time-series benchmarks, all time series and textual data in TimesX are from real-world observations.

2. We propose two new mechanisms in the dataset generation pipeline of TimesX to help prevent data leakage and guarantee the validity of the textual context. The first one is an automated data collection pipeline that adopts strict timestamp alignment and isolation during each of its steps. The second one is a hypothesizer-verifier-enricher framework for fact-checking and enriching the textual contexts. These mechanisms combined ensures that TimesX is verifiable, leakage-free, and can be updated in the future for benchmarking methods with new pretraining cut-off dates.

3. We empirically verify that TimesX addresses the shortcomings of existing benchmarks. In addition, we conduct thorough empirical studies comparison of current (zero-shot) multimodal TSF approaches on TimesX, involving more than 312,000 independent LLM inferences and revealing new observations that were not discovered by existing benchmarks. In particular, we discover that earlier benchmarks like Williams et al. (2024) (while being a great test for instruction following in forecasting) tend to severely over-estimate the performance of LLMs over TSF models. At the same time, benchmarks like Liu et al. (2024b) under-report

the importance of textual context information in forecasting accuracy.

Limitations and future works are detailed in Appendix A.

Conflict of Interest Disclosure. Y.Z., R.S., and A.D. are employed by Google, which develops Gemini and TimesFM models evaluated in this paper. H.L. was a Student Researcher at Google Research during this work. We disclose these relationships in accordance with the ICML 2026 camera-ready policy.

## 2. Related Work

In addition to the time-series foundation models mentioned previously, recent works (Jin et al., 2023; Liu et al., 2024c; Wang et al., 2025b) have started to adapt pre-trained LLMs to perform forecasting when textual contexts are available, often by aligning the modalities of time series and natural language. The development of these models has been paralleled by new benchmarks designed to evaluate them. ChatTS (Xie et al., 2025), for instance, generates synthetic time series paired with detailed attribute descriptions to train a multimodal LLM for understanding and reasoning tasks. Similarly, the Time-MQA framework (Kong et al., 2025) introduces the TSQA dataset, which contains real-world time series with template-based meta information and turns time-series forecasting into a question-answer format. Benchmarks such as those referenced in CiK (Williams et al., 2024) contain real-world time series and manually crafted textual contexts that are critical for forecasting. Time-MMD (Liu et al., 2024b) provides a context enriched benchmark constructed by keyword-based web searches of nine domains, one variable for each. MTBench (Chen et al., 2025) aligns stock prices with financial news and weather reports with temperature records, curating tasks that require reasoning about. MoTime (Zhou et al., 2025) dataset suite provides a collection of multimodal datasets pairing time series with static modalities like text and images. Noticeably, these early efforts were often constrained by a scarcity of high-quality, large-scale, real-world datasets, leading some to rely on synthetic data and specific forms of textual contexts. We summarize several representative benchmarks compared to our proposed TimesX in Table 1.

## 3. The TimesX Benchmark

### 3.1. Design Principles of TimesX

We will first present the core principles that distinguish TimesX from other multimodal forecasting benchmarks: (i) real-world data, (ii) leakage mitigation, and (iii) comprehensive, high-quality context and (iv) large-scale evaluation. Note that we further provide empirical evidence that demonstrates the importance of these principles and where other

| Benchmark | Real Data | Leakage-Free | Context Types | Datasets |
|---|---|---|---|---|
| MT-Bench | Yes | No | Meta+Event | 2 |
| ChatTime | Yes | No | Meta+Calendar+Covariates | 3 |
| MoTime | Yes | No | Meta | 8 |
| Time-MMD | Yes | No | Meta+Event | 9 |
| CiK | No | N/A | Meta+Event+Covariates | 71 |
| **TimesX** | **Yes** | **Yes** | **Meta+Calendar+Covariates+Event** | **190** |

*Table 1.* Comparison of multi-modal TSF benchmarks. Leakage-Free indicates whether the benchmark can guarantee no data leakage for the latest pretrained models, detailed in Section 3.1.2.

benchmarks might fall short.

### 3.1.1. REAL-WORLD DATA

**Principle Description**: TSF is highly complex due to real-world dynamics, complex processes, and diverse domains. On the other hand, many existing benchmarks use either synthetic time series, synthetic textual contexts, or both, without demonstrating how the conclusions on synthetic data can transfer to real-world scenarios (Williams et al., 2024; Tan et al., 2024). TimesX differentiates itself from those benchmarks by **using real-world data for both the time series and the textual contexts**.

**Why it matters**: To test whether conclusions on synthetic data can transfer to real data, we evaluate three methods on a synthetic context dataset (CiK) and our real dataset (TimesX). We report the aggregated MASE (Mean Absolute Scaled Error) of the three methods which are: (i) TimesFM-2.5 (a TFM) using only the time series, (ii) Gemini-2.0-Flash (an LLM) using both the time series and the textual contexts, and (iii) CODEREV, Gemini-2.0-Flash writing and executing code to revise TimesFM-2.5 outputs based on the context. More details are in Appendix N.

As shown in Figure 1, **the rankings of the three methods on CiK (synthetic) and our real data are completely different**. On CiK, directly feeding the time-series along with associated text context to Gemini has a huge lead over TimesFM-2.5. Moreover, asking Gemini to edit the output of TimesFM-2.5 via code does even better, significantly outperforming the other two methods. On the other hand on TimesX, we see that all the methods are much closer, and TimesFM-2.5 is actually better than Gemini. Further CODEREV degrades the performance of using both TimesFM-2.5 and Gemini directly. This shows that CiK has a strong bias that clouds its conclusions: in CiK, textual contexts are both synthetic and extremely specific - these specific contexts are then used to write code to modify an input time-series and generate the final forecasting task. Therefore, such tasks can be easily solved by instruction-following language models and even better by language models that generate code to accomplish a simple modification task. These conclusions, however, do not generalize to the real-world tasks, whose contexts are less specific and do

not relate to the forecasting task via the coding path.

*Remark* 3.1 (**The value of synthetic data**). *We highly recommend using both synthetic and real-world benchmarks for a more robust evaluation. Carefully designed synthetic benchmarks can be useful for testing specific capabilities such as instruction following or different types of reasoning.*

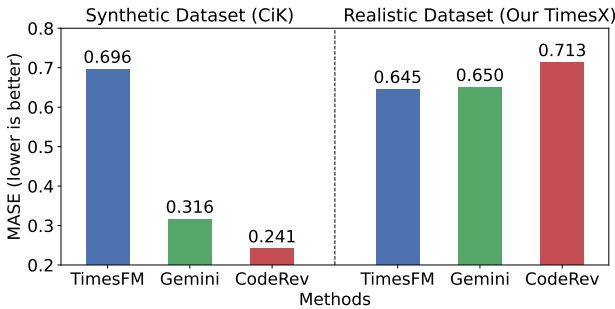

*Figure 1.* Synthetic (CiK) versus real-world (TimesX) performance under the same setup (lower MASE is better). The ordering of TimesFM-2.5, Gemini-2.0-Flash, and CODEREV flips between the two datasets, showing that synthetic generation can bias rankings toward instruction-following LLM and coding. We show results on the Future subset of CiK here. Table 15 further reports the average performance across all subsets with the same conclusion.

### 3.1.2. MITIGATING DATA LEAKAGE

**Principle Description**: A fair benchmark dataset should be able to mitigate any potential leakage of benchmarks tasks into the pretraining datasets of the models being evaluated. Given that many pretrained models do not release their pretraining datasets, existing benchmarks that claim to use data from unused sources (data isolation) for evaluation have three drawbacks: (1) Potentially unidentifiable data contamination. (2) Limited number of clean tasks left for benchmarking multiple methods. (3) Short-lived validity i.e future versions of the models considered may unintentionally leak the benchmark into their pretraining data.

In contrast, **TimesX adopts strict *time isolation* to prevent data contamination**: we timestamp TimesX thoroughly, and recommend strictly using benchmark tasks that occur after a target model's knowledge cutoff date. Moreover, **the TimesX pipeline is designed as re-updateable over time**, i.e data collection, validation, alignment, and model evalu-

| Method | Model | MASE Before 2024.06 | MASE After 2024.06 | % Change |
|--------|-------|---------------------|--------------------|----------| 
| LLM | Gemini-2.0-Flash | 0.514 | 0.594 | 14.81% |
| LLM | DeepSeek-V3 | 0.606 | 0.681 | 12.38% |
| TFM | TimesFM-2.5 | 0.563 | 0.573 | 1.78% |
| TFM | Moirai-2.0 | 0.691 | 0.696 | 0.72% |

*Table 2.* Performance before and after the two LLMs' knowledge cutoff (June 2024) on the Search Trend subset (120 variables). Lower MASE is better. Setup is detailed in Appendix B.

ation can all be automated. To the best of our knowledge, TimesX is the first benchmark capable of being automatically refreshed for evaluating future pretrained methods.

**Why it matters**: To study how data contamination affects evaluation results, we evaluate on 120 variables from the Search Trend (see Section 3.2) subset of TimesX. We keep the numeric series, windowing, and prompts fixed, and split the test period by the public knowledge cutoff of Gemini-2.0-Flash and DeepSeek-V3 (June 2024). We report the MASE aggregated by Geometric Mean in Table 2.

The pretraining data descriptions of the two TFMs preclude any time-series data leakage in both time periods, and **both TFMs report stable results** with less than 2% delta. In contrast **both LLMs see an increase of error by about 13% after their knowledge cutoff.**

### 3.1.3. COMPREHENSIVE AND HIGH-QUALITY CONTEXT

**Principle Description**: Most multimodal forecasting benchmarks have a very limited set of text contexts - restricted to either static metadata (Zhou et al., 2025), date and weather derived features (Wang et al., 2025b) or synthetically generated contexts (Williams et al., 2024). In contrast, TimesX uses not only **a union of all common context types**, but also a more **high-quality description of text events** and corresponding alignment of these contexts.

In particular, we categorize these various kinds of contexts into: (1) **Metadata**: descriptive high level summaries of the forecasting variable in question (2) **Calendar**-derived features like holidays (3) **Covariates** which cover textual information around other related time series (4) **Time-stamped events**: Textual description of related events aligned with time-windows of the variable in question. Section 3.2.2 contains more details about how we generate these contexts.

**Why it matters**: TimesX links each target time-series to all these available context types and aligns them with the timestamps of the time series. The end result is a benchmark with diverse, high-quality textual contexts. To quantify its effect on TSF accuracy, we run a controlled experiment where we take the time-series in the Time-MMD benchmark and use our methodology to replace the textual events using

the construction rules in Appendix E. We keep the numeric series, the LLM (Gemini-2.0-Flash), and the prompt fixed.

Table 3 (full results in Table 18) compares the performance of Gemini-2.0-Flash given these newly generated text events versus the original context in the Time-MMD benchmark. Across all nine Time-MMD datasets, this **context swap reduces the geometric–mean aggregated MASE from** 0.906 **to** 0.840 **(a** 7.3% **relative drop)**, suggesting that conclusions drawn under low–quality context might be misleading; **high–quality context changes the picture**.

We further show that **with the current dataset scale (typically fewer than 20 variables), model performance rankings are quite unstable**, as illustrated in Figure 34 and Appendix T.

| Method | MASE |
|--------|------|
| LLM (Our Context) | **0.840** |
| LLM (Time-MMD Context) | 0.906 |

*Table 3.* Controlled replacement of textual context on *Time-MMD*. More detailed results are in Appendix S.

### 3.2. Construction of TimesX

We first provide a high level summary of our dataset and dataset construction methodology. TimesX contains time-series obtained from **19 domains**. Each domain has 10 variables, resulting in a total of **190 *variables*** (quantities evolving with time). TimesX covers *different geographical regions*: North America (United States, Canada, Mexico), Asia (China, India among others), Europe (United Kingdom, Norway among others), South America and Africa.

Note that we have a separate portion of the dataset that is derived from the time-series in Time-MMD (Liu et al., 2024b), where the text context is generated using our methodology, but only used for ablations in Section 3.1.3.

*Remark* 3.2 (**Originality**). *The TimesX dataset is constructed **entirely from scratch**, rather than by merging existing datasets. Moreover, TimesX is designed to be **automatically refreshable** over time, enabling continuous mitigation of information leakage as pretrained models evolve*

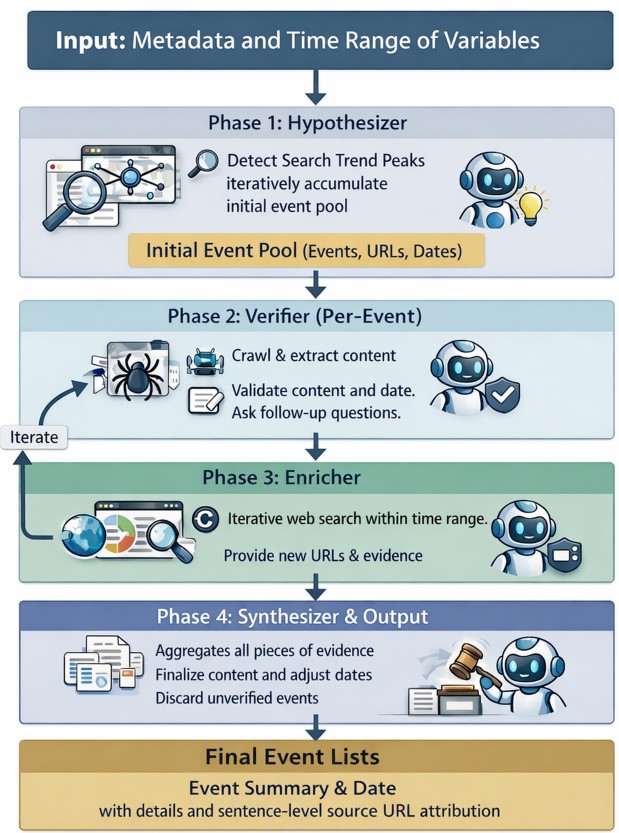

*Figure 2.* Overview of the dataset agents for automatic construction of event contexts. The multi-agent pipeline iteratively retrieves, verifies, and enriches event information. In Phase 3, strict programmatic time range constraints are enforced to ensure timestamp accuracy and prevent future info leakage. Detailed execution logs are provided in Appendix F.

### 3.2.1. NUMERICAL TIME SERIES CONSTRUCTION

The numerical component consists of target variables whose future values are forecasted given a historical window.

**Granularities and sources.** TimesX includes two temporal granularities: *Weekly* series from Google Search Trends across 12 domains[1]; and *Daily* series including (i) commodity prices from a market data API[2] and (ii) major USD exchange rates from a currency rates API[3]. These API data can be *automatically refreshed* to extend the benchmark over time. Monthly/quarterly coverage is discussed in Appendix A.1.

**Time alignment and missing values.** All series within the same frequency share the same timestamps. Missing values are handled as described in Appendix D.1. Appendix I provides the full list of domains and variables.

---

[1] https://trends.google.com/trends/
[2] https://marketstack.com/
[3] https://frankfurter.dev/

### 3.2.2. TEXT CONTEXT CONSTRUCTION

Our core contribution is aligning comprehensive and diverse text context information with the variables that evolve over time. We will see that methods that capture both this context information along with the past of the time-series can achieve superior performance compared to methods that only use the past time-series component. As mentioned before we provide 4 kinds of text contexts for all time-series variables: Metadata, Calendar features, Covariates and Time-Stamped Events. Section 3.2.3 shows an example of a time series with each of these four contexts.

**Metadata and calendar contexts.** The Metadata and Calendar text contexts are easily constructed from the sources of the numerical time-series themselves and Python Holidays library respectively, as described in Appendix C.

**Covariate contexts.** For each time-series in a given domain, we use the other time-series from that domain to extract Covariate text contexts. Specifically, we calculate features of these covariates over the historical window, including mean, median, max and min with the corresponding date, and overall trend direction. These features are then transformed into natural language descriptions, detailed in Appendix D.

**Time-stamped events via dataset agents.** Constructing and aligning textual events is the most challenging component. Following our design principles in Section 3.1, we must satisfy three goals at once: *real-world* data, *leakage-free (refreshable)*, and *high-quality*. A single LLM or a naive web search cannot reliably meet these goals. Therefore, we adopt a **multi-agent automated workflow** with four agents: *Hypothesizer*, *Verifier*, *Enricher*, and *Synthesizer*, as illustrated in Figure 2. Each agent interacts with an LLM under constraints and has dedicated tools (time-bounded web search and lightweight crawlers). Overall, the Hypothesizer and the Verifier act adversarially to ensure event truthfulness and timestamp accuracy, which prevents information leakage; the Verifier then guides the Enricher to supply missing details thus ensuring quality. The whole workflow, is automated, enabling regular updates of the benchmark data. We provide the details of each LLM role:

(i) the *Hypothesizer* identifies points of interest in the time series (e.g., local maxima, unexpected movements or peak in the corresponding search trend) and iteratively calls an LLM agent with integrated web search to build an initial event set that are able to match these points.

(ii) then, the *Verifier* uses crawlers to fetch relevant URLs corresponding to each event window and conducts fact checks, filters out any hallucination or leakage, and prepares a checklist of missing details.

(iii) The *Enricher* resolves the checklist using strictly time-bounded web searches, merging multiple sources to fill in

the details.

(iv) finally, the *Synthesizer*, a reasoning LLM, aggregates all pieces of evidence to finalize the event description, adjust timestamps, and discard events with unresolved doubts.

Using this agent, TimesX automatically produces an event corpus that matches the time window, contains verifiable facts (each claim has a supporting URL), accurate time stamps, and rich detail. More details and hyperparameter analysis experiments are provided in Appendix E. An execution log demo is in Appendix F.

*Remark* 3.3 (**Scalability and Cost**). *Our pipeline runs in three-month time blocks and is naturally parallelizable across variables and blocks; Phase 1 uses coverage-based early stopping to reduce cost. Users can control the budget by choosing different LLMs per role and tuning the maximum iterations (Phase 1) and search budget (Phase 3). Under our current setup (Gemini 2.5 Pro + Gemini 2.5 Flash), the cost is about $0.7 per variable per three-month block; it can be reduced by using open-source LLMs or batching verification so multiple events reuse the same calls.*

### 3.2.3. DATA EXAMPLE

We provide below an example from TimesX corresponding to the GAS_PRICE time-series.

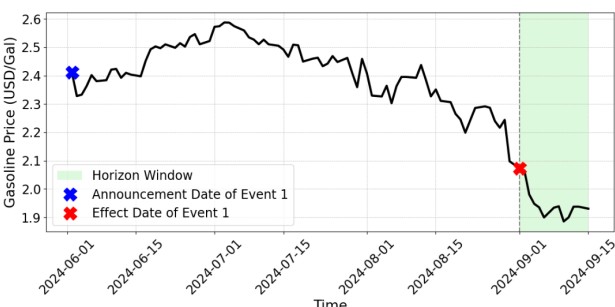

*Figure 3.* The numerical example from the GAS_PRICE.

**Time series**: See Figure 3.

**Metadata**: This time series records gasoline price (USD/-GAL) in the Commodity Price domain, with daily frequency. Prediction target period: from 2024-09-01 to 2024-09-15.

**Date**: In the format of (timestamp, value), historical data is: (2024-06-02, 2.4116), (2024-06-03, 2.3279),... Upcoming holidays in the prediction window: Labor Day (2024-09-02).

**Covariates**: from 2024-06-01 to 2024-08-31: (1) Brent Crude Oil (USD/BBL): The maximum value was 87.43, occurring on July 4, the minimum..., showing an overall downward trend. (2)...

**Events**: (1) On June 2, 2024, OPEC+ agreed to extend deep

oil output cuts [1,3], the cut of 2.2 million bpd would be extended until September 2024, after which it would be gradually phased out [2,3]. Source: [1] [2] [3]; (2) ...

### 3.2.4. QUALITY CHECKS.

**Diversity.** We profile numerical and textual features of TimesX (Appendix J) and visualize them with PCA/t-SNE (Appendix M). Samples are broadly dispersed, and variables from different sources form distinct clusters, supporting the benefit of multi-source integration.

**Factual correctness.** Here, we focus on the most sensitive component: the accuracy of automatic timestamp annotation. We ensure the accuracy via a three-stage pipeline (Appendix G): verification, programmatic time-bounded search ($\pm k$ days), and cross-source adjudication in the agent workflow. A manual audit of 50 samples shows *94% exact matches; 4% conservative (later) offsets; and 2% earlier due to date ambiguity* (Appendix G).

### 3.3. TimesX Release and Refresh Consideration

The core TimesX dataset now contains 190 variables spanning Jan 2018 to Oct 2025 to support both training (2018–2023) and evaluation (2023–2025). We plan to update the TimesX dataset every three months and version each release.

The extended TimesX further supports 11 languages beyond English (Afrikaans, French, German, Hindi, Japanese, Korean, Portuguese, Simplified Chinese, Spanish, Swahili, and Turkish), as well as 5 rare diseases. See Appendix A.5

## 4. Empirical Study

### 4.1. Evaluation Settings

We focus on zero-shot solutions and consider the following:

**Pretrained TFMs**: We select three SOTA TFMs including TimesFM-2.5, Moirai-2.0, and Sundial. All are towards the top on GIFT-Eval (Aksu et al., 2024) benchmark, with TimesFM-2.5 being the top open model with no leakage.

**Pretrained LLMs**: We consider three well-adopted LLMs (implementation details in Appendix Q): closed-source Gemini-2.0-Flash and GPT-4o, and open-source DeepSeek-V3.

**Composed (Ensemble And Agentic) Solutions**: We construct the following four solutions: (1) AVGENS: An ensemble that averages the forecasts from a pair of TFM and LLM. We simply set equal weights.(2) TEXTREV: An LLM taking the forecast of a TFM as text and revising it according to the context. (3) CODEREV: An LLM writing and executing code to revise the forecast of a TFM according to the context. (4) FUNCREV: Similar to CODEREV, but the LLM is limited to a selection of functions. See Appendix Q.

| | Method | MASE | Rank |
|---|---|---|---|
| Naive | SeasonalNaive | 1.000 | 12.196 |
| Unimodal Zero-Shot TFM | Sundial | 0.771 | 9.556 |
| | Moirai-2.0 | 0.722 | 7.968 |
| | TimesFM-2.5 | 0.645 | 5.757 |
| | AVGENS: TimesFM-2.5 + Moirai-2.0 | 0.668 | 6.45 |
| Multimodal Zero-Shot LLM | DeepSeek-V3 | 0.708 | 7.73 |
| | Gemini-2.0-Flash | 0.650 | 6.603 |
| | GPT-4o | 0.643 (#3) | 5.466 (#3) |
| Multimodal Composed Solution | FUNCREV: TimesFM-2.5 + Gemini-2.0-Flash | 0.720 | 7.665 |
| | CODEREV: TimesFM-2.5 + Gemini-2.0-Flash | 0.713 | 6.968 |
| | TEXTREV: TimesFM-2.5 + Gemini-2.0-Flash | 0.653 | 5.63 |
| | AVGENS: TimesFM-2.5 + GPT-4o | 0.627 (#2) | 4.735 (#2) |
| | AVGENS: TimesFM-2.5 + Gemini-2.0-Flash | 0.619 (#1) | 4.249 (#1) |

*Table 4.* Overall benchmark results (mean over 10 runs) of the 13 selected methods. The top 3 methods per metric are numbered in the parentheses. The continuous ranked probability score is further used to reveal how context helps model uncertainty, in Appendix U.1.

| | Meta | Meta+Date | Meta+Date+Cov | Meta+Date+Event | Meta+Date+Event+Cov |
|---|---|---|---|---|---|
| MASE | 0.787 | 0.670 | 0.674 | 0.674 | **0.650** |
| Rank | 3.704 | 3.048 | 3.079 | 2.968 | **2.635** |

*Table 5.* Gemini-2.0-Flash MASE and MASE rank when provided with different context types. Including all context types provides significant improvement over any other combinations. See detailed results in Appendix Y.3

For each variable in TimesX and its context corpus, the look-back window and the forecast horizon are set to 96 and 12, respectively. The rolling window is set to 4 for weekly data and 12 for daily data to balance sample count and sample diversity. We then include all relevant metadata, calendar info and covariates info. To avoid future info leakage, we select the $K$ most recent textual events whose announcement dates strictly precede the first timestamp of the prediction horizon to add to the context. We fix $k = 10$ to balance prompt length and information effectiveness, following common practices (Liu et al., 2024b; Li et al., 2025).

TimesX spans 2018–2025 and is refreshable. However, to avoid pretraining data contamination, we construct evaluation examples whose forecast horizon begins after the pretraining cutoff of all involved models, detailed in Appendix P. Unless stated otherwise, we use **2024-07-01** as the cutoff, i.e., evaluating only the subset whose horizon start time is 2024-07-01 or later. See models' knowledge cutoffs in Appendix P. Web access is limited to the build-time agent; the benchmark runs offline. Appendix P.1 further discusses how to prevent test-time leakage.

Following the conventions in other popular forecasting benchmarks like GIFT-Eval (Aksu et al., 2024) we use normalized MASE (mean absolute scaled error) as our main metric. Since the different variables have very different scales, we calculate the average MASE over all rolling windows of a variable and normalize that by the average MASE of a seasonal naive baseline. Then we take the Geometric Mean (GM), for robustness to normalization choice, of

these normalized MASE ratios across all variables. See Appendix N. We also compute the average MASE rank of each method over the variables. For both metrics, smaller is better.

*Remark* 4.1 (**Evaluation Scale**). *This setup yields about 2.5K samples (Appendix O). To account for stochasticity, we repeat each evaluation 10 times with different seeds when applicable, resulting in more than 312K independent LLM inferences. Note that using a shorter rolling-window stride can increase the sample count but also the evaluation cost.*

### 4.2. Benchmarking Results

Table 4 shows the benchmark results on TimesX (details in Appendix Y.1). As multimodal solutions, *though the zero-shot LLMs have the edge over the unimodal zero-shot TFMs, this edge is not as significant on TimesX as observed on other synthetic benchmarks*. This points out the gap between synthetic contexts and real world contexts, that some synthetic benchmarks tend to severely over-estimate the performance of LLMs over TSF models because the synthetic contexts were providing the exact information required for bettering the forecast. Real world contexts are in contrast too nuanced to always deliver large performance boost even when guaranteed to be related. This conclusion is further confirmed by the observation that *the agentic* CODEREV *method is performing worse* than its two components working alone, contradicting the claim of (Williams et al., 2024).

In terms of the composed solutions, to our surprise *the best performers on TimesX are the simple average ensembles*

| Method | A&E | C&E | Econ | E. Tech | Fin | P&A | Pub. H. | PPG | Sci | Shop | SSSG | Traf | Crops | Energy | Lvstk. | RMC | SAM | SHVM | Curr |
|---|---|---|---|---|---|---|---|---|---|---|---|---|---|---|---|---|---|---|---|
| | | | | | | | | | | Domain | | | | | | | | | |
| SeasonalNaive | 13 | 13 | 13 | 13 | 13 | 13 | 13 | 13 | 13 | 13 | 13 | 13 | 12 | 13 | 13 | 12 | 11 | 13 | 12 |
| Sundial | 10 | 11 | 12 | 11 | 11 | 12 | 10 | 12 | 10 | 11 | 12 | 10 | 9 | 10 | 9 | 10 | 9 | 12 | 13 |
| Moirai-2.0 | 12 | 12 | 10 | 10 | 10 | 10 | 11 | 11 | 11 | 10 | 9 | **1** | 4 | 4 | 5 | 5 | 5 | 5 | 7 |
| TimesFM-2.5 | 2 | 4 | 4 | 6 | 4 | **1** | 3 | 2 | 3 | 7 | **1** | 4 | 8 | 8 | 2 | 8 | 8 | 10 | 10 |
| AvgEns: TimesFM + Moirai | 9 | 9 | 8 | 8 | 8 | 7 | 6 | 9 | 7 | 8 | 6 | 6 | 2 | 2 | **1** | 7 | 6 | 8 | 8 |
| DeepSeek-V3 | 11 | 10 | 11 | 12 | 12 | 11 | 12 | 10 | 12 | 5 | 11 | 11 | 7 | 3 | 6 | **1** | 2 | **1** | 3 |
| Gemini-2.0-Flash | 6 | 6 | 5 | 7 | 9 | 9 | 8 | 8 | 8 | **1** | 8 | 8 | 6 | 6 | 8 | 3 | 4 | 3 | 4 |
| GPT-4o | 7 | 3 | 6 | 9 | 7 | 5 | 5 | 6 | 9 | 4 | 5 | 7 | 4 | 7 | 5 | 4 | **1** | 4 | 2 |
| FuncRev: TimesFM + Gemini | 8 | 8 | 9 | 3 | 2 | 8 | 9 | 7 | 5 | 9 | 7 | **1** | 11 | 12 | 11 | 13 | 13 | 11 | 9 |
| CodeRev: TimesFM + Gemini | 5 | 7 | 7 | 5 | 3 | 6 | 7 | 5 | 6 | 10 | 9 | 12 | 13 | 11 | 12 | 11 | 12 | 9 | 11 |
| TextRev: TimesFM + Gemini | 3 | 5 | 3 | 4 | 5 | 4 | 2 | 4 | 4 | 6 | 3 | 3 | 10 | 9 | 10 | 9 | 10 | 7 | **1** |
| AvgEns: TimesFM + GPT | 4 | **1** | 2 | 2 | 6 | 3 | 4 | 3 | 2 | 2 | 4 | 5 | 3 | **1** | 3 | 6 | 7 | 6 | 6 |
| AvgEns: TimesFM + Gemini | **1** | 2 | **1** | **1** | **1** | 2 | **1** | **1** | **1** | 3 | 2 | 2 | 6 | 6 | 8 | 3 | 4 | 3 | 6 |

*Table 6.* Breakdown of the MASE ranks by domain. See Appendix Z for MASE. Acronyms are used to shorten domain names (see Appendix I).

of different pretrained models (i.e., AvgEns). We by no means want to suggest it as the optimal composition, but instead want to reiterate the stochastic and flexible nature of LLMs and the fact that it would take a great deal of effort to design the interaction with them in a composed solution to just outperform simple averaging ensembles.

We also use the continuous ranked probability score (CRPS) metrics to measure uncertainty, detailed in Appendix U.1. We initially observe that *all three LLMs achieve lower CRPS than the TFMs*. This suggests that our constructed context helps LLMs model future uncertainty better than TFMs, which only use numerical values.

As an early experiment, we also evaluate reasoning LLMs, including GPT-5, Gemini-2.5-Flash, and DeepSeek-R1. To reduce data contamination, we only use evaluation samples after Jan 2025. Table 17 (detailed in Appendix Y.4) shows that these *reasoning models do not provide a clear advantage*.

### 4.3. Ablation: Context Types

Table 5 presents the MASE and MASE rank when different combinations of contexts are provided to a chosen multimodal method (Gemini-2.0-Flash). Comparing to only using the high level metadata, the inclusion of either calendar, calendar and covariates, or calendar and events brings significant improvement. Using all context types improves the accuracy further eventually being around 16% better than using only static metadata. It is worth noticing that we cannot observe the incremental gain from adding covariates or events alone. We speculate there is crucial interaction between those two context types, for instance the effect of an event on the target variable can be quantified by a similar leading effect on a covariate. Figure 4 suggests this speculation may generalize to other LLMs. This observation is aligned with our expectation that *a multimodal method can compound the gains by composing the information in different context types*. Appendix V reports domain-level holiday and covariate effects.

### 4.4. Ablation: Domains

Table 6 breaks down the ranking of the benchmark methods on each of the 19 domains (see Appendix Z for MASE). There is no dominance of one zero-shot method over others - in fact the orderings of TimesFM-2.5 and Gemini-2.0-Flash are complementary. In terms of composed solutions, the observation is consistent with Table 4 that AvgEns performs well while other compositions can be worse than its components acting alone. We also observe two domain groups where multimodal solutions show a clear advantage over TFMs. First, in Shopping, calendar information likely provides strong signals about when sales occur, which is hard to infer from the time series alone. Second, in RMC (Raw Materials & Constructions), SAM (Specialty & Advanced Materials), SHVM (Strategic & High-Value Materials), and Curr (Currency), the series are strongly affected by external policies, which are captured in the event context.

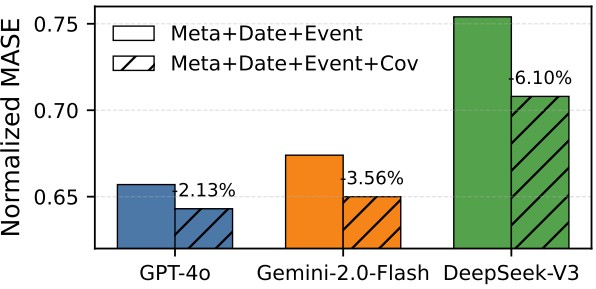

*Figure 4.* The compounded gains from all available context types are present for all LLMs. See detailed results in Appendix Y.2

### 4.5. Why Agentic (Revision) Methods Underperform?

None of the three revision methods (FUNCREV, CODEREV, and TEXTREV) outperforms the simple AVGENS on TimesX. Figure 5 shows that their forecast errors have a wider spread and more severe outliers than TimesFM-2.5, which leads to worse geometric means. Gemini-2.0-Flash alone shows a similar pattern. We speculate that this is due to LLM stochasticity and the fact that LLM-based revision does not reliably preserve the temporal structure in the initial forecast. Among the three, TEXTREV performs best, which suggests that effective code-based revision may require instruction fine-tuning.

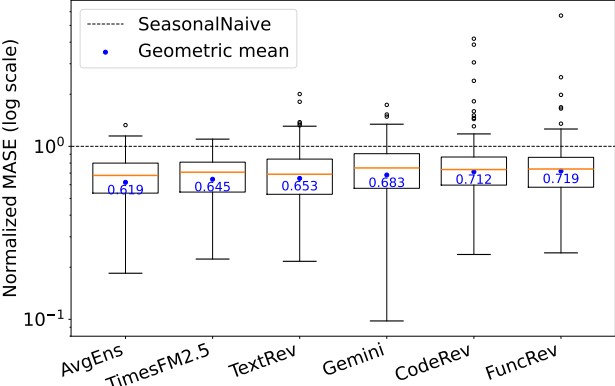

*Figure 5.* The boxplot of TimesFM-2.5, Gemini-2.0-Flash and their composed method's performances.

### 4.6. Future Solutions: Could Training Help?

This work focuses on zero-shot methods. **To support future training-based research, TimesX provides both training (2018–2023) and testing (2023–2025) splits.**

As a first step, we *use in-context learning (ICL) (Brown et al., 2020) as a probe* to verify whether training-based solutions, i.e., learning from historical instances, are promising.

Specifically, we test TEXTREV-ICL, which retrieves one temporally nearby historical example as a demonstration for each TEXTREV instance, and TEXTREV-ICL-C, which further adds an explicit conservative no-change option to the prompt inspired by Figure 5 which highlights t he instability issue (see Section X). Table 7 shows that both variants make TEXTREV more effective: they outperform Gemini-2.0-Flash and TimesFM-2.5, and even surpass the best-performing AVGENS. Thus, we suggest that future training-based methods consider effectiveness and stability.

## 5. Conclusions

We introduce a new multimodal time-series forecasting benchmark, TimesX, which contains real-world, cross-

*Table 7.* Geometric mean MASE on the last three (to avoid demo-caused information leakage) evaluation instances per variable.

| Method | Geometric-mean MASE |
|---|---|
| TEXTREV-ICL-C | 0.707 |
| AVGENS | 0.712 |
| TEXTREV-ICL | 0.737 |
| Gemini-2.0-Flash | 0.748 |
| TimesFM-2.5 | 0.750 |
| TEXTREV | 0.782 |

domain time series with high-quality, detailed textual context. TimesX includes a dataset generation pipeline that keeps the benchmark leakage-free via time isolation and enables automatic refresh via dataset agent. We conduct a detailed empirical study of zero-shot multimodal TSF approaches on TimesX and find that earlier benchmarks either overestimate the performance of LLMs compared with TSF models or understate the importance of textual context for forecasting accuracy. Our evaluation also shows that simple ensemble methods outperform seemingly stronger agentic solutions on TimesX.

## Impact Statement

This paper presents work whose goal is to advance the field of machine learning. There are many potential societal consequences of our work, none of which we feel must be specifically highlighted here.

## Acknowledgment

This work was partly supported by the NSF (Expeditions CCF-1918770, CAREER IIS-2028586, Medium IIS-1955883, Medium IIS-2403240, Medium IIS-2106961), NIH (1R01HL184139), Meta, Modal.com and Dolby faculty gifts.

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

# Appendix

## Appendix Roadmap

This appendix is organized as follows. We first discuss limitations and future directions, including low-frequency coverage, bias, out-of-distribution variables, and future training-oriented extensions. We then provide dataset overview details, textual-event structure, domain and variable breakdowns, construction details, leakage controls, evaluation protocols, and additional quantitative results.

# A. Limitations and future directions

## A.1. Coverage of low-frequency time series

The current release of TimesX focuses on daily and weekly frequencies. There are two main reasons for this choice. First, our goal is to perform leakage-free evaluation using data strictly after the knowledge cutoffs of mainstream LLMs (around June 2024). Under this constraint, even if we collect monthly or quarterly data from July 2024 to August 2025 (submission deadline), there would be only about 13 monthly points or 4 quarterly points per variable, which is too short for a meaningful forecasting evaluation.

Second, many real-world monthly or quarterly indicators can be constructed by aggregating higher-frequency series. Users of TimesX can aggregate our daily or weekly variables into monthly or quarterly indicators when they wish to study lower-frequency behavior. This provides a practical way to analyze coarser temporal patterns on top of TimesX.

We fully agree that native monthly and quarterly series would further improve coverage. We are actively searching for real-world, regularly updated low-frequency series that satisfy the same leakage-free constraint, and we plan to add such variables in future versions of TimesX.

## A.2. Coverage of multivariate time series

The current release focuses on multimodal univariate forecasting. Homogeneous multivariate multimodal TSF, such as forecasting over multiple related sensors, is an important direction for future releases.

## A.3. Coverage of more solutions

This work focuses on the evaluation of zero-shot multimodal time series (TS) forecasting. Accordingly, we benchmarks four advanced zero-shot TSFMs, six LLMs, and their architectural compositions, including ensembles and agents. In contrast, works dedicated to TS understanding and reasoning operate under a distinct paradigm from our forecasting setting. We recommend them as promising backbones for future fine-tuning research and provide the performance of ChatTS (Xie et al., 2025) as a representative reference.

Future work could also consider exploring additional methodologies, such as visual tokenization (Liu et al., 2025a), test-time computation strategies (Liu et al., 2025b), and training-based fusion schemes (Wang et al., 2025a), and general event forecasting (Liu et al., 2026).

## A.4. Geographic and media bias

Event texts in the current version of TimesX are written in English. We adopt English as a starting point, in line with common practice in NLP where a robust English setup is built first and then extended to multilingual settings. At the same time, the numeric variables themselves already cover multiple regions, including North America (United States, Canada), Asia (China, India), Europe (United Kingdom, Norway, Germany), South America, and Africa.

We are aware that geographic and media biases can still arise. As shown in the workflow diagram in Figure 2, our dataset construction agent already takes several steps to mitigate these biases. In Steps 1b and 3b, the web search is configured with a global scope rather than a single country or region. In Step 3b, the search engine retrieves information from multiple sources. In Step 4a, the Synthesizer LLM is instructed to cross-check evidence across sources and to give higher weight to authoritative outlets when producing the final event description. These design choices help, but they cannot completely remove bias.

In future versions of TimesX, we plan to make biases more transparent and more controllable by explicitly disentangling each stored event description into two parts: an "Objective Facts" segment that records dates, numbers, and events that have already occurred, and a "Subjective Analysis" segment that records market sentiment and speculative commentary. This structured separation will allow users to focus on objective information when needed and to design more targeted robustness analyses.

## A.5. Out-of-distribution evaluation: multilingual and sparse-event variables

To explore out-of-distribution (OOD) generalization, we extend TimesX with two types of additional variables that are not part of the main in-distribution set: multilingual variables and sparse-event rare-disease variables.

**Multilingual variables.** The main release of TimesX uses English event texts, but we agree that multilingual and region-specific events are important. To this end, we construct 11 new non-English variables that span five continents and cover the following languages: Afrikaans, French, German, Hindi, Japanese, Korean, Portuguese, Simplified Chinese, Spanish, Swahili, and Turkish. Each variable focuses on region-specific topics in the corresponding language. These multilingual variables are available in an extended

*Table 8.* Statistics of sparse-event rare-disease variables in TimesX. "Events" is the number of constructed events per variable between 2023 and mid-2025; "Avg. summary len" is the average length of event descriptions; the remaining columns are time-series characteristics computed as in Tables 11 and 15.

| Variable | Events | Avg. summary len | Transition | Shifting | Seasonality | Trend | Stationarity |
|---|---|---|---|---|---|---|---|
| Chagas disease | 29 | 443.28 | 0.00651 | -0.733 | 0.555 | 0.239 | $2.49 \times 10^{-3}$ |
| Guinea worm disease | 18 | 423.17 | 0.00536 | 0.107 | 0.724 | 0.134 | $4.48 \times 10^{-14}$ |
| Huntington's disease | 56 | 437.13 | 0.00690 | -0.145 | 0.550 | 0.167 | $7.38 \times 10^{-6}$ |
| Marburg virus disease | 30 | 569.47 | 0.01316 | -0.237 | 0.590 | 0.395 | $1.93 \times 10^{-16}$ |
| Nipah virus | 24 | 478.67 | 0.03846 | -0.412 | 0.372 | 0.403 | $4.77 \times 10^{-7}$ |
| Average | 31.4 | 470.34 | 0.01408 | -0.284 | 0.558 | 0.268 | $4.99 \times 10^{-4}$ |

split of TimesX. We reserve them for OOD evaluation rather than including them in the main in-distribution benchmark.

**Sparse-event rare-disease variables.** We also construct five rare-disease variables to study sparse-event settings: Chagas disease, Huntington's disease, Guinea worm disease, Marburg virus disease, and Nipah virus. For these variables, the numeric component currently uses search-trend signals; we are actively looking for stable, regularly updated sources of case counts or incidence rates.

Figure 6 visualizes the search-trend signals for these five rare-disease variables. Even though the events are sparse, the series still show noticeable spikes around news or outbreak-related periods.

We further compute statistics over the constructed events for these rare-disease variables. Table 8 reports the number of events, the average event summary length, and several time-series characteristics. Compared with typical variables in TimesX (see Tables 11 and 15), the rare-disease variables indeed have fewer events, but still more than 30 events per disease between 2023 and mid-2025 on average. This suggests that our dataset agent remains effective even in sparse-event domains and can recover a reasonable amount of external context.

### A.6. Support for finetuning and future exploration

The main focus of this paper is to address the lack of an appropriate multimodal TSF evaluation benchmark. We see this as a key bottleneck before fully exploring finetuning strategies. At the same time, we aim to make TimesX a useful testbed for future work on finetuning TFMs and LLMs.

To support finetuning, we construct an additional dataset that covers the years 2018–2023 for the same 190 in-distribution variables, including both numeric series and textual contexts. This split is intended as a training resource, while the 2023–2025 split serves as the main evaluation period.

We have observed that adding a single ICL example already improves the performances in Table 7. This suggests that

finetuning or more advanced adaptation schemes on TimesX could be promising. A thorough study of finetuning TFMs and LLMs on TimesX would, however, require substantial additional effort in terms of computation and method design (including tokenization, alignment strategies, and loss functions), which is beyond the scope of this dataset-and-benchmark paper. We hope that the finetuning split, the refreshable leakage-free evaluation set, and our empirical findings can serve as a solid foundation for future work on finetuning models for multimodal time-series forecasting.

## B. Detailed Experiment Setup of Table 2

For evaluation cost consideration, we conduct the data leakage validation only on the variables in *SearchTrend* subset of TimesX. All other experimental settings follow Section 4.1.

## C. Details on Metadata and Calendar Context Construction

### C.1. Metadata Construction

For each variable in TimesX, we create a static metadata description that summarizes the essential attributes of the time series. The metadata is generated using a fixed template with three components: (1) the variable name and its measurement unit, (2) the domain the variable belongs to, and (3) the collection frequency and the target prediction window.

The general template is as follows:

> "Meta Info": "This time series records [*variable name and unit*] in the [*domain*] domain, with a collection frequency of [*frequency*]. Prediction target period: from [*start date*] to [*end date*]."

For example, for the gasoline price series in the Commodity Price domain, the metadata is:

> "Meta Info": "This time series records gasoline price (USD/Gal) in the Commodity Price domain, with a collection frequency of daily. Prediction

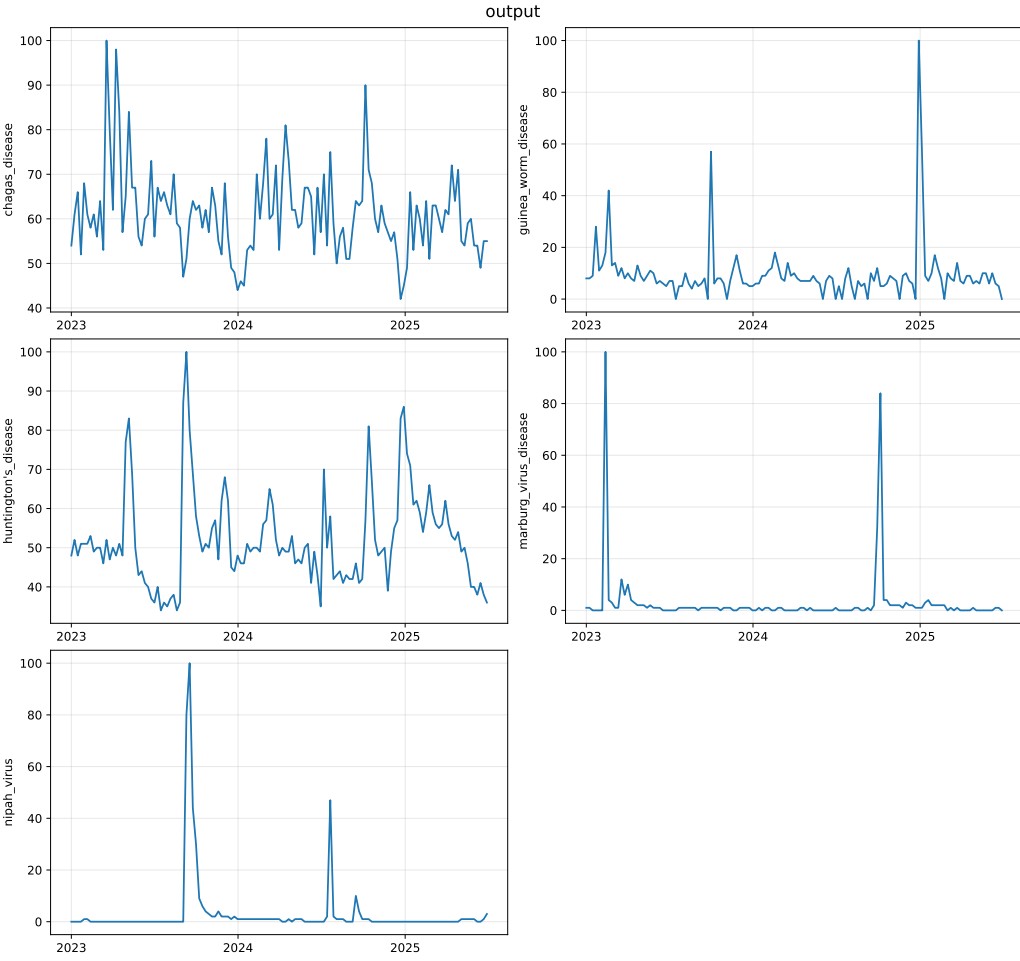

*Figure 6.* Search-trend signals for the five rare-disease variables in TimesX. Each panel shows a normalized trend series for one disease, where spikes often align with news or outbreak-related events.

target period: from 2024-09-01 to 2024-09-15."

## C.2. Calendar Context Construction

We generate calendar-based context features by automatically identifying holidays and special dates that fall within the forecasting horizon. Specifically, we use the `Python Holidays` library[4] to retrieve country- and region-specific holidays.

The construction process follows three steps:

1. Convert each time series into a sequence of (timestamp, value) pairs.

2. For each forecasting horizon, query the library to extract all holidays that overlap with the horizon window.

3. Format the results into textual annotations describing the holiday names and dates, which are then aligned

---

[4]https://pypi.org/project/holidays/

with the corresponding timestamps.

For example, if the prediction horizon is from 2024-09-01 to 2024-09-15, the generated calendar context includes:

"Upcoming holidays in the prediction window: Labor Day (2024-09-02)."

## D. Covariate Context Construction

To generate covariate-based textual contexts, we compute descriptive statistics for each covariate in the same domain as the target series. Specifically, for each covariate we calculate:

- The average and median value over the observation window.

- The maximum value and the corresponding date.

- The minimum value and the corresponding date.

- The overall trend (upward or downward).

These statistics are automatically extracted using simple Python scripts and converted into structured textual descriptions. For example, one covariate may be described as:

> "from 2024-06-01 to 2024-08-31: The average value of `Brent Crude Oil (USD/BBL)` is 81.5. The maximum value of 87.4 occurred on 2024-07-04, and the minimum value of 76.0 occurred on 2024-08-21, showing an overall downward trend."

### D.1. Missing-value handling

We handle missing values separately for the input and target windows. For target-window missing values, evaluation metrics are computed only on observed timestamps. For input-window missing values, short gaps are linearly interpolated for time-series foundation models. In the current data, the average maximum contiguous missing gap per sample is 0.7962. We also remove 3 anomalous samples whose maximum contiguous gap exceeds 6.

## E. Details of Textual Event Construction

Execution log demo is provided in Section F

**Stage 1: Time-Series-Aware Event Hypothesis Generation    Input.** A target variable with its numeric series $y_{1:T}$ and a target period $[t_s, t_e]$. The period is partitioned into fixed-length time blocks $\mathcal{B} = \{B_1, \ldots, B_M\}$ (e.g., week or month).

**Process.** For each block $B$, an LLM proposes an initial set of event hypotheses $H_B = \{h_1, \ldots\}$. Each $h$ contains a tentative title, a draft timestamp, involved entities, and at least one candidate source URL. We aim to ensure through multiple iterations that $H_B$ sufficiently explains the prominent movements in $y_t$ within $B$. Specifically, We define a peak set $\mathcal{P}_B$ from $y_t$ (by a standard peak detector with fixed hyperparameters). A peak $p \in \mathcal{P}_B$ is *covered* if at least one $h \in H_B$ is temporally aligned with $p$ (within a small window) and topically relevant to the target variable. The coverage is

$$\mathrm{cov}(H_B) \;=\; \frac{|\{p \in \mathcal{P}_B : p \text{ is covered by } H_B\}|}{|\mathcal{P}_B|}.$$

We keep querying the LLM to add hypotheses iteratively and stop when $\mathrm{cov}(H_B) \geq \theta$ or when a step limit $K_{\max}$ is reached. This rule balances completeness and cost without relying on unrestricted search.

In our ablation, this peak-based heuristic reduces construction cost by about 21%.

**Output.** For each block $B$, a hypothesis set $H_B$ with draft timestamps, entities, and seed URLs. All retrieval during Stage 1 is time-bounded by $[t_s, t_e]$ to keep the search window consistent with the evaluation period and to avoid leakage.

**Leakage control.** All queries use an explicit upper bound $t_e$. Sources with edited or republished pages after $t_e$ are kept only if the original publication date is within $[t_s, t_e]$ and the content is accessible in that state.

**Prompt**. The prompt for this role is detailed in Fig 7

**Stage 2: Rigorous Verification and Temporal Characterization    Input.** Hypotheses $\{H_B\}$ from Stage 1.

**Process.** A verification role call LLMs to re-fetch evidence under the same time bound $[t_s, t_e]$ and constructs a structured temporal view for each hypothesis. The verifier normalizes titles, resolves canonical entities, and extracts two dates: announcement_date and occurrence_date. It then assigns a temporal type from a closed set: *Scheduled* (announced in advance), *Contemporaneous* (announcement and occurrence are near in time), *Retrospective* (backward-looking report), *Predictive* (forward-looking signal), or *Mixed*. For each atomic claim, the verifier requires an accessible source URL (HTTP 200 at crawl time), stores the access date, and records a short quote that supports the extracted field. Multi-source cross-checking removes items with unresolved contradictions and de-duplicates near-duplicates by normalized title, entity set, and date tuple.

**Output.**    A verified event set with (announcement_date, occurrence_date, type), consolidated sources, and a confidence score that reflects agreement across sources and the precision of dates (day-level preferred over month-level). The final timestamp and type come with a concise rationale that explains corrections to the Stage 1 draft.

**Leakage control.** Verification refuses any evidence whose first-publication date is after $t_e$. If a page is updated after $t_e$ but preserves the original content and date within $[t_s, t_e]$, the verifier keeps the archived or cited original. Otherwise the evidence is discarded.

**Prompt**. The prompt for this role is detailed in Fig 8

**Stage 3: Conditional Enrichment for Narrative Depth Input.** Verified events from Stage 2.

**Process.** An evaluation role checks information sufficiency for forecasting. If key fields are missing (e.g., actors, locations, magnitudes, explicit dates, or links to related variables), the pipeline runs an iterative but bounded deep search under the same time bound $[t_s, t_e]$. Each step adds at most one new high-value source. The process stops when all

required fields are present or when a small step cap $L_{max}$ is reached. The reporting role then writes a concise narrative (a few sentences) that states what happened, when, who is involved, and why it likely relates to the target variable. Each factual sentence is grounded by one or more quotes with URLs.

**Prompt**. The prompt for this role is detailed in Fig 9.

The prompt for the final synthesizer role is detailed in 10

Across the construction pipeline, the Synthesizer discards about 32.20% of unverified candidate events.

Note that, in our configuration, we empirically set $K_{max} = 3$, $\theta = 90\%$, and $L_{max} = 3$. Under this setting, the system can efficiently construct high-quality corpora with reasonable runtime and cost. We also conduct a small-scale sensitivity test: when we increase $K_{max}$ from 3 to 5 on a subset of variables, the total number of accepted events increases by only about $2\%$. Overall, we encourage users to adjust these hyperparameters according to their budget and domain, while using our configuration as a default recommendation.

Under our current configuration (Gemini 2.5 Pro plus Gemini 2.5 Flash), the construction cost is about \$0.7 per variable per three-month time block, including reruns due to network errors. This cost can be further reduced by using open-source LLMs or by batching verification steps so that multiple candidate events share the same LLM calls.

```
prompt = f"""
You are a professional research analyst specializing in the {domain} field.
Your task is to use your web search capabilities to identify and structure
significant events related to '{keyword}' in {geography_str} that occurred
between {start_date} and {end_date}.

**Key Guidelines:**

1. **Source of Information:** Please base your responses on the information
   retrieved from your web search and general common sense. It is important
   to avoid relying on internal knowledge or generating speculative details
   (hallucinations).

2. **Date Extraction Principles:** It is helpful to distinguish between
   two key types of dates. If a date cannot be found from the sources,
   please use 'null'.
   * 'announcement_date': The date when the news about the event was
     **published or first announced**. For example, if a news article
     from **2024-01-10** announces an upcoming product launch.
   * 'occurrence_date': The date when the event **actually took place
     or is scheduled to take place**. For example, if the product launch
     mentioned above happens on **2024-02-02**.

3. **Event Type Classification:** Please classify the event into the
   following types based on its certainty and timing.
   * 'Scheduled Event': A high-certainty event that has been officially
     announced to occur at a future date.
   * 'Predictive Information': A lower-certainty piece of information
     about the future, such as an analyst forecast, a target price change,
     or a credible rumor.
   * 'Contemporaneous Event': An event that occurs at the same time it
     is announced, often unexpected.
   * 'Retrospective Report': An analysis or report about an event or
     period that has already passed.

4. **Geographic Focus:** Please focus on events within the {geography_str}
   region.

5. **Source Verifiability:** Each event should be supported by at least
   one verifiable, high-quality URL.

**Output Format:**

Please provide your response as a JSON array of event objects. Each object
in the array should conform to the following structure. If any field's value
cannot be determined from the sources, use 'null'.

```json
[
    {{
      "event_summary": "A concise, factual description of the event.",
      "announcement_date": "YYYY-MM-DD or null",
      "occurrence_date": "YYYY-MM-DD or null",
      "event_type": "Scheduled Event|Predictive Information|Contemporaneous Event|
          Retrospective Report or Mixed or null",
      "source_urls": ["url1", "url2"],
      "confidence_score": 0.8
    }}
]
```

Please ensure the entire response is only the valid JSON array, without
any surrounding text or explanations.
"""
```

*Figure 7.* Prompt used for Hypothesizer Role: Initial Event Discovery via LLM with Web Search Integration.

```
prompt = f"""
You are a meticulous fact-checker. Your task is to verify claims against
source content and identify invalid pages.

**CRITICAL: First determine if the source content is valid.**

**Important Definitions:**
- **content_date**: The date when the event itself occurred (e.g., product
  launch, announcement)
- **publish_date**: The original publication date of the source (NOT "updated"
  or "last modified" dates)

**Few-Shot Learning Examples:**

**Example 1 - Valid Content:**
Event Claim: "Apple Vision Pro will be available on February 2, 2024"
Source Text:
```
<h2>Apple Vision Pro Available in the U.S. on February 2</h2>
POSTED ON JANUARY 8, 2024
<p>Apple today announced Apple Vision Pro will be available beginning
Friday, February 2...</p>
<footer>Last updated: January 10, 2024</footer>
```
Expected Output:
```json
{{
  "page_status": "valid_content",
  "verified_statements": [
    {{
      "statement": "Apple Vision Pro will be available on February 2, 2024",
      "status": "Confirmed",
      "supporting_quote": "Apple Vision Pro will be available beginning Friday,
          February 2"
    }}
  ],
  "overall_timing": {{
    "content_date": "2024-02-02",
    "publish_date": "2024-01-08"
  }},
  "reasoning": "Valid press release content. Used original publish date (Jan 8),
      ignored 'last updated' footer."
}}
```

**Example 2 - 404 Error Page:**
Event Claim: "iPhone 16 rumors surface in March 2024"
Source Text:
```
<title>Page Not Found - TechNews</title>
<h1>404 - Page Not Found</h1>
<p>The page you're looking for doesn't exist.</p>
<div class="sidebar">Today's Hot Topics: July 28, 2025</div>
```
Expected Output:
```json
{{
  "page_status": "error_page_404",
  "verified_statements": [],
  "overall_timing": {{
    "content_date": null,
    "publish_date": null
  }},
  "reasoning": "This is a 404 error page with no valid content. Sidebar dates are
      irrelevant template content."
}}
```

**Now analyze the actual content:**

**1. The Event Claim to Analyze:**
{event_claim}

**2. The Evidence (Source Text):**
---
{source_evidence}
---

**Instructions:**
1. First, determine page_status: valid_content, error_page_404, access_denied,
   or login_wall
2. If page_status is NOT "valid_content", return empty verified_statements
   and null dates
3. If valid_content, decompose claim into atomic facts and verify each one
4. For dates: Use ORIGINAL publish dates, ignore "updated", "modified",
   or sidebar dates
5. Classify fact status: Confirmed, Anticipated, Speculation, or Not_Found

**JSON Output format:**
{{
  "page_status": "<valid_content|error_page_404|access_denied|login_wall>",
  "verified_statements": [
    {{
      "statement": "<atomic factual statement>",
      "status": "<Confirmed|Anticipated|Speculation|Not_Found>",
      "supporting_quote": "<exact quote from text or null>"
    }}
  ],
  "overall_timing": {{
    "content_date": "<YYYY-MM-DD when the event occurred or null>",
    "publish_date": "<YYYY-MM-DD when source was originally published or null>"
  }},
  "reasoning": "<Brief explanation of your analysis process>"
}}

Respond with ONLY the JSON object, no additional text.
"""
```

*Figure 8.* Prompt used for Verifier Role: Atomic Fact Verification with Evidence Matching.

```
prompt = f"""
You are a research strategist analyzing information gaps and planning next steps.

**IMPORTANT ACTION LIMIT**: Please limit your next_actions to a maximum of
{max_actions} items. Focus on the most critical information gaps that need
to be addressed. Each action should be high-quality and targeted.

**Original Event Claim:**
{original_claim}

**Currently Verified Information:**
{statements_summary}

**Your Task:**
1. **Analyze completeness**: Compare verified information against the
   original claim
2. **Identify information gaps**: List missing or unconfirmed facts
3. **Plan next actions**: For each gap, determine if it's common knowledge
   or requires search

For each information gap, classify as:
- **Common knowledge**: Facts that can be resolved internally (e.g.,
  "Apple's fiscal Q1 is Oct-Dec")
- **Requires search**: Facts needing external verification

**JSON Output format:**
{{
  "is_sufficient": <true if all key facts are confirmed, false otherwise>,
  "next_actions": [
    {{
      "info_gap": "<description of missing information>",
      "is_common_knowledge": <true|false>,
      "action_type": "<resolve_internally|search>",
      "resolved_answer": "<answer if common knowledge, null otherwise>",
      "query": "<search query if action_type is search, null otherwise>"
    }}
  ]
}}

Please respond with ONLY the JSON object.
"""
```

*Figure 9.* Prompt used for Enricher Role (Phase 1): Information Sufficiency Evaluation and Action Planning.

```
prompt = f"""
You are an Information Integration Specialist. Your task is to produce the
final, authoritative version of an event by reviewing all provided evidence.
Your summary must:

- Correct and enrich the original claim with additional verified details.
- For factual or scheduled events, prioritize the most authoritative sources.
- For subjective analyses or predictions, explicitly include multiple credible
  viewpoints, if available. Clearly acknowledging any conflicting or uncertain
  claims along with their sources, if available
- Accurately adjudicate or revise the event's announcement date (the date the
  news was published) and occurrence date (the actual date of the event),
  using web search if necessary to ensure accuracy.
- If you are unsure, use NA and avoid making up information.

**1. Original Event Claim:**
{event_summary}

**2. Detailed Factual Evidence:**
**Confirmed Facts:**
{confirmed_facts}

**Anticipated/Planned Facts:**
{anticipated_facts}

**Internal Knowledge Resolutions:**
{internal_facts}

**3. Detailed Timing Evidence from Sources:**
{timing_summary_detailed}

**4. Initial Date:**
The following dates are preliminary findings and do not represent 100% accuracy.
Please select the most reasonable date based on the content and use online
search tools if necessary.
- **All Publish Dates Found:** {unique_publish_dates}
- **All Content Dates Found:** {unique_content_dates}

**Your Final Task:**
Respond with ONLY a single JSON object. Do not add any text, explanations,
or markdown formatting before or after the JSON block.

**JSON Output Format:**
{{
  "final_summary_text": "<Your comprehensive summary here. This text should be well-
      written, accurate, detailed and reflect your final decision on the dates.>",
  "authoritative_dates": {{
    "announcement_date": "<The single, most credible YYYY-MM-DD publish date of
        content. If none, use NA.>",
    "occurrence_date": "<The single, most credible YYYY-MM-DD date when the event
        actually took place. If none, use NA.>"
  }},
  "reasoning_for_date_choice": "<A brief, one-sentence explanation for your date
      selection. e.g., 'Chose the earliest publish date from a primary news source
      .'>"
}}
"""
```

*Figure 10.* Prompt used for Synthesizer Role: Final Information Synthesis with Authoritative Date Determination.

# F. Detailed Execution Log: Cotton Price Case Study

To further elucidate the leakage prevention and verification mechanisms of the Hypothesizer-Verifier-Enricher (H-V-E) framework, we present a detailed step-by-step execution log for the variable "cotton price" over the historical window from 2024-01-01 to 2024-03-31.

PHASE 1: HYPOTHESIZER

- **Input:** Keyword: "cotton price"; Time Range: 2024-01-01 to 2024-03-31.

- **Action 1 (Peak Discovery):** The system analyzed the time-series data and identified 10 significant peaks requiring explanation (e.g., 2024-01-08, 2024-02-19, etc.).

- **Action 2 (Event Discovery):** The LLM (Gemini-2.5-Pro) performed an initial search and generated 14 candidate events. Examples include:

    - `Event 66e0`: "The USDA's January 2024 WASDE report..." (Source: cottongrower.com/...)
    - `Event 57e9`: "The USDA's weekly export sales report..." (Source: ccfgroup.com/...)
    - `Event a7a7`: "An early 2024 report highlighted... Panama Canal..." (Source: terrain.sc.eg/...)

PHASE 2 & 3: VERIFIER & ENRICHER (PARALLEL STREAMS)

The system spawned 14 independent verification tasks for the candidate events. We illustrate the robustness of the pipeline using three representative tasks:

**Task 1: Successful Verification (Event 66e0 - USDA WASDE Report)**

- **Verify:** Crawler accessed `cottongrower.com/...` (Status: Success).

- **Verify:** The `claim_verifier` agent (Gemini-2.5-Flash) cross-referenced the crawled content with the claim and confirmed consistency.

- **Enrich (Evaluate):** The `info_sufficiency_evaluator` deemed the information sufficient.

- **Finalize:** The `final_synthesis` agent generated the final summary.

- **Output:** Status **VERIFIED**.

**Task 2: Verification Failure and Discard (Event 57e9 - USDA Weekly Sales)**

- **Verify:** Crawler attempted to access `ccfgroup.com/...` (Status: Failed, error_page_404).

- **Enrich (Evaluate):** Evaluator deemed information insufficient, triggering an Action Plan.

- **Enrich (Act):** Agent generated a new query: "cotton price official announcement".

- **Enrich (Act - Leakage Prevention):** The system executed a time-bounded search.

    ```
    INFO - Executing Date-Restricted
    Search: 2024-01-05 to
    2024-01-19
    INFO - [SEARCH DEBUG] tbs:
    cdr:1,cd_min:01/05/2024,cd_max:01/19/2024
    ```

- **Verify (Loop 2):** System crawled new URLs (e.g., usda.gov), but the `claim_verifier` could not verify the specific statistics ("262,500 running bales") from the new sources.

- **Finalize:** Verification failed after max attempts.

- **Output:** Status **UNVERIFIED**. The event was discarded.

**Task 3: Recovery via Enrichment (Event a7a7 - Panama Canal)**

- **Verify:** Crawler attempted to access `terrain.sc.eg/...` (Status: Failed, net::ERR_NAME_NOT_RESOLVED).

- **Enrich (Evaluate):** Evaluator deemed information insufficient.

- **Enrich (Act):** Agent generated a new query: "cotton price... Panama Canal... details".

- **Enrich (Act - Leakage Prevention):** The system executed a time-bounded search to find alternative sources.

    ```
    INFO - Executing Date-Restricted
    Search: 2024-01-03 to
    2024-01-17
    INFO - [SEARCH DEBUG] tbs:
    cdr:1,cd_min:01/03/2024,cd_max:01/17/2024
    ```

- **Enrich (Act):** Search successfully retrieved valid new URLs (e.g., `windward.ai/...` and `porttechnology.org/...`).

- **Verify (Loop 2):** Crawler accessed `windward.ai/...` (Status: Success).

- **Verify (Loop 2):** The `claim_verifier` successfully verified the facts regarding the Panama Canal drought impact.

- **Finalize:** The `final_synthesis` agent generated the final summary using the verified facts from the new source.

- **Output:** Status **VERIFIED**.

# G. Details on Time Isolation and Event Timestamp Verification

Addressing the risk of sample-level information leakage—specifically, the inclusion of future events within the historical window—is a critical challenge in constructing time-series datasets. As stated in Section 4.1, our evaluation protocol strictly incorporates textual events with timestamps that precede the prediction window. Consequently, the integrity of our benchmark hinges on the Dataset Agent's capability to accurately annotate event timestamps.

Our Dataset Agent does not operate as a simple, unrestricted web search tool, but rather as a rigorous, multi-stage verification pipeline governed by the Hypothesizer-Verifier-Enricher framework (see Figure 2). To mitigate hallucinations and prevent timestamp errors, we design a three-tier correction mechanism:

- **Verifier (Phase 2a-2b):** Following the initial event list generation by LLM A in the Hypothesizer phase, the Verifier employs a web crawler to fetch content from specific URLs. It then utilizes LLM B to strictly validate whether the dates within the webpage content align with the proposed event, thereby filtering out discrepancies.

- **Enricher (Phase 3):** For details that the Verifier cannot conclusively confirm, the Enricher formulates search queries. Crucially, in Phase 3b, we enforce hard constraints on the search engine API (e.g., the `tbs` parameter in the Google Search API), restricting results to a time window of $\pm k$ days around the candidate event date. This API-level constraint provides a strong guarantee for timestamp validity.

- **Synthesizer (Phase 4):** LLM E aggregates all verified evidence to make a final adjudication. Any event that fails to achieve cross-source verification regarding its date is rejected.

### G.1. Manual Audit of Timestamp Accuracy

To empirically validate this mechanism, we conducted a manual audit on 50 randomly sampled events generated by the pipeline.

- In 47 cases (94%), the automatically annotated timestamps matched the human annotation exactly.

- In 2 cases (4%), the agent's timestamp was 1–2 days later than the human annotation. This represents a conservative error that does not constitute leakage.

- Only 1 case (2%) was dated 1 day earlier than the human annotation. Upon inspection, this discrepancy arose from ambiguity in defining the date for a complex event involving multiple sequential developments.

### G.2. Case Study: Correcting Reporting Lag

We observe that the agent is capable of reducing reporting lag while maintaining strict leakage prevention. We present a representative case study regarding Medicare drug costs to demonstrate this capability:

- **Initial Signal:** A news article reports on a study regarding rising drug costs on February 18 (Source: USC News https://hscnews.usc.edu/medicare-beneficiaries-face-much-higher-drug-costs-as-plans-shift-to-coinsurance).

- **Agent Action:** The agent traces the primary source cited within the news report.

- **Correction:** The agent successfully retrieves the original JAMA Health Forum article published on February 14 and corrects the event timestamp from February 18 to February 14.

This precision ensures that the event is correctly aligned with the historical window, capturing the earliest valid signal without violating temporal causality.

# H. Overview of TimesX

TimesX contains 20 domains and 200 variables in total (balanced design: $20 \times 10$). The collection window spans from **2022-01-01** to **2025-06-30** for the daily subset, and from **2023-01-01** to **2025-06-30** for the weekly subset.[5] Geographical coverage includes North America (United States, Canada, Mexico), Asia (for example, China, India), Europe (for example, United Kingdom, Norway), South America (for example, Brazil), and Africa.

**Variables and Frequencies.** **Weekly** series consist of Google Search Trend signals[7] across 12 domains. **Daily** series include (i) commodity prices[8] covering raw materials, energy, metals, and agriculture, and (ii) major USD exchange rates[9]. All series are aligned to a unified calendar per frequency, with clear missing-value handling policies documented in the dataset card.

## H.1. Structure of Textual Events

To maximize scientific utility and trust, we choose two complementary representations of the events: a **structured event corpus** for modeling and the **complete verification logs** for audit. The former provides clean, time-aligned annotations with compact narratives. The latter records the search queries, source URLs, access timestamps, and short evidence quotes produced by the verifier.

Each event is organized into two layers that match modeling needs and evidence needs:

**Core semantics for modeling.** A short, fact-checked narrative, distinct *announcement* and *occurrence* dates, and a categorical *event type* (*Scheduled*, *Contemporaneous*, *Retrospective*, *Predictive*, or *Mixed*).

**Evidence and provenance.** A small set of independent sources that support each claim, with verbatim text snippets and the corresponding access timestamps. For pages updated after the evaluation cut-off, archived versions or original publication records are linked.

# I. Breakdown of TimesX by Domains and Variables

Here are the acronyms we used for each domain when applicable:

- A&E: Arts & Entertainment;

- C&E: Climate & Environment;

- Econ: Economy;

- E. Tech: Electronic Technology;

- Fin: Finance;

- P&A: Pets & Animals;

- Pub. H.: Public Health;

- PPG: Public Policy & Governance;

- Sci: Science;

- Shop: Shopping;

- SSSG: Society Security & Social Good;

- Traf: Traffic;

- Crops: Crops & Staples;

- Energy: Energy & Fuels;

- Lvstk.: Livestock & Food Products;

- RMC: Raw Materials & Construction;

- SAM: Specialty & Advanced Materials;

- SHVM: Strategic & High-Value Materials;

- Curr: Currency.

Table 10, 11 and 12 list the variables under each of the 19 domains in TimesX.

---

[5]Weekly Google Trends series are included from 2022-01-01 due to stable availability and consistent retrieval settings.

[7]https://trends.google.com/trends/

[8]https://marketstack.com/

[9]https://frankfurter.dev/

*Table 9.* Dataset summary.

| Item | Description |
|---|---|
| Domains | 20 |
| Variables | 200 (balanced: about 10 per domain) |
| Frequencies | Weekly (Search Trend), Daily (Commodities, Exchange Rates) |
| Time span | 2022-01-01 to 2025-06-30[6] |
| Geographies | North America, Asia, Europe, South America, Africa (country and subnational coverage where applicable) |

| Domain | Variables (Search Keywords) |
|---|---|
| Arts & Entertainment | art_exhibitions; broadway_shows; comic_con; esports; film_festivals; food_&_wine_festivals; major_league_baseball; music_festivals; national_basketball_association; national_football_league |
| Climate & Environment | deforestation; drought; endangered_species; flooding; global_warming; heatwave; heavy_rainfall; marine_pollution; sustainable_fashion; water_scarcity |
| Economy | cost_of_living; federal_budget_deficit; government_spending; healthcare_costs; inflation; international_trade; minimum_wage; student_loans; taxes; unemployment_rate |
| Electronic Technology | alphabet; amazon; apple_inc.; artificial_intelligence; consumer_electronics; drones; meta_platforms; microsoft; nvidia; robotics |
| Finance | asset_management; cryptocurrency; financial_regulation; goldman_sachs; hedge_funds; investment_banking; mortgage_rates; private_equity; stock_market; venture_capital |
| Pets & Animals | animal_migration; animal_rescue; animal_welfare; beekeeping; biodiversity; invasive_species; marine_life; pest_control; pet_adoption; pet_health |
| Public Health | air_pollution; climate_change; diabetes; drug_overdose; food_safety; hiv_aids; infectious_disease; mental_health; obesity; opioid_crisis |
| Public Policy & Governance | carbon_emissions; federal_reserve; healthcare_policy; human_rights; immigration_reform; national_debt; presidential_election; refugee_support; renewable_energy; space_exploration; wildlife_conservation |
| Science | cancer_research; data_breach; earthquake; food_recall; gene_editing; meteor_shower; nobel_prize; quantum_computing; vaccine_research; volcanic_eruption |
| Shopping | air_conditioner; back_to_school; black_friday_deals; christmas_gifts; fashion_week; flu_shot; halloween_costumes; organic_food; ski_gear; tax_software |
| Society Security & Social Good | affordable_housing; cybersecurity; data_privacy; domestic_violence; gender_equality; homelessness; income_inequality; infrastructure_spending; protest; wildfires |
| Traffic | air_travel; autonomous_driving; electric_vehicle; formula_1; gas_prices; rocket_launch; tesla; tour_de_france; traffic_insurance; used_car |

*Table 10.* Search Trend (weekly) coverage by domain and variables (2023-01-01–2025-06-30). Each domain lists ten representative keywords.

| Domain | Variables (Commodity Price) |
|---|---|
| Crops & Staples | barley-INR-T; canola-CAD-T; cocoa-USD-T; cotton-USD-Lbs; diammonium-USD-T; oat-USD-Bu; potatoes-EUR-100KG; rice-USD-cwt; sugar-USD-Lbs; tea-INR-Kgs; urea-USD-T; wheat-USD-Bu |
| Energy & Fuels | bitumen-CNY-T; brent-USD-Bbl; coal-USD-T; ethanol-USD-Gal; gasoline-USD-Gal; methanol-CNY-T; naphtha-USD-T; propane-USD-Gal; rapeseed-EUR-T; uranium-USD-Lbs |
| Livestock & Food Products | beef-BRL-Kg; butter-EUR-T; cheese-USD-Lbs; coffee-USD-Lbs; corn-USD-BU; milk-USD-CWT; poultry-BRL-Kgs; salmon-NOK-KG; soybeans-USD-Bu; wool-AUD-100Kg |
| Raw Materials & Construction | aluminum-USD-T; copper-USD-Lbs; lead-USD-T; lumber-USD-1000_board_feet; polyethylene-CNY-T; polypropylene-CNY-T; rubber-USD_CENTS_-_Kg; steel-CNY-T; tin-USD-T; zinc-USD-T |
| Specialty & Advanced Materials | gallium-CNY-Kg; germanium-CNY-Kg; indium-CNY-Kg; magnesium-CNY-T; manganese-CNY-T; molybdenum-CNY-Kg; molybdenum-USD-Kg; polyvinyl-CNY-T; tellurium-CNY-Kg; titanium-CNY-KG; titanium-USD-KG |
| Strategic & High-Value Materials | cobalt-USD-T; gold-USD-t_oz; lithium-CNY-T; manganese-CNY-mtu; neodymium-CNY-T; nickel-USD-T; palladium-USD-t_oz; platinum-USD-t_oz; rhodium-USD-t_oz; silver-USD-t_oz |

*Table 11.* Daily dataset coverage by domain and variables (2022-01-01–2025-06-30). Each domain lists ten representative instruments.

| Domain | Variables (Exchange Rate) |
|---|---|
| Currency | USDtoAUD-ExchangeRate; USDtoBRL-ExchangeRate; USDtoCAD-ExchangeRate; USDtoCHF-ExchangeRate; USDtoGBP-ExchangeRate; USDtoHKD-ExchangeRate; USDtoINR-ExchangeRate; USDtoKRW-ExchangeRate; USDtoMXN-ExchangeRate; USDtoSGD-ExchangeRate |

*Table 12.* ExchangeRate: domains and variables

## J. Feature Definition of Time Series

Let a univariate series be $\{x_t\}_{t=1}^T$. We decompose it with STL into trend $T_t$, seasonal component $S_t$, and remainder $R_t$:

$$x_t = T_t + S_t + R_t. \tag{1}$$

Define the de-trended series $x_t^{\text{detr}} = x_t - T_t$ and the de-seasonalized series $x_t^{\text{deseas}} = x_t - S_t$.

**Seasonality**

$$\text{Seasonality} = \max\left(0,\ 1 - \frac{\text{Var}(R_t)}{\text{Var}(x_t^{\text{detr}})}\right). \tag{2}$$

Higher values mean a clearer periodic pattern explains more variance in $x_t$.

**Trend**

$$\text{Trend} = \max\left(0,\ 1 - \frac{\text{Var}(R_t)}{\text{Var}(x_t^{\text{deseas}})}\right). \tag{3}$$

Higher values mean a smoother long-term trend explains more variance in $x_t$.

**Nonstationarity**  We report the Augmented Dickey–Fuller $p$-value Liu et al., 2022.

**Short-term distributional change (Short_term_jsd)**  Using a short window of length $w_s = 30$, for each window $W$ form a histogram estimate $\hat{p}_W$ on fixed bins and a Gaussian reference $\hat{q}_W = \mathcal{N}(\mu_W, \sigma_W^2)$ discretized on the same bins, where $\mu_W$ and $\sigma_W$ are the window mean and standard deviation. The Jensen–Shannon divergence in window $W$ is

$$\text{JSD}(\hat{p}_W, \hat{q}_W) = \tfrac{1}{2}\,\text{KL}\!\left(\hat{p}_W \,\Big\|\, \tfrac{\hat{p}_W + \hat{q}_W}{2}\right) + \tfrac{1}{2}\,\text{KL}\!\left(\hat{q}_W \,\Big\|\, \tfrac{\hat{p}_W + \hat{q}_W}{2}\right) \tag{4}$$

The metric is the average over all windows:

$$\text{Short\_term\_jsd} = \frac{1}{N_w} \sum_W \text{JSD}(\hat{p}_W, \hat{q}_W). \tag{5}$$

Larger values indicate that short-term empirical distributions deviate more from a Gaussian shape.

**Shifting**  It summarizes typical level changes while retaining the influence of rare but large deviations.

**Transition**  Discretize $\{x_t\}$ into three equiprobable states $s_t \in \{1, 2, 3\}$ (tertiles). Let $\pi_i = \Pr(s_t = i)$ and $T_{ij} = \Pr(s_{t+1} = j \mid s_t = i)$. The score is the sum of diagonal covariances between successive states:

$$
\begin{aligned}
\text{Transition} &= \sum_{i=1}^{3}\Big[\Pr(s_t = i, s_{t+1} = i) \\
&\qquad - \Pr(s_t = i)\Pr(s_{t+1} = i)\Big] \\
&= \sum_{i=1}^{3}\left(\pi_i T_{ii} - \pi_i^2\right).
\end{aligned}
\tag{6}
$$

Higher values indicate that the process tends to stay in the same state more often than expected by chance.

Implementation details are in https://github.com/decisionintelligence/TFB.

## K. Feature Analysis of TimesX

The feature of both time series and text aggregated by domain is summarized in Table 13.

The feature aggregated by data source is summarized in Table 14

## L. Visualization of Numeric Dataset in TimesX by Domain

We provide visualizations of the numeric time series in TimesX grouped by domain, to give an overview of the series patterns and scales across domains (Figs. 13–31).

## M. PCA and t-SNE visualizations

In this section we visualize the diversity of TimesX in both the textual and numeric spaces. For textual features, we compute embeddings for each variable's context and then apply PCA and t-SNE to obtain two-dimensional representations. For numeric features, we extract summary statistics or learned representations of each time series and apply the same dimensionality-reduction procedures. We color the points either by domain or by data source.

Across all plots we make two consistent observations. First, there is clear diversity in both the textual and numeric spaces, with points spread over the two-dimensional maps rather than concentrating in a few tight clusters. Second, the three data sources form distinct clusters in feature space, which supports the value of integrating multiple sources to achieve broad coverage and diversity.

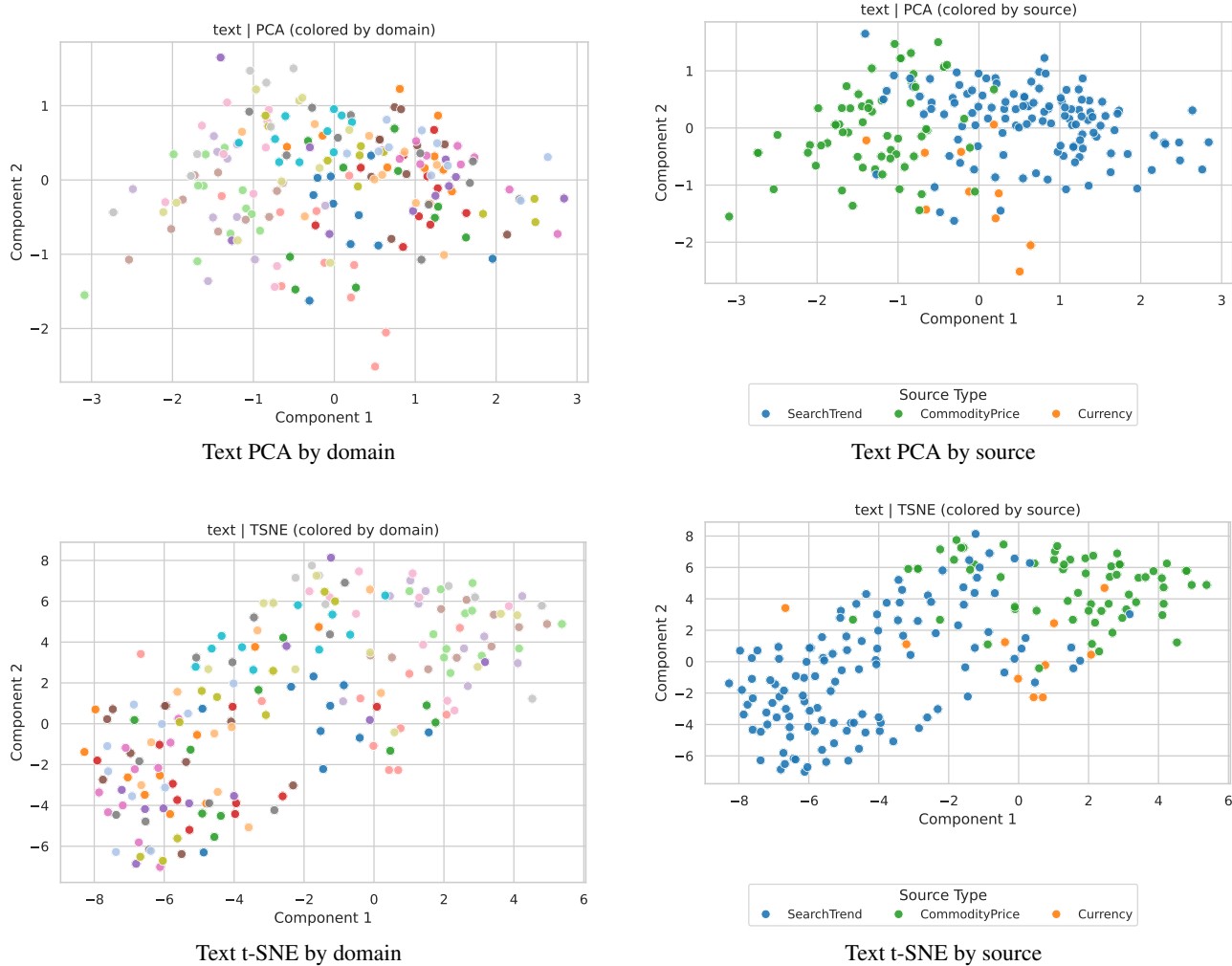

*Figure 11.* PCA and t-SNE visualizations of textual features in TimesX. Each point corresponds to a variable, colored either by domain or by data source.

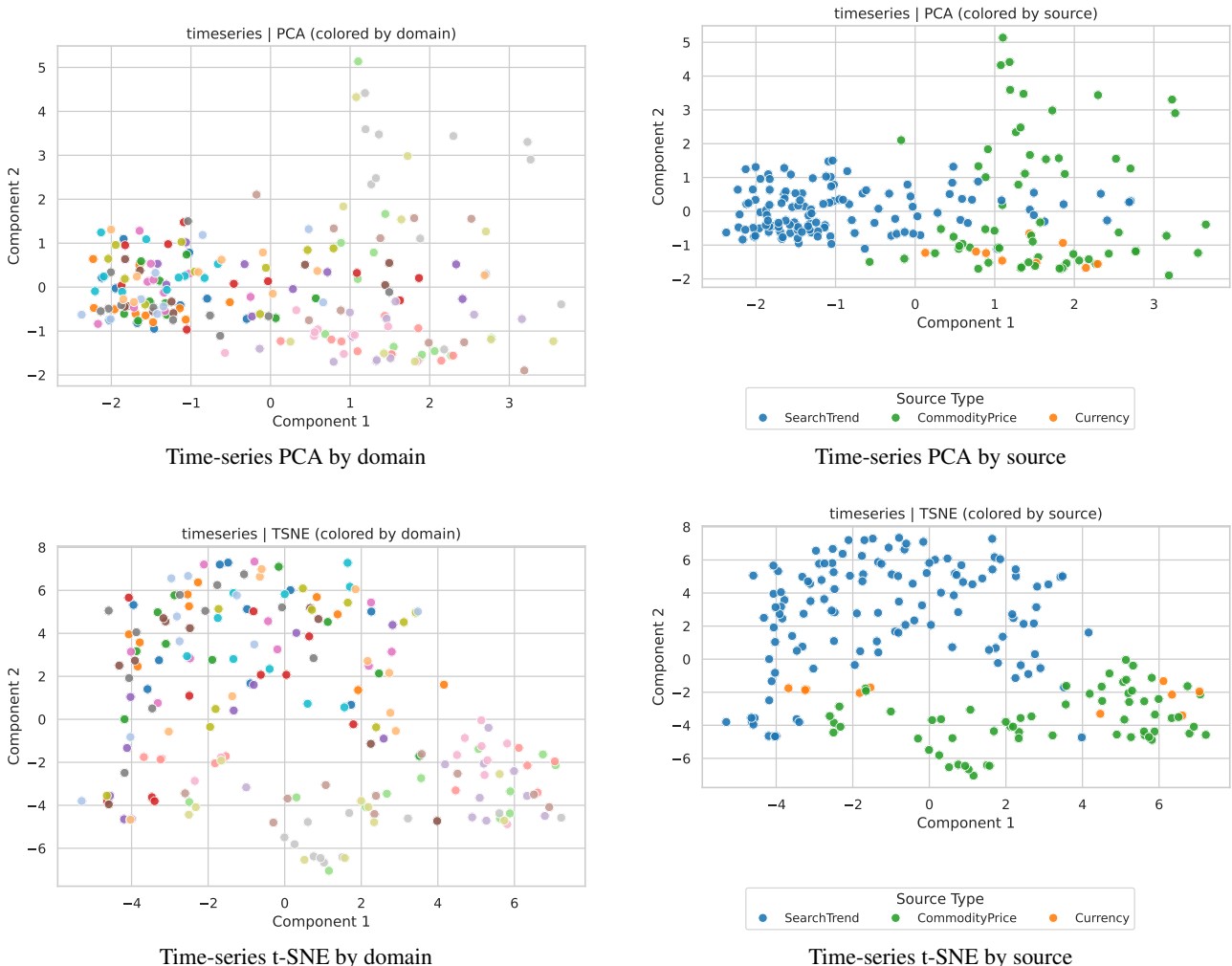

*Figure 12.* PCA and t-SNE visualizations of numeric time-series features in TimesX. Each point corresponds to a variable, colored either by domain or by data source.

| Domain | Text features | | Numeric features | | | | | |
|---|---|---|---|---|---|---|---|---|
| Domain | AvgEventsCount | AvgEventSummaryLen | Transition | Shifting | Seasonality | Trend | NonStationarity | Short_term_jsd |
| Arts & Entertainment | 105.7000 | 711.2444 | 0.0190 | 0.0832 | 0.8596 | 0.4155 | 0.0048 | 0.1905 |
| Climate & Environment | 98.1000 | 815.6727 | 0.0216 | 0.1514 | 0.8904 | 0.3970 | 0.0010 | 0.1463 |
| Crops & Staples | 51.3333 | 590.8046 | 0.0857 | -0.3797 | 0.6621 | 0.9228 | 0.4201 | 0.2056 |
| Currency | 111.0000 | 639.6416 | 0.0926 | 0.0923 | 0.5879 | 0.8801 | 0.3396 | 0.0595 |
| Economy | 108.9000 | 733.3640 | 0.0310 | 0.0647 | 0.9030 | 0.5522 | 0.0557 | 0.1984 |
| Electronic Technology | 115.7000 | 783.1571 | 0.0462 | 0.4609 | 0.8191 | 0.6149 | 0.2529 | 0.3264 |
| Energy & Fuels | 53.7000 | 615.0142 | 0.1007 | -0.6639 | 0.6575 | 0.8752 | 0.3803 | 0.1231 |
| Finance | 102.3000 | 777.7485 | 0.0773 | 0.5291 | 0.8193 | 0.6827 | 0.3351 | 0.2984 |
| Livestock & Food Products | 55.4000 | 596.3044 | 0.0751 | -0.2187 | 0.6389 | 0.8821 | 0.4700 | 0.2769 |
| Pets & Animals | 103.1000 | 815.8348 | 0.0418 | 0.2342 | 0.8459 | 0.5464 | 0.1143 | 0.2487 |
| Public Health | 119.1000 | 848.1958 | 0.0184 | 0.0750 | 0.8370 | 0.3992 | 0.0016 | 0.2419 |
| Public Policy & Governance | 96.4545 | 794.9547 | 0.0460 | 0.3298 | 0.8756 | 0.4869 | 0.0441 | 0.1710 |
| Raw Materials & Construction | 49.8000 | 654.2472 | 0.0582 | -0.5388 | 0.6690 | 0.8925 | 0.2443 | 0.0805 |
| Science | 98.2000 | 789.5100 | 0.0411 | 0.2402 | 0.7741 | 0.3854 | 0.1114 | 0.3549 |
| Shopping | 58.4000 | 748.0253 | 0.0411 | 0.0815 | 0.9724 | 0.4336 | 0.0037 | 0.3374 |
| Society Security & Social Good | 108.0000 | 841.8060 | 0.0499 | 0.4071 | 0.8296 | 0.4276 | 0.2003 | 0.2484 |
| Specialty & Advanced Materials | 28.8182 | 672.6066 | 0.1002 | -0.1111 | 0.6388 | 0.9281 | 0.3969 | 0.4774 |
| Strategic & High-Value Materials | 58.8000 | 683.1720 | 0.0918 | -0.1931 | 0.6468 | 0.8971 | 0.4695 | 0.2657 |
| Traffic | 94.0000 | 766.4479 | 0.0517 | -0.1120 | 0.8239 | 0.4905 | 0.1612 | 0.3555 |

*Table 13.* Domain-level textual and numeric features. "AvgEventSummaryLen" is the average number of characters in the event summary text within each domain. The numeric feature set follows TFB (Qiu et al., 2024). The exact computation is detailed in Section J

| Data Source | Numeric features | | | | | |
|---|---|---|---|---|---|---|
| Data Source | Transition | Shifting | Seasonality | Trend | NonStationarity | Short_term_jsd |
| CommodityPrice | 0.09 | -0.35 | 0.65 | 0.90 | 0.40 | 0.24 |
| ExchangeRate | 0.09 | 0.09 | 0.59 | 0.88 | 0.34 | 0.06 |
| SearchTrend | 0.04 | 0.21 | 0.85 | 0.49 | 0.11 | 0.26 |

*Table 14.* Data-source-level mean of numeric characteristics (Transition, Shifting, Seasonality, Trend, NonStationarity (ADF p-value), Short_term_jsd).

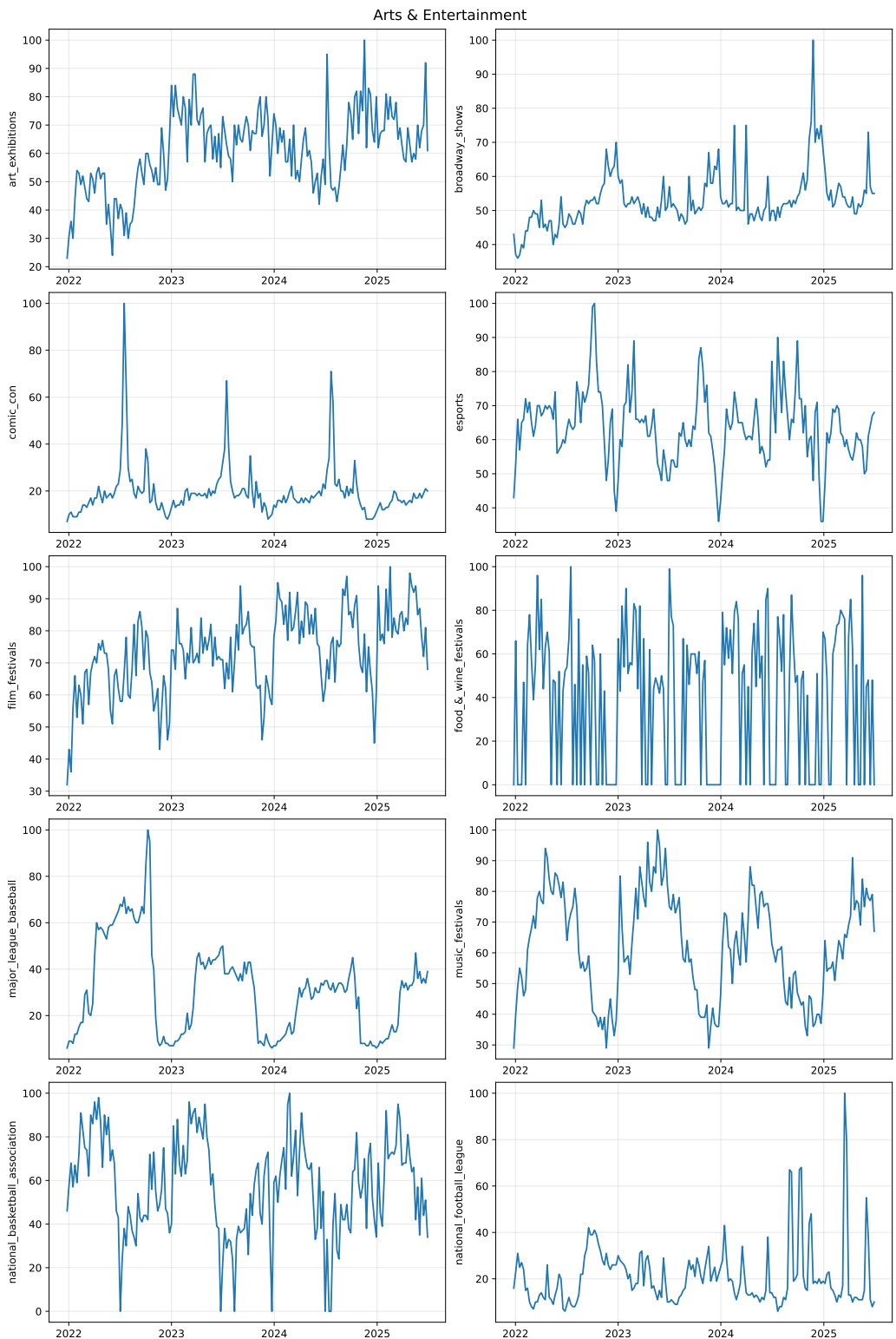

*Figure 13.* Numeric series visualization for the Arts and Entertainment domain.

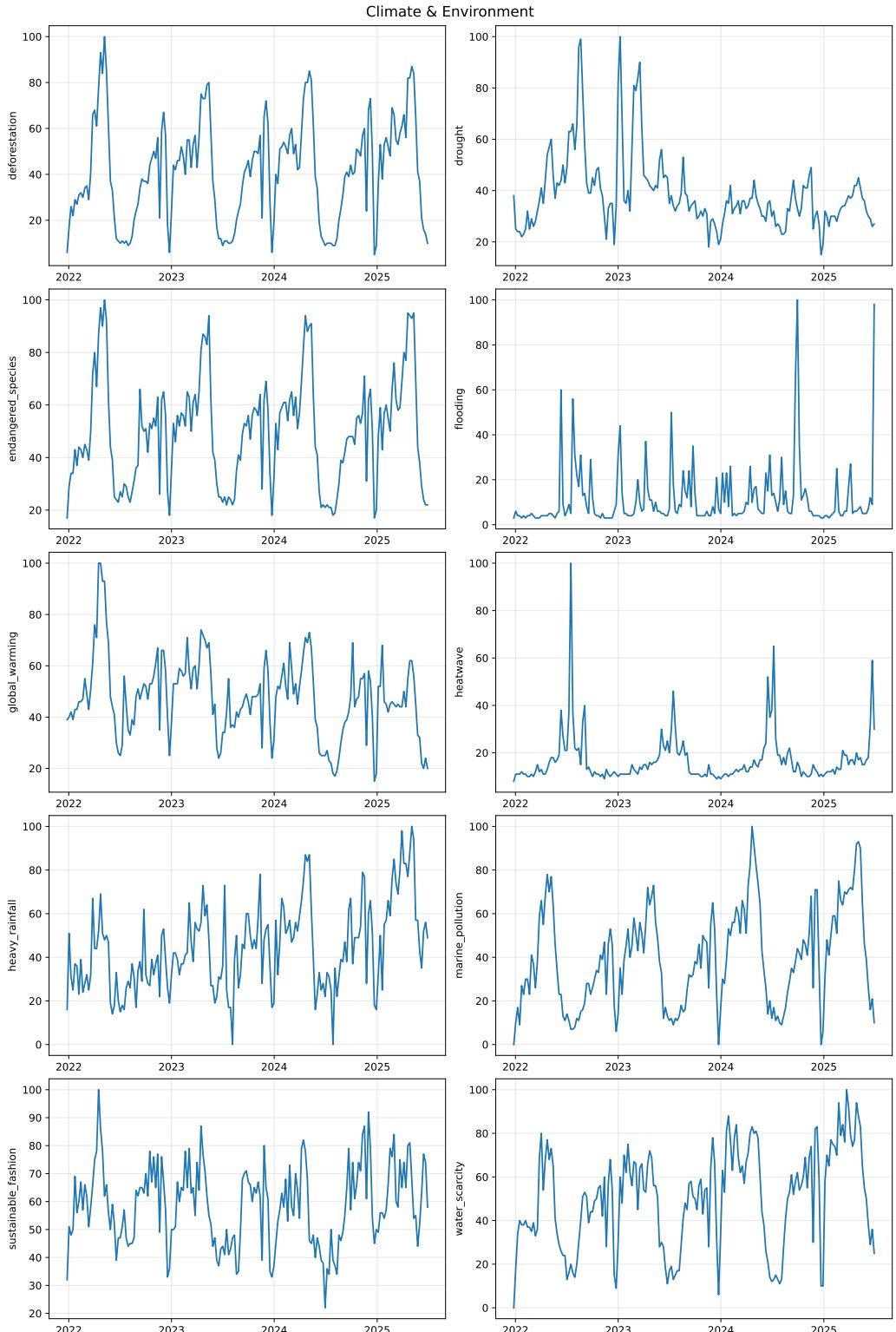

*Figure 14.* Numeric series visualization for the Climate and Environment domain.

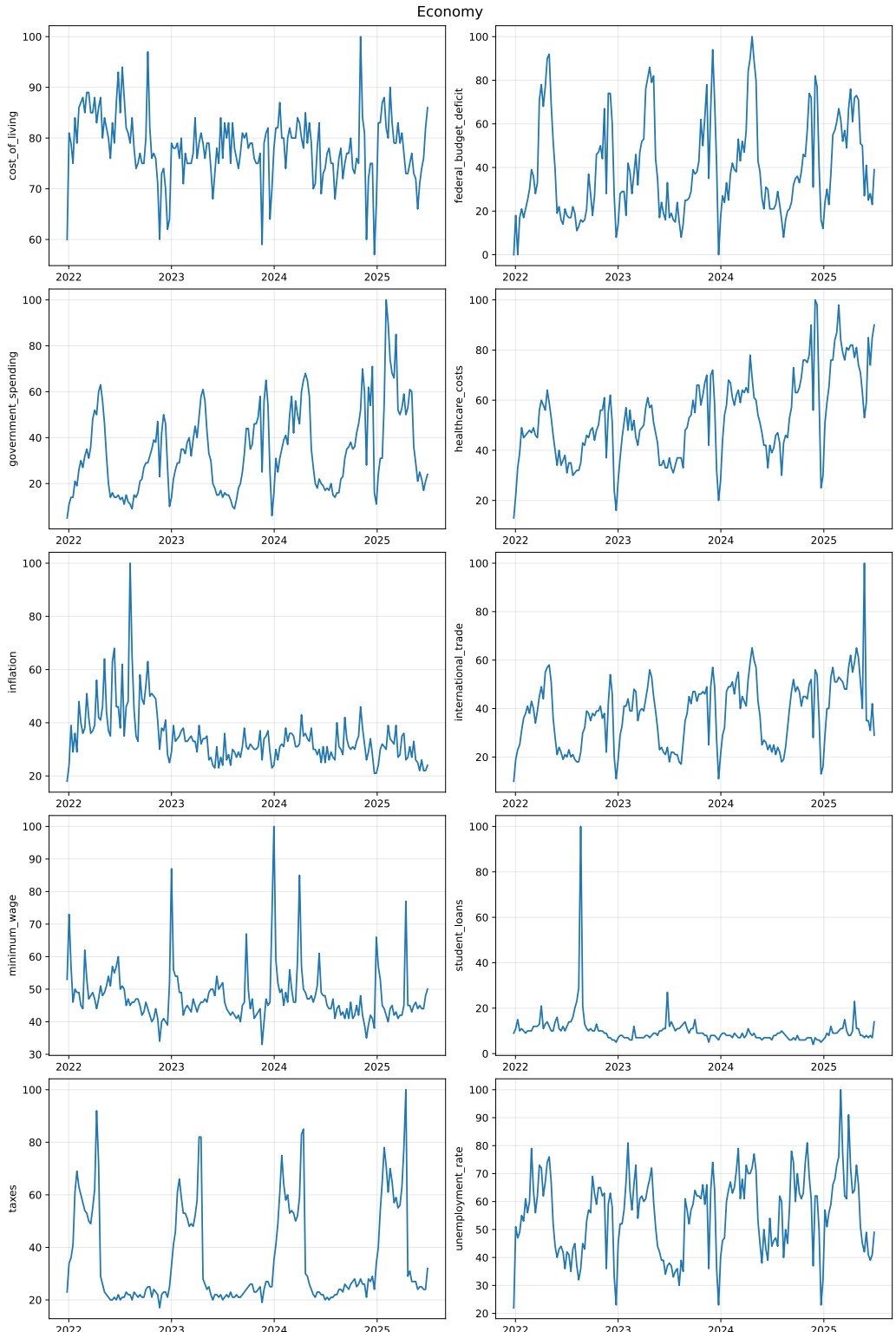

*Figure 15.* Numeric series visualization for the Economy domain.

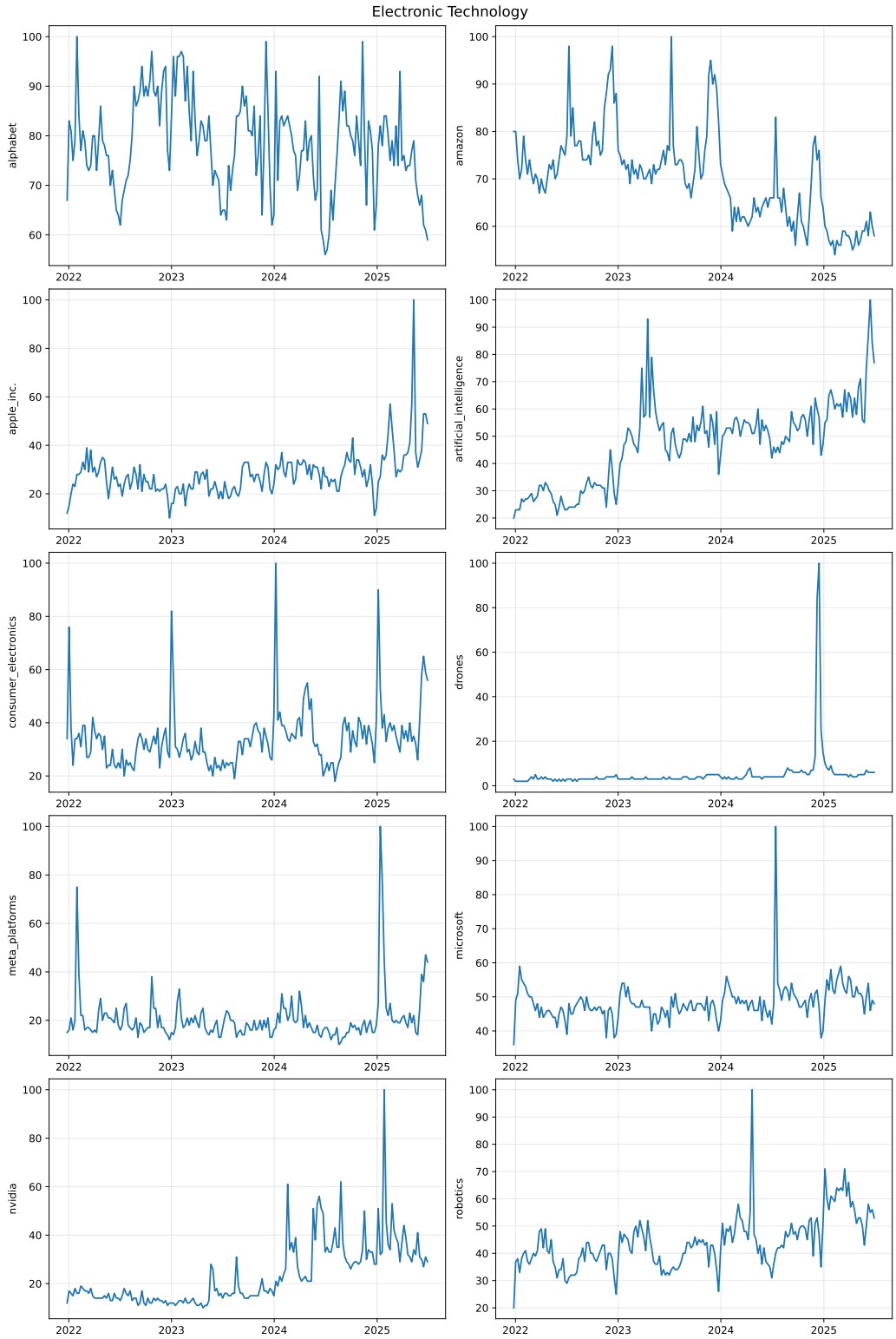

*Figure 16.* Numeric series visualization for the Electronic Technology domain.

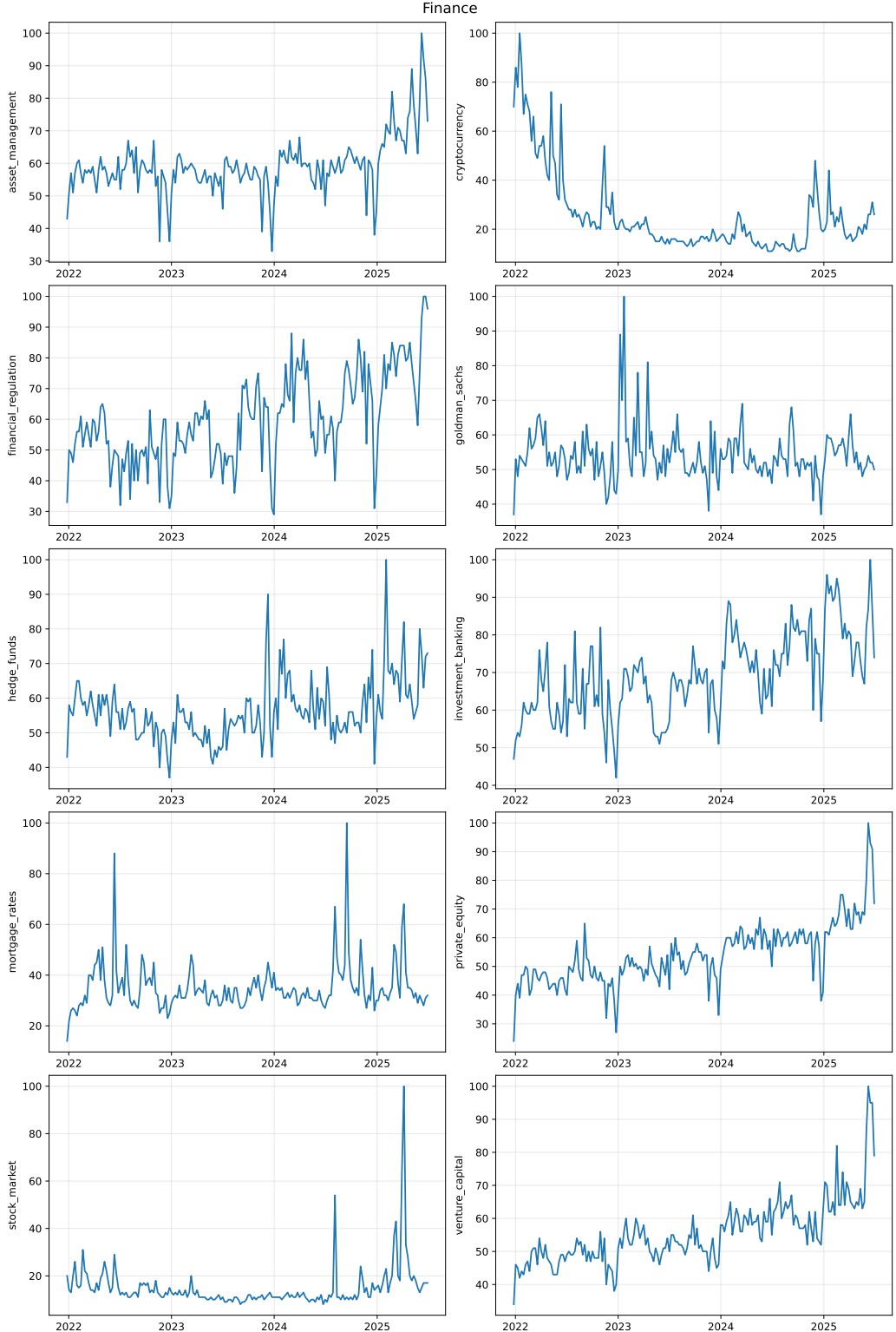

*Figure 17.* Numeric series visualization for the Finance domain.

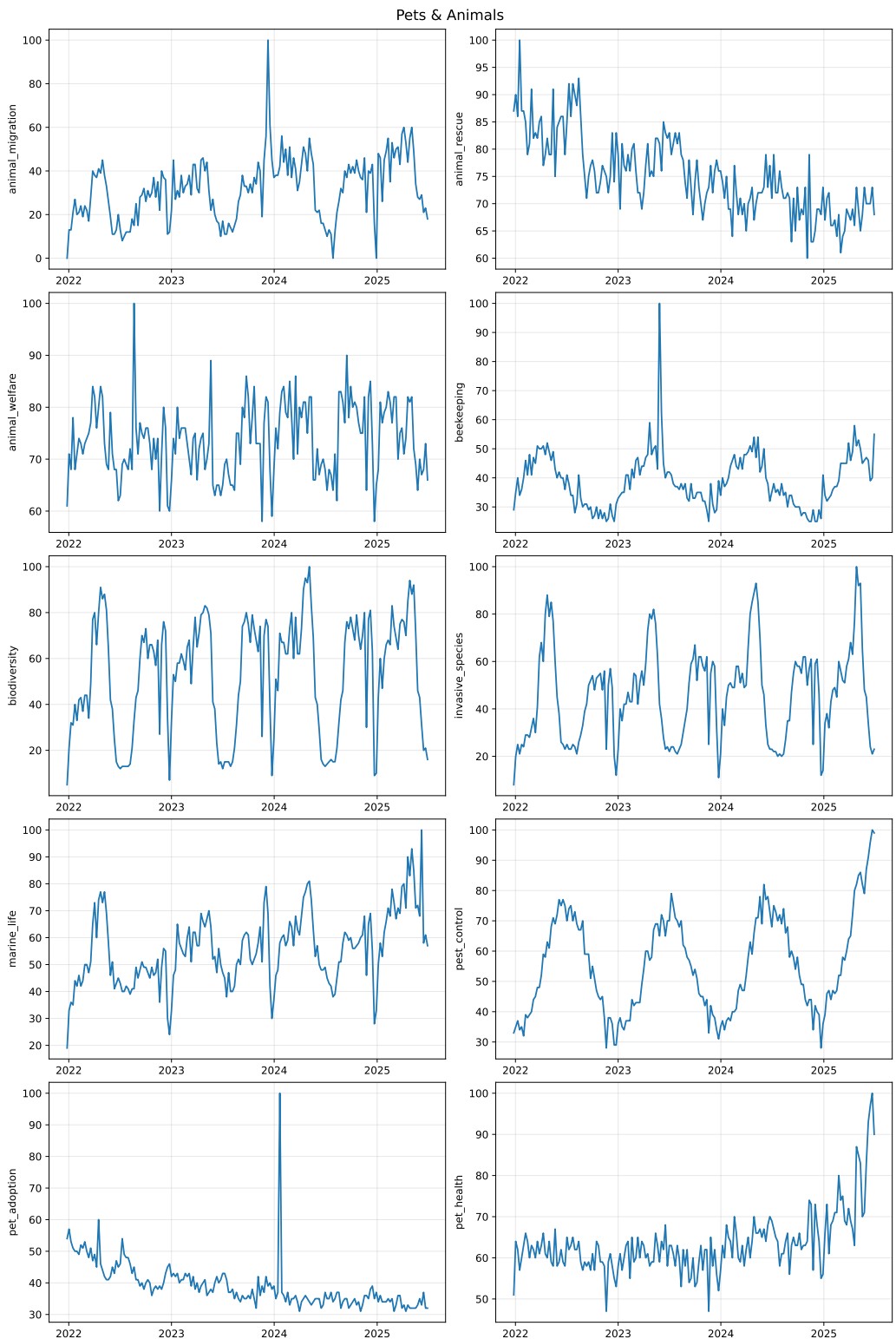

*Figure 18.* Numeric series visualization for the Pets and Animals domain.

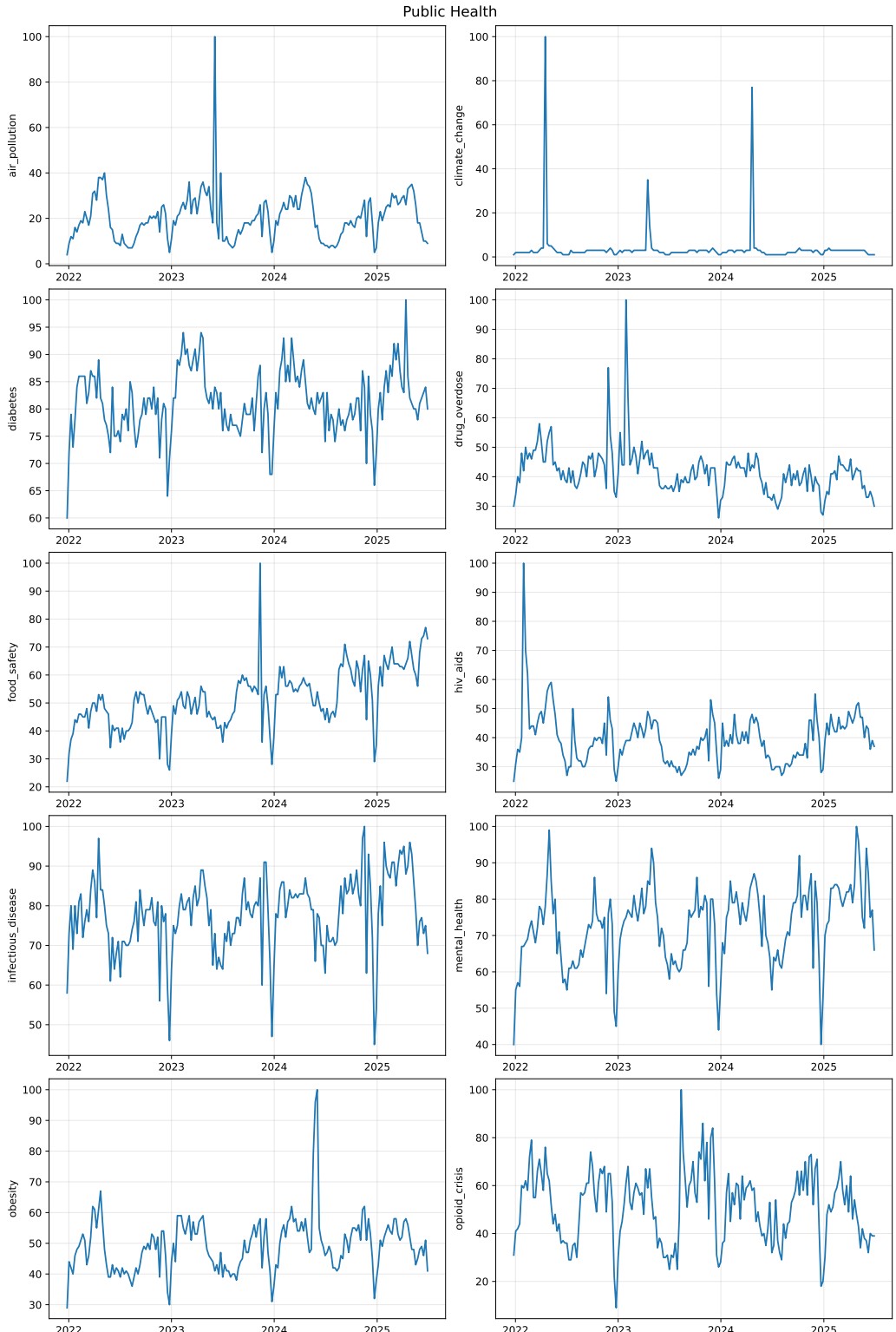

*Figure 19.* Numeric series visualization for the Public Health domain.

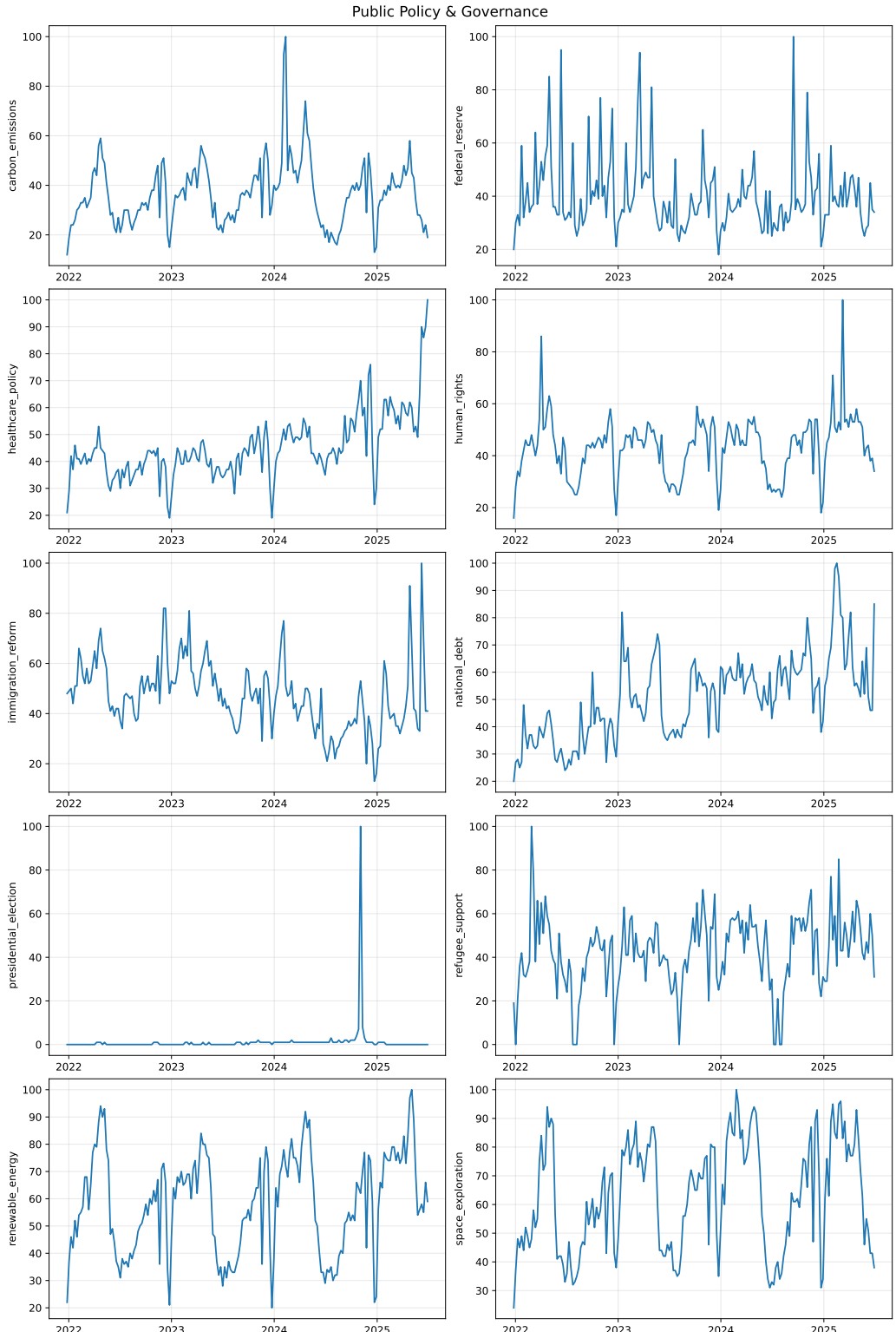

*Figure 20.* Numeric series visualization for the Public Policy and Governance domain.

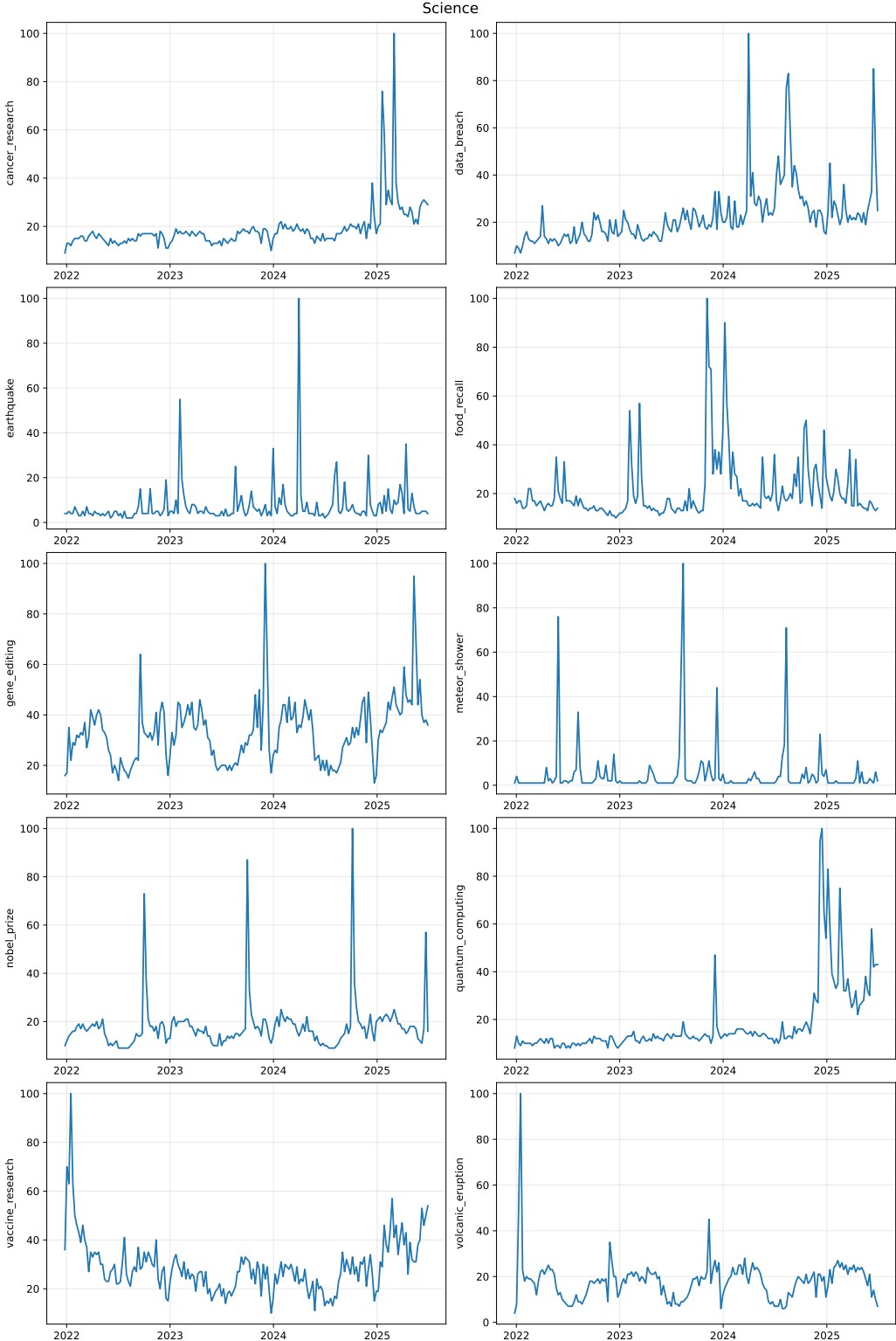

*Figure 21.* Numeric series visualization for the Science domain.

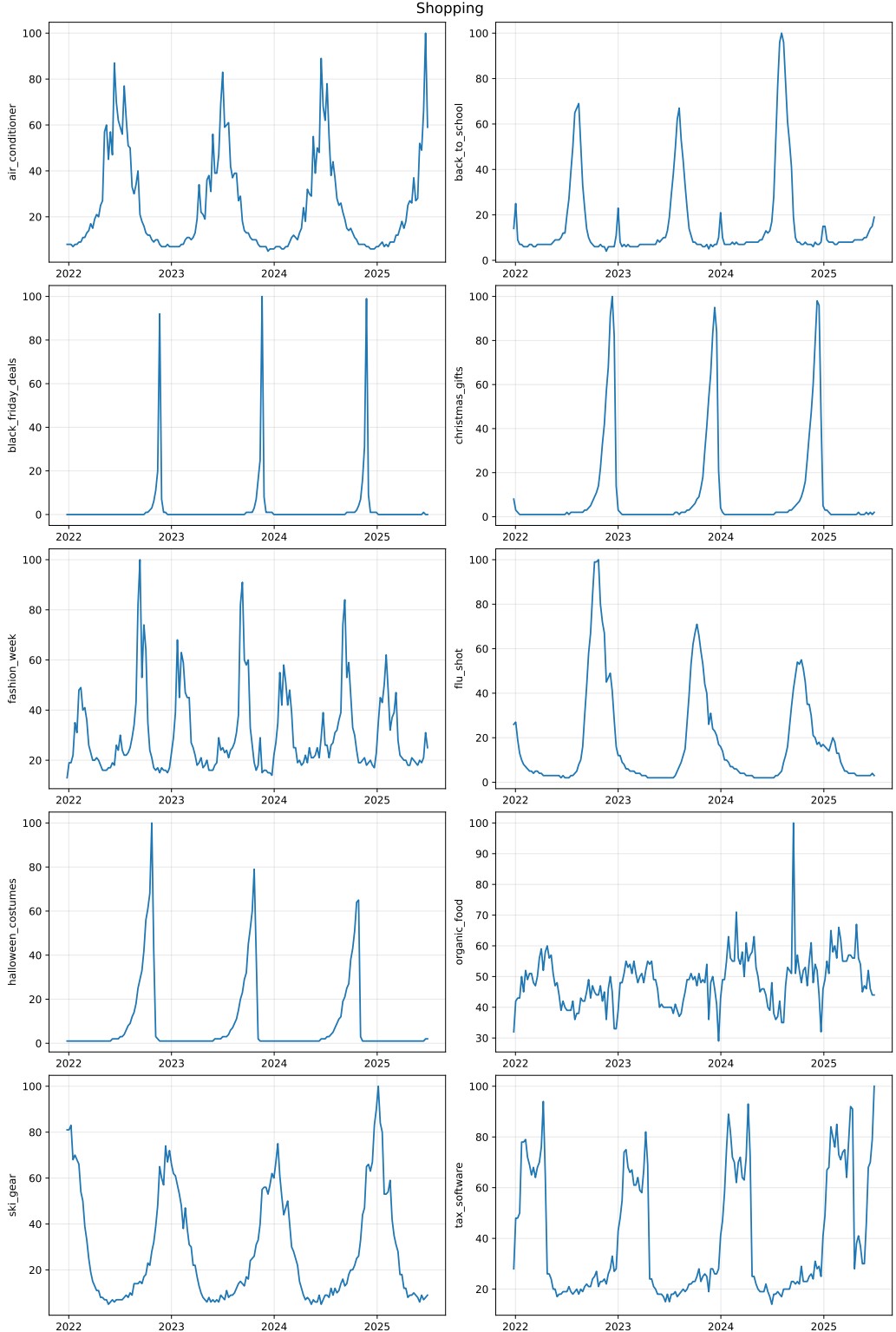

*Figure 22.* Numeric series visualization for the Shopping domain.

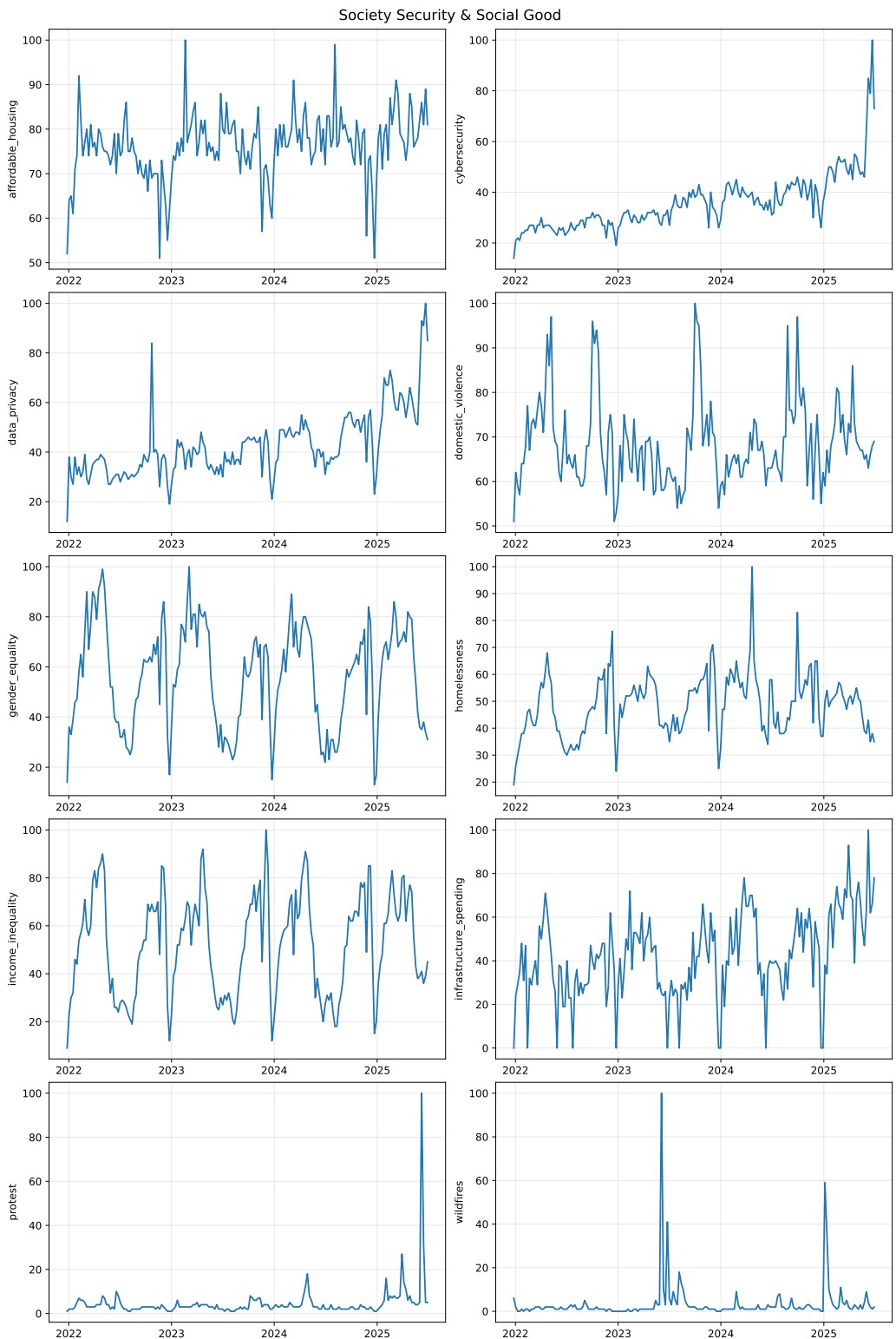

*Figure 23.* Numeric series visualization for the Society, Security, and Social Good domain.

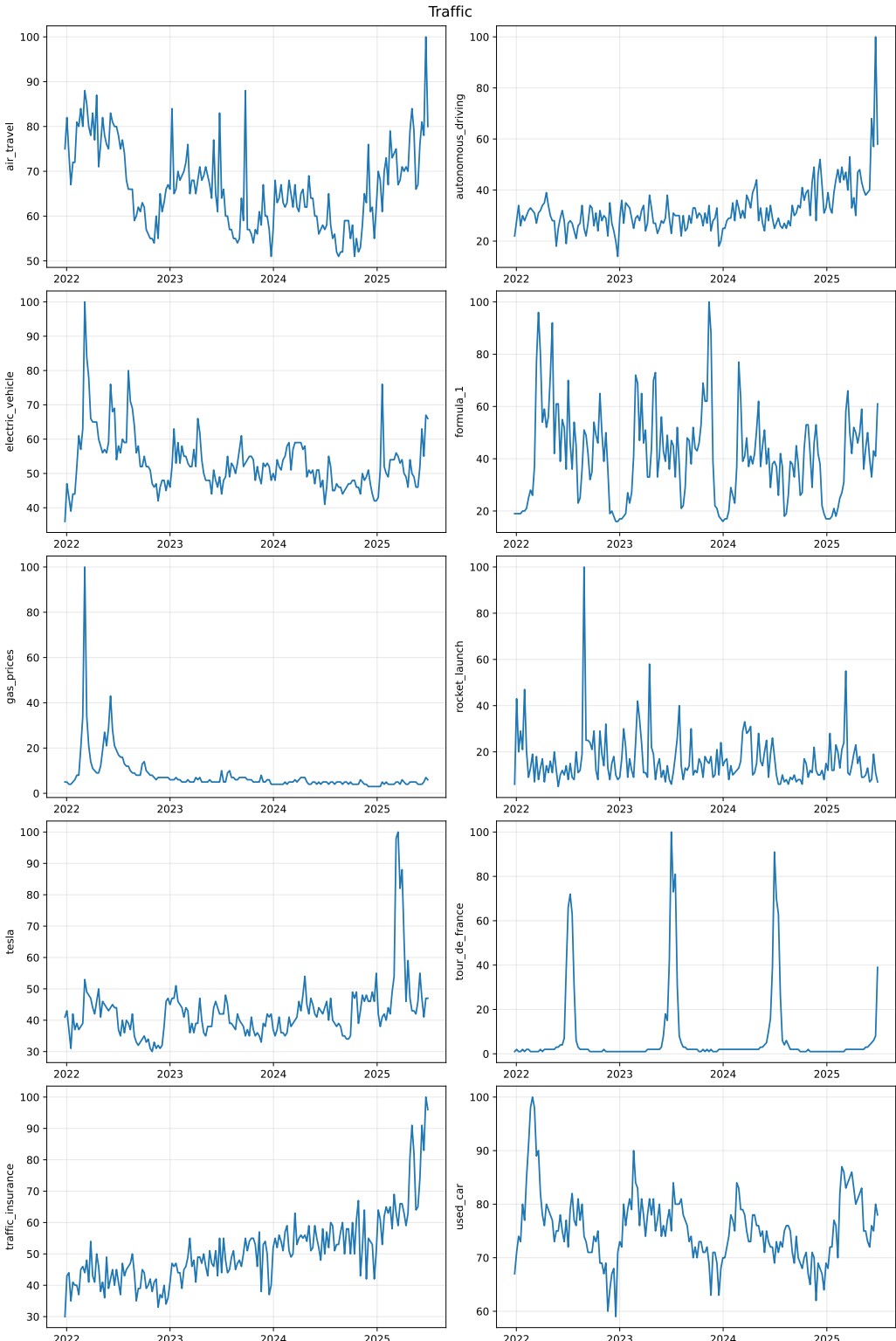

*Figure 24.* Numeric series visualization for the Traffic domain.

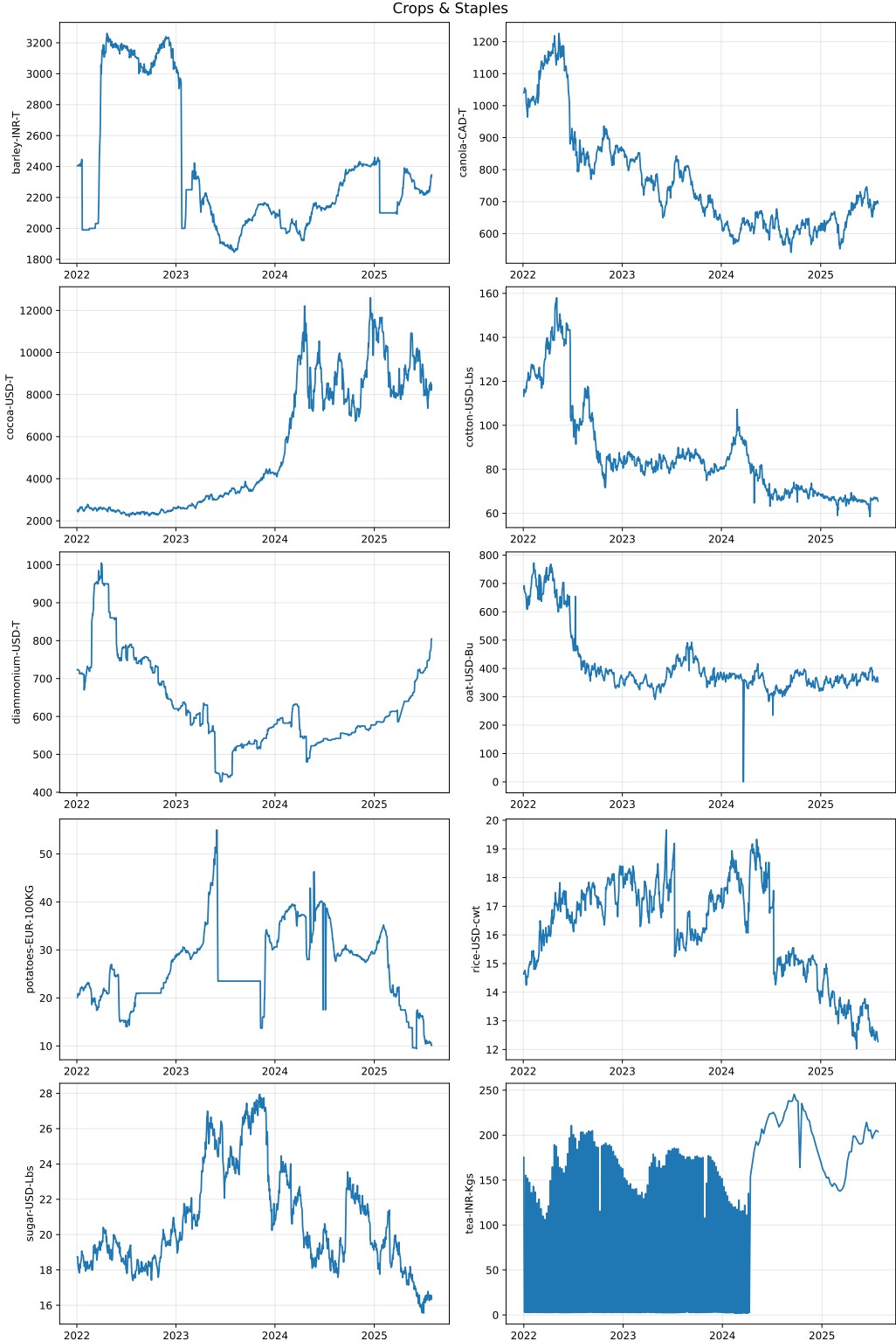

*Figure 25.* Numeric series visualization for the Crops and Staples domain.

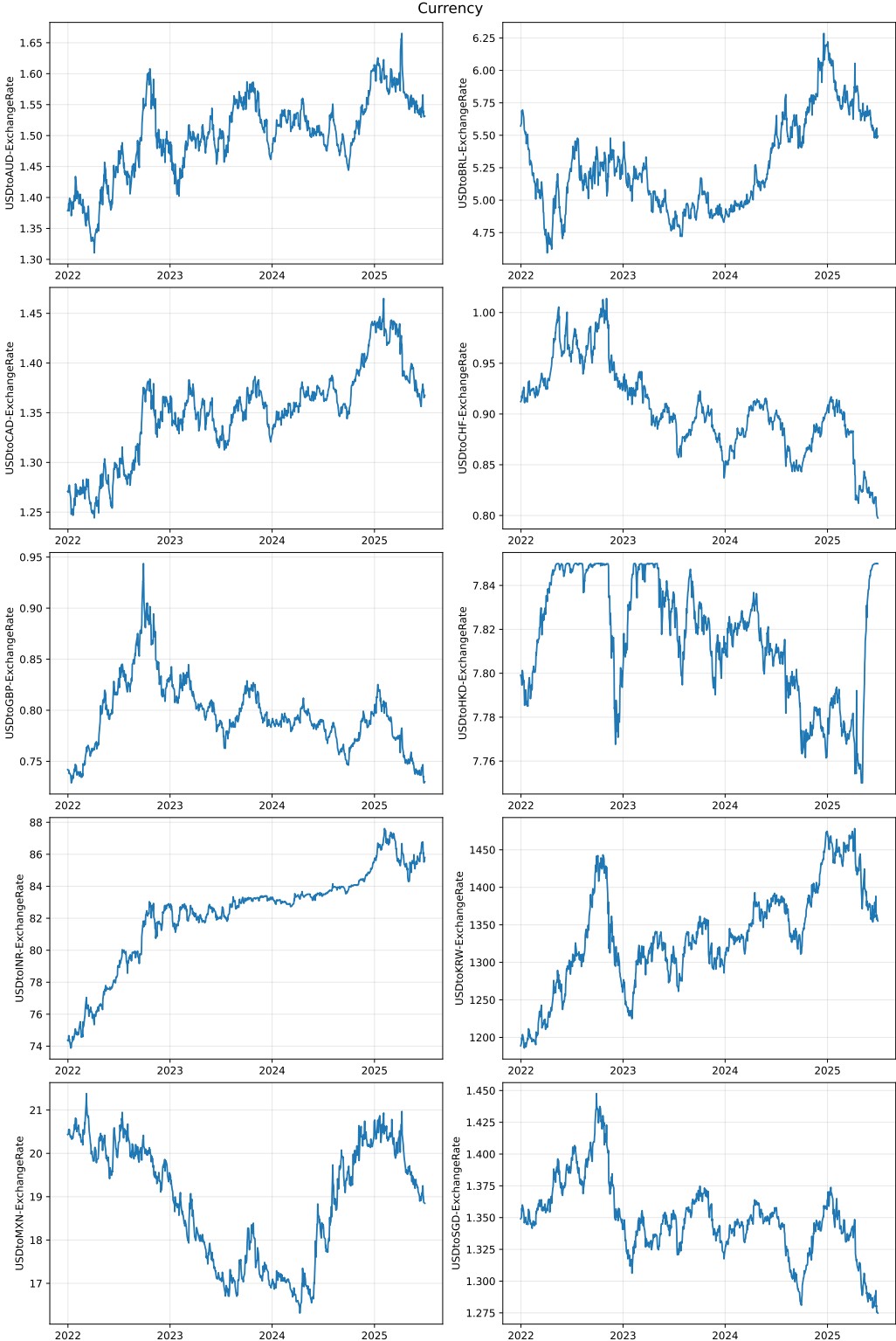

*Figure 26.* Numeric series visualization for the Currency domain.

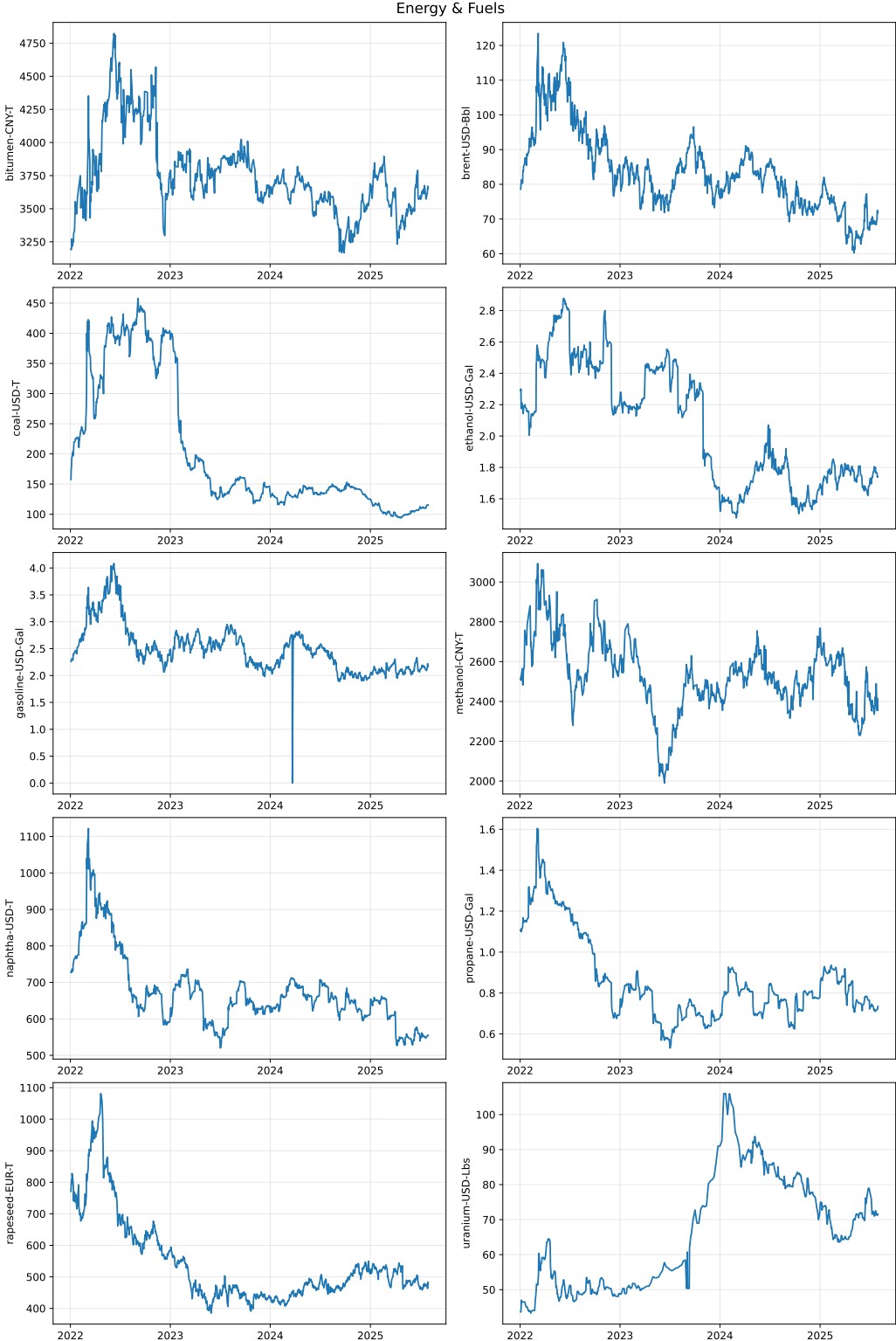

*Figure 27.* Numeric series visualization for the Energy and Fuels domain.

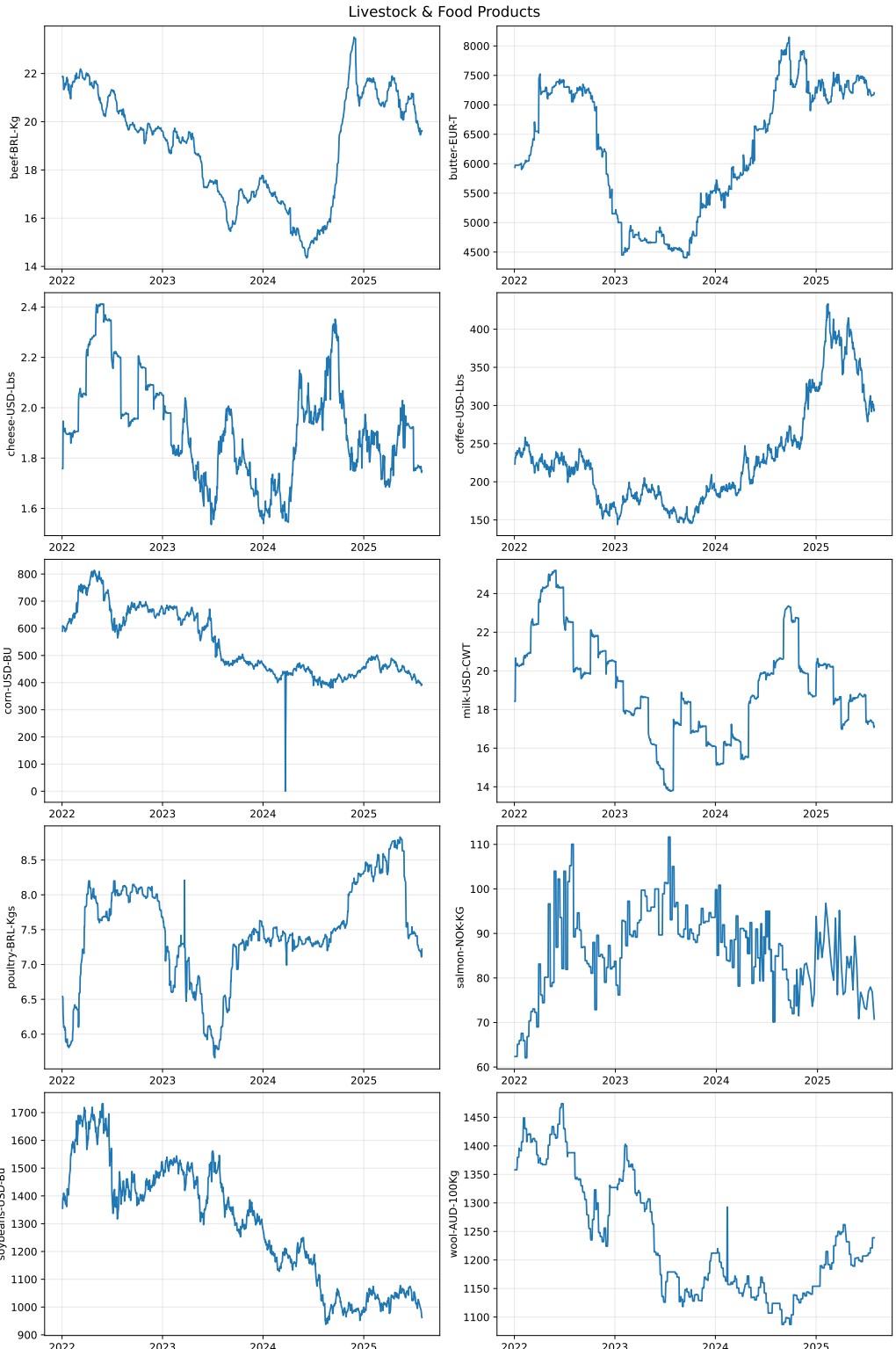

*Figure 28.* Numeric series visualization for the Livestock and Food Products domain.

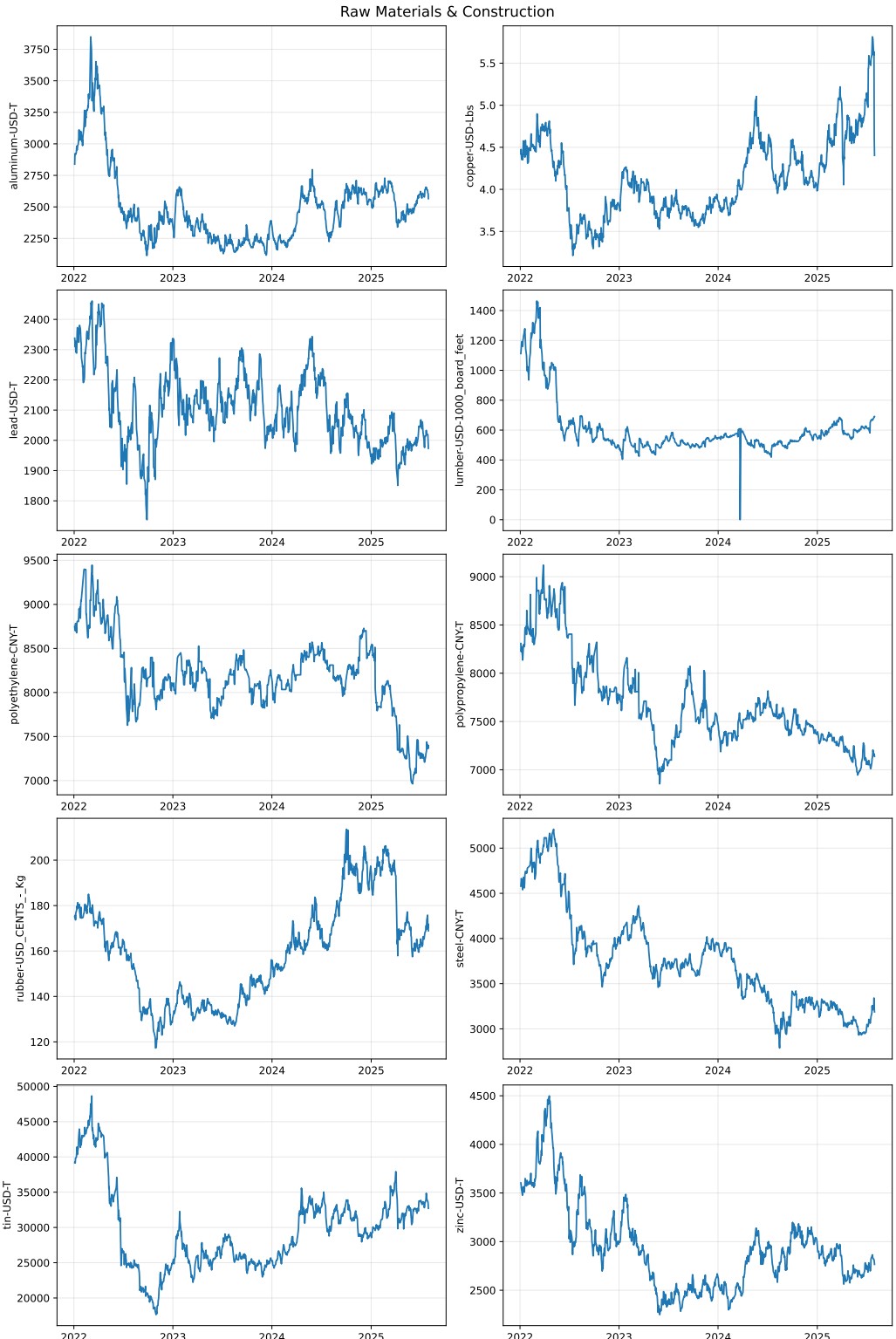

*Figure 29.* Numeric series visualization for the Raw Materials and Construction domain.

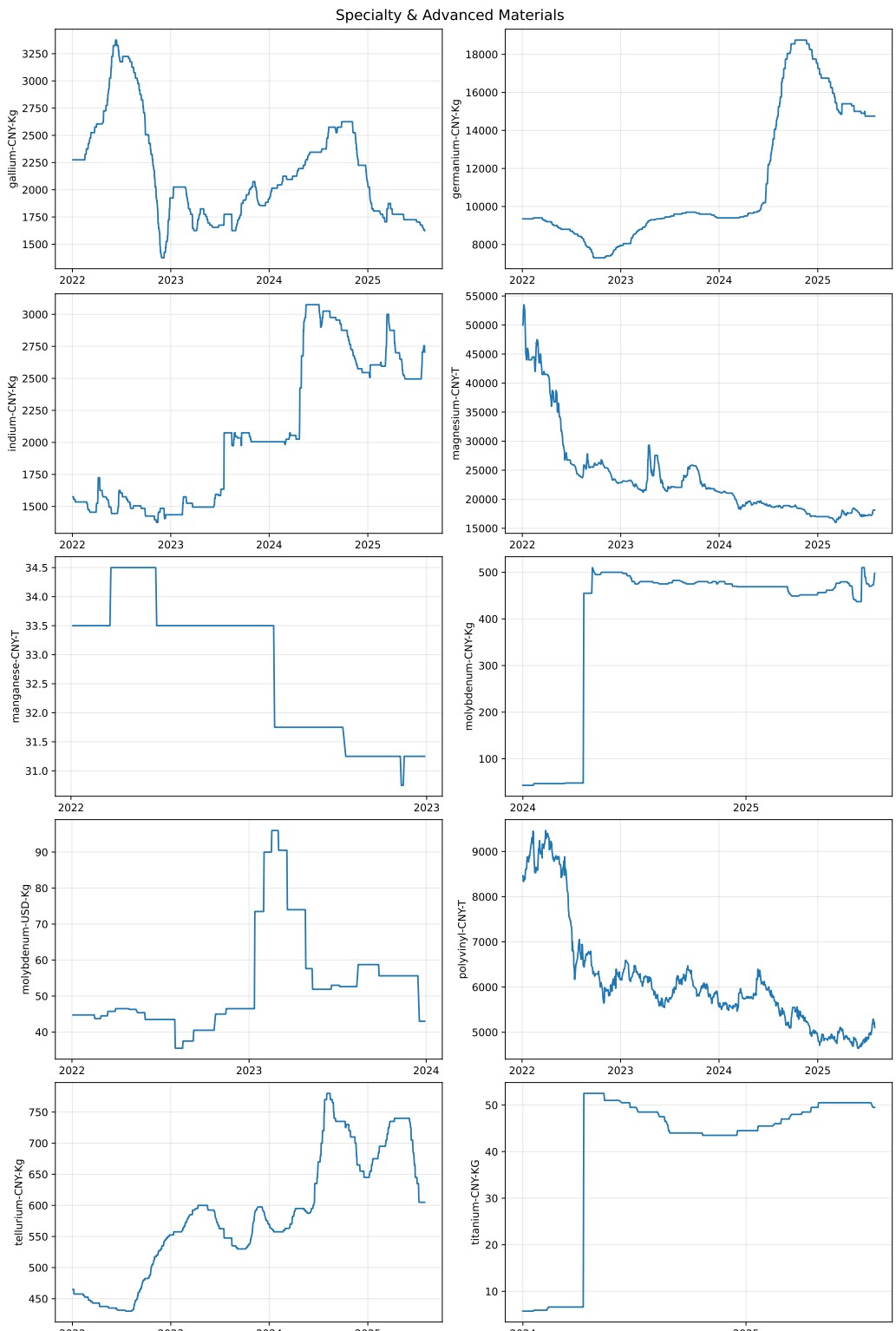

*Figure 30.* Numeric series visualization for the Specialty and Advanced Materials domain.

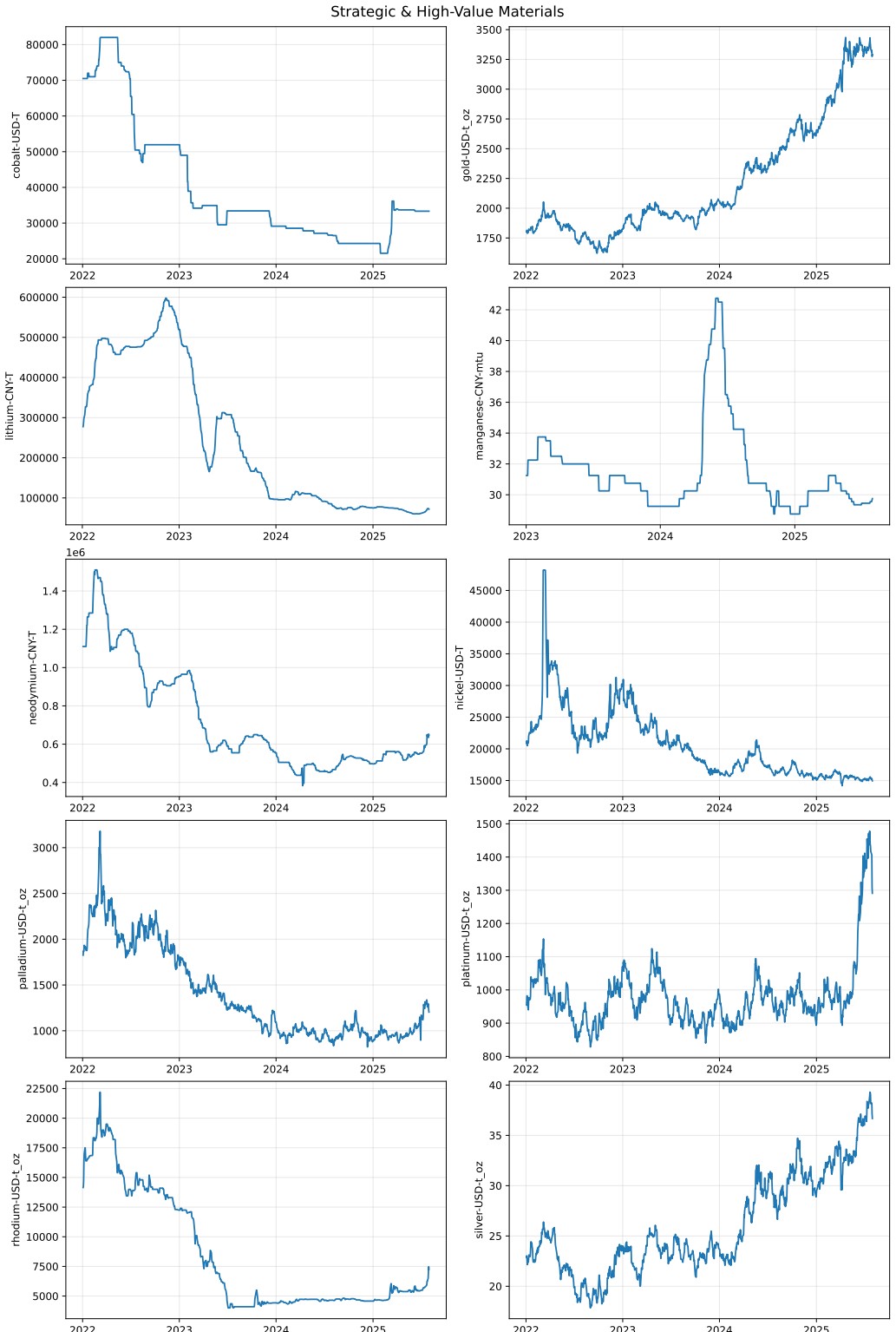

*Figure 31.* Numeric series visualization for the Strategic and High-Value Materials domain.

# N. Evaluation Metrics

**Principle** We use the Mean Absolute Scaled Error (MASE) as the core metric and normalize it by a seasonal naive baseline. This choice of MASE is following GIFT-EVAL (Aksu et al., 2024) and Chronos Benchmark 2 (Ansari et al., 2024). We aggregate performance across datasets using the *geometric mean* of normalized scores rather than the arithmetic mean, since prior work proved that the geometric mean is more robust to the choice of normalization baseline (Fleming & Wallace, 1986).

**Per-window MASE.** Let dataset $i \in \{1, \ldots, D\}$ be a single variable. Its full series is $\{y_t^{(i)}\}_{t=1}^{T_i}$. Window $w \in \{1, \ldots, W_i\}$ has forecast origin $\tau_{i,w}$ and horizon $H_{i,w}$, so the history is $\{y_t^{(i)}\}_{t=1}^{\tau_{i,w}}$. Given a seasonality $m$, the in-history seasonal scale is

$$Q_{i,w} = \frac{1}{\tau_{i,w} - m} \sum_{t=m+1}^{\tau_{i,w}} \left| y_t^{(i)} - y_{t-m}^{(i)} \right|. \quad (7)$$

For a model with forecasts $\{\widehat{y}_{\tau_{i,w}+h}^{(i)}\}_{h=1}^{H_{i,w}}$, the per-window MASE is

$$\text{MASE}_{i,w}(\text{model}) = \frac{1}{H_{i,w}} \sum_{h=1}^{H_{i,w}} \frac{\left| y_{\tau_{i,w}+h}^{(i)} - \widehat{y}_{\tau_{i,w}+h}^{(i)} \right|}{Q_{i,w}}. \quad (8)$$

This follows the GIFT-EVAL scaling but replaces a separate training split with the history up to the forecast origin.

**Seasonal naive baseline.** The seasonal naive forecast repeats the last observed seasonal cycle from the history. Let $\mathbf{s}^{(i,w)} = \left(y_{\tau_{i,w}-m+1}^{(i)}, \ldots, y_{\tau_{i,w}}^{(i)}\right)$. Then

$$\widehat{y}_{\tau_{i,w}+h}^{(i),\text{SNAIVE}} = \mathbf{s}_{1+\left((h-1) \bmod m\right)}^{(i,w)}, \quad h = 1, \ldots, H_{i,w}. \quad (9)$$

We compute $\text{MASE}_{i,w}(\text{SNAIVE})$ by substituting equation 9 into equation 8.

**Per-dataset normalization.** For each dataset $i$, we aggregate across its windows and form a normalized *MASE ratio*:

$$R_i(\text{model}) = \frac{\sum_{w=1}^{W_i} \text{MASE}_{i,w}(\text{model})}{\sum_{w=1}^{W_i} \text{MASE}_{i,w}(\text{SNAIVE})}. \quad (10)$$

Values $R_i < 1$ indicate improvement over the seasonal naive baseline on dataset $i$.

**Primary aggregate: geometric mean of ratios.** We report the geometric mean across all datasets as the primary summary:

$$\text{GM}(\text{model}) = \left(\prod_{i=1}^{D} R_i(\text{model})\right)^{\frac{1}{D}}. \quad (11)$$

In our release, $D = 200$.

**Secondary aggregate: average rank.** As a complementary, scale-free indicator, we rank models on each dataset by $R_i$ (lower is better). Let $\text{rank}_i(\text{model}) \in \{1, 2, \ldots\}$ be the rank of a model on dataset $i$. We report the average rank

$$\text{AvgRank}(\text{model}) = \frac{1}{D} \sum_{i=1}^{D} \text{rank}_i(\text{model}). \quad (12)$$

**More Details.** Unless otherwise specified, we set the seasonality to $m = 12$ for monthly data, $m = 4$ for weekly data, and $m = 7$ for daily data, which matches the construction of our series and the seasonal naive baseline used for normalization.

# O. Evaluation Samples Statistics

Under our rolling-window evaluation setup, we obtain a total of 2,434 forecasting samples on TimesX. These samples cover 19 domains and 190 variables. On average, each domain contributes about 128.1 samples, and each variable contributes about 12.8 samples.

# P. Knowledge Cutoff of Evaluated Models

Since pretrained models may have ingested training data only up to specific points in time, we explicitly document the knowledge cutoff date of each model considered in our experiments. This ensures a fair evaluation by avoiding potential data leakage from future information. The knowledge cutoffs are as follows:

- **GPT-5**: September 30, 2024
- **Gemini-2.5-Flash**: January 2025
- **Gemini-2.0-Flash**: June 2024
- **GPT-4o**: October 2023
- **DeepSeek-R1**: prior to June 2024
- **DeepSeek-V3**: prior to June 2024

For all evaluations, we align the forecasting horizons such that the prediction targets fall strictly after each model's knowledge cutoff date, thereby minimizing the risk of contamination.

## P.1. Test-time access control

We additionally control test-time access to external information. Plain LLM baselines are evaluated with no web search, no tools, and no function calling. For code/function/agent-style methods, generated code is executed in a no-network

sandbox. The allowed function list excludes API-access functions, and we manually checked execution logs without observing test-time leakage.

# Q. Implementation Details of Methods

**Implementation Details of Pretrained LLMs** We use the LLM decoding settings recommended by OpenRouter.[10]. We follow the prompt format used in CiK (Williams et al., 2024).

**Implementation Details of Agentic Forecasting Methods**

These three methods demonstrate different strategies for integrating TFM with LLMs, ranging from simple textual corrections to complex code generation, providing diverse approaches for context-aware time series forecasting.

## Q.1. Text Revision Method (TextRev)

### Q.1.1. METHOD OVERVIEW

The Text Revision method employs a two-stage approach: first generating initial numerical forecasts using TimesFM, then leveraging large language models to perform context-aware textual corrections on these predictions. This method transforms time series forecasting into a text manipulation task, enabling LLMs to understand and modify numerical predictions through natural language processing.

### Q.1.2. IMPLEMENTATION STEPS

1. **Foundation Forecast Generation**: TimesFM generates point forecasts based on historical time series data

2. **Text-based Revision**: The TimesFM predictions are converted to timestamp-value pairs and fed to the LLM along with contextual information

3. **Result Parsing**: The corrected forecast values are extracted from the LLM response

### Q.1.3. CORRECTION PROMPT TEMPLATE

See Figure 35.

## Q.2. Function Call Revision Method (FuncRev)

### Q.2.1. METHOD OVERVIEW

The Function Call Revision method extends the text revision approach by providing LLMs with a structured set of predefined functions for forecast adjustments. This method incorporates multi-round conversation mechanisms with text/visual/hybrid critic feedback modes, offering systematic and reproducible forecast modifications.

---

[10]https://openrouter.ai/

### Q.2.2. IMPLEMENTATION STEPS

1. **Initial Prediction**: TimesFM generates the baseline forecast

2. **Multi-round Revision Loop**:
   - Critic analyzes current forecast and provides feedback
   - Forecaster calls predefined functions to adjust predictions based on feedback
   - Process repeats until maximum rounds reached

3. **Final Output**: Returns the forecast from the last revision round

### Q.2.3. PREDEFINED FUNCTION SET

The system provides 13 forecast adjustment functions organized into five categories:

- **Basic Transformations**: `shift(offset)`, `scale(factor)`, `linear_transform(slope, intercept)`

- **Trend Adjustments**: `add_linear_trend(slope)`, `add_exponential_trend(base, growth_rate)`, `adjust_trend_strength(factor)`

- **Smoothing Operations**: `moving_average_smooth(window)`, `exponential_smooth(alpha)`

- **Seasonality Modifications**: `add_seasonal_pattern(period, amplitude, phase)`, `remove_seasonal_pattern(period)`

- **Data Normalization**: `standardize()`, `normalize_range(min_val, max_val)`, `clip_outliers(lower_percentile, upper_percentile)`

### Q.2.4. FUNCTION CALL PROMPT TEMPLATE

See Figure 36.

## Q.3. Code Revision Method (CodeRev)

### Q.3.1. METHOD OVERVIEW

The Code Revision method provides maximum adjustment flexibility by allowing LLMs to generate and execute free-form Python code for forecast modifications. This approach operates within a secure execution environment, supports multiple scientific computing libraries, and includes timeout and retry mechanisms for robust operation.

### Q.3.2. IMPLEMENTATION STEPS

1. **Initial Prediction**: TimesFM generates the baseline forecast

2. **Multi-round Revision Loop**:

   - Critic provides feedback on current forecast
   - LLM generates Python code for forecast adjustments
   - Code executes safely in restricted environment
   - Retry mechanism activates if execution fails (maximum 3 attempts)

3. **Code Execution Environment**: Pre-imported scientific libraries and forecast data variables are provided

### Q.3.3. SUPPORTED LIBRARIES

The execution environment includes the following pre-imported libraries: `numpy`, `pandas`, `math`, `datetime`, `requests`, `sqlite3`, `csv`, `json`, `sympy`, `statsmodels`, `networkx`.

### Q.3.4. CODE GENERATION PROMPT TEMPLATE

See Figure 37.

## Q.4. Additional ChatTS result

As an additional related baseline, we evaluated ChatTS Qwen3-8B on TimesX. Its aggregated MASE on TimesX is 0.8933.

## R. Additional results on CiK dataset

See results on all the five CiK subsets in Table 15.

## S. Additional results for context–quality study on Time-MMD dataset

**Setup.** We evaluate on all nine *Time-MMD* datasets. For LLM methods we fix the prompt template, the decoding settings, and the model **Gemini 2.0 Flash**; we *only* replace the textual context (ours vs. the original *Time-MMD* context). The numeric series remain unchanged. We use the period from **2021-06-30** to **2024-04-01**, which is the cutoff date of *Time-MMD* dataset. For **daily** datasets we set hist_window = 365, pred_window = 120, slide_window = 40. For **weekly** datasets we set hist_window = 96, pred_window = 12, slide_window = 4. For **monthly** datasets we set hist_window = 16, pred_window = 4, slide_window = 2. These choices balance sample count and sample diversity while keeping the same evaluation protocol across methods.

**Complete per-domain results.** Table 18 reports normalized performance (MASE ↓) for each domain and the geometric mean across domains, together with the average rank. We use two decimals for all numbers and do not report the arithmetic mean.

## T. Model Rankings Robustness Analysis to Benchmark Size

We study how the size of the evaluation benchmark affects the stability of model performance. Starting from the 190 in-distribution variables in TimesX, we construct smaller benchmark variants by randomly sampling subsets of variables without replacement.

For a target subset size $K \in \{10, 20, \ldots, 190\}$, we draw 20 subsets of $K$ distinct variables. For each subset and each forecasting model, we compute the geometric-mean normalized MASE over the variables in that subset. This gives, for every $(K, \text{model})$ pair, a distribution of MASE values across random subsets.

Figure 34 summarizes these results. The horizontal axis is the number of variables $K$, and for each model we plot the mean MASE over the 20 subsets (solid line) together with the 5th–95th percentile band (shaded area). When $K$ is between 10 and 40—which is similar to the sizes of most existing multimodal TSF benchmarks listed in Table 1—the percentile bands are very wide and strongly overlap across models. This means that, in this small-scale regime, the apparent ranking of models can change substantially depending on which variables are included in the benchmark. As $K$ increases, the bands shrink and the relative ordering becomes more stable.

These observations support our motivation of designing a large-scale benchmark for more stable and reliable comparisons.

## U. More Evaluation Results on TimesX

### U.1. CRPS-based probabilistic evaluation

In the main text we focus on the (normalized) MASE because most LLM-based methods in our study output point forecasts rather than full predictive distributions. Here we complement this view with a probabilistic evaluation based on the continuous ranked probability score (CRPS).

For a predictive distribution $F$ over a scalar outcome $y$, the CRPS is defined as

$$\text{CRPS}(F, y) = \int_{-\infty}^{+\infty} \left( F(z) - \mathbf{1}\{z \geq y\} \right)^2 \, \mathrm{d}z. \quad (13)$$

When we only have samples $x_1, \ldots, x_M$ from the predictive distribution, we estimate the CRPS following CiK (Williams

*Table 15.* Results on all the five CiK subsets (MASE ↓). The geometric-mean column shows that CODEREV performs best, followed by Gemini-2.0-Flash, and then TimesFM-2.5. This ordering is the opposite of what we observe on real-world data (TimesX), illustrating that synthetic generation can flip model rankings. While, carefully designed synthetic benchmarks like CiK are very useful for testing specific capabilities such as instruction following and different types of reasoning over controlled contexts. We highly recommend using both synthetic and real-world benchmarks for a more complete and robust evaluation

| Method | CiK_Causal | CiK_Covariates | CiK_History | CiK_Intemporal | CiK_Future | Geom. mean MASE |
|---|---|---|---|---|---|---|
| CODEREV | 0.76 | 0.58 | 0.67 | 0.66 | 0.24 | **0.38** (#1) |
| Gemini-2.0-Flash | 0.77 | 0.57 | 0.76 | 0.61 | 0.32 | **0.43** (#2) |
| TimesFM-2.5 | 0.73 | 0.73 | 0.69 | 0.71 | 0.70 | **0.58** (#3) |

et al., 2024). Let $x_1 \leq \cdots \leq x_M$ denote the samples sorted in ascending order. An efficient estimator is

$$
\widehat{\mathrm{CRPS}}(\tilde{X}, y) \approx \frac{1}{M} \sum_{n=1}^{M} |x_n - y| + \frac{1}{M} \sum_{n=1}^{M} x_n
$$
$$
- \frac{2}{M(M-1)} \sum_{n=1}^{M} (n-1)\, x_n,
$$
(14)

where $\tilde{X} = \{x_1, \ldots, x_M\}$ and the $x_n$ are sorted. This estimator has $\mathcal{O}(M \log M)$ time complexity due to the sorting step and is numerically equivalent to the standard unbiased estimator based on pairwise distances.

In our experiments we treat each stochastic run of a method as one sample from its predictive distribution. For every method and every evaluation instance on TimesX, we produce $M = 10$ stochastic forecasts, compute the CRPS using Equation equation 14, and then aggregate CRPS across variables using geometric mean . Table 16 reports the resulting aggregated CRPS values (lower is better).

*Table 16.* Geometric-mean CRPS on TimesX when estimating predictive uncertainty with 10 stochastic samples per instance. Lower is better.

| Method | Geometric-mean CRPS |
|---|---|
| GPT-4o | 0.258 |
| Gemini-2.0-Flash | 0.276 |
| DeepSeek-V3 | 0.277 |
| Sundial | 0.278 |
| TimesFM-2.5 | 0.293 |
| Moirai-2.0 | 0.327 |

All three LLMs achieve lower geometric-mean CRPS than the three TFMs. This is consistent with our MASE results and suggests that LLMs can use the rich textual context in TimesX to assign probability mass to multiple plausible futures, while TFMs rely only on numeric series. We view this CRPS study as an initial step toward more systematic probabilistic evaluation on TimesX; future work may explore improved uncertainty estimation and training objectives that directly optimize probabilistic scores such as CRPS.

## V. Effects of various variable characteristics

In this section we investigate what variable level features can impact the ordering of LLM based multimodal solutions vs TFMs.

In Fig. 32, we plot the win rate of Gemini-2.0-Flash against TimesFM-2.5 as we climb the quartiles of event counts, length of event details and seasonality. We can see that with increasing event information the multimodal solutions that leverage these become better than TFMs which are time-series only. In the case of seasonality, extremely seasonal series are easy to predict and therefore that edge that TFMs have over LLMs in pure forecasting tasks reduces.

In Fig. 33, we plot the same while varying various time-series characteristics like trend, non-stationarity and transition/ change-points. Increasing quartiles of these indicate the hardness of the pure time-series forecasting task irrespective of the text context, and therefore TFMs can perform better on these time-series tasks. Consequently, very strong trends and high non-stationarity reduce Gemini's edge over TFMs.

**Domain-level effects of calendar and covariate contexts.** Holidays are effective in 12/19 domains, with an average intra-domain win rate of 65.2%. The largest gains appear in Shopping, Climate & Environment, Public Health, and Public Policy & Governance. Covariates are effective in 12/19 domains, with an average intra-domain win rate of 57.9%. The largest gains appear in Specialty & Advanced Materials, Raw Materials & Construction, Livestock & Food Products, and Currency.

## W. Detailed Experiment Results for Advanced Reasoning Models

We further extend our experiments to advanced reasoning LLMs. Specifically, for GPT-4o, Gemini-2.0-Flash, and DeepSeek-V3, we introduce their corresponding reasoning versions, GPT-5, Gemini-2.5-Flash, and DeepSeek-R1. To avoid data contamination, we select evaluation examples whose forecast horizons begin after January 2025. As shown in Table 17, we observe that while GPT-5 outperforms GPT-

| Capability | Multimodal TSF | Factuality | MM LongContext | Mathematics | Science |
| Benchmark | *Our TimesX (↓)* | *SimpleQA (↑)* | *MRCR (↑)* | *AIME2025 (↑)* | *GPQA2025 (↑)* |
|---|---|---|---|---|---|
| GPT-5 | 0.61 | 51.1 | 96.0 | 91.7 | 85.4 |
| GPT-4o | 0.65 | 38.4 | 55.8 | 6.00 | 51.1 |
| Gemini-2.0-Flash | 0.66 | 28.2 | 69.2 | 21.7 | 62.3 |
| DeepSeek-V3 | 0.72 | 29.5 | 33.8 | 26.0 | 55.7 |
| Gemini-2.5-Flash | 0.75 | 27.8 | 32.0 | 73.7 | 68.3 |
| DeepSeek-R1 | 0.78 | 31.9 | 18.0 | 76.0 | 81.3 |

*Table 17.* Relationships between LLMs' multimodal TSF performance and four core capabilities. We use data with sample start dates after January 2025 on TimesX. On TimesX, lower MASE indicates better performance, while on the other four benchmarks, higher scores indicate better performance.

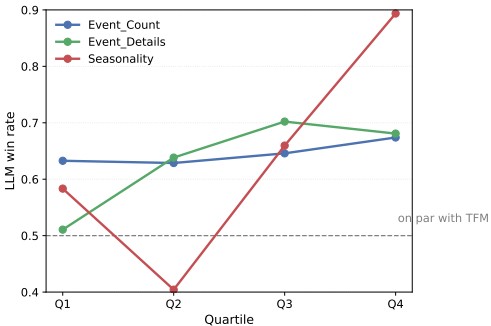

*Figure 32.* We plot the aggregated MASE (lower is better) as a function of variable level features like event count, length of event details and seasonality.

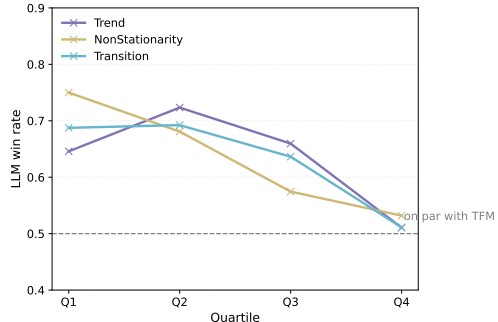

*Figure 33.* We plot the aggregated MASE (lower is better) as a function of variable level time-series features like trend, non-stationarity and transitions.

4o, the reasoning models in the other two pairs perform significantly worse than their non-reasoning counterparts. To further understand the drivers of TSF performance, we cross reference the TSF performance with other four core LLM capabilities: factuality (SimpleQA (Wei et al., 2024)), multimodal long-context understanding (MRCR), mathematics (AIME (Guha et al., 2025)), and science (GPQA (Rein et al., 2024)). The results suggest that multimodal TSF is unrelated to the mathematics and science skills emphasized by current reasoning models. Instead, it depends more on factuality and multimodal long-context understanding, which are better captured by non-reasoning LLMs.

## X. Details of in-context learning experiments

**Experimental setup.** We denote in-context learning as ICL, and ICL-C as its conservative variant that allows a no-change option. Due to the computational cost and to avoid leakage from in-context demonstrations, we evaluate all methods on only the last three evaluation instances per variable, and use the remaining instances as the candidate pool for demonstration retrieval. All other experimental settings remain unchanged.

**Sample selection (one-shot, leakage-free nearest neighbor).** For each target evaluation instance, we select a single

historical instance as the in-context demonstration. To guarantee no leakage, we require the demonstration instance to satisfy *horizon-end < target-horizon-start*. Among all candidates that satisfy this constraint, we choose the temporally closest one (i.e., with the smallest time gap), as it is expected to be more similar to the target instance.

**ICL-C: conservative no-change option.** In ICL-C, the prompt allows TEXTREV to take a conservative strategy when it is uncertain about the revision, i.e., keep the unimodal forecast unchanged.

**Prompt templates.** Figure 38 shows the prompt template for TEXTREV-ICL. In the demonstration, the `<demo_forecast>` block uses the ground-truth future values of the historical (demonstration) instance. For TextRev-ICL-C, we further add the following sentence to explicitly allow a conservative *no-change* option: *If you are not confident about the revision based on the context, you are allowed the initial forecast unchanged.*

## Y. Detailed Per-Series MASE Results

### Y.1. Detailed Main Benchmarking Results

The detailed per-series MASE results corresponding to Table 4 are reported in Tables 19–56.

### Y.2. Detailed MASE Results of Event Type Attribution Analysis with all LLMs

The detailed per-series MASE results of the event type attribution analysis corresponding to Figure 4 are reported in Tables 57–94.

### Y.3. Detailed MASE Results of Event Type Deep Analysis Using Gemini

The detailed per-series MASE results of the event type deep analysis with Gemini correspondin to Table 5 are reported in Tables 95–132.

### Y.4. Detailed MASE Results of Reasoning Models with Data after January 2025

The detailed per-series MASE results of reasoning models evaluated on data after January 2025 corresponding to Table 17 are reported in Tables 133–170.

## Z. Detailed Benchmarking Results by Domain

The detailed per-domain MASE results corresponding to Table 4 are reported in Table 171.

*Table 18.* Per-dataset results on *Time-MMD* (MASE ↓). Rows are datasets and summary metrics; columns are methods. LLM settings (model/prompt/temperature) are fixed; only textual context differs.

| Dataset / Metric | LLM with our context | LLM with Time-MMD context | Moirai2 | TimesFM2.5 | SeasonalNaive |
|---|---|---|---|---|---|
| traffic | 0.76 | **0.70** | 3.74 | 2.95 | 1.00 |
| economy | **0.76** | 0.77 | 1.16 | 1.42 | 1.00 |
| health | 0.92 | 0.89 | 0.99 | **0.81** | 1.00 |
| social | 0.80 | 1.02 | **0.75** | 1.40 | 1.00 |
| environment | 0.82 | 0.82 | 0.75 | **0.71** | 1.00 |
| agriculture | 0.81 | 0.97 | 0.78 | **0.57** | 1.00 |
| energy | **0.68** | 0.95 | 0.86 | 0.80 | 1.00 |
| security | 0.96 | 0.91 | **0.81** | 1.10 | 1.00 |
| climate | 1.15 | 1.21 | **0.77** | 0.78 | 1.00 |
| **Geometric Mean** | **0.84** | 0.91 | 1.02 | 1.04 | 1.00 |
| **Average Rank** | **2.50** | 3.06 | 2.56 | 2.89 | 4.00 |

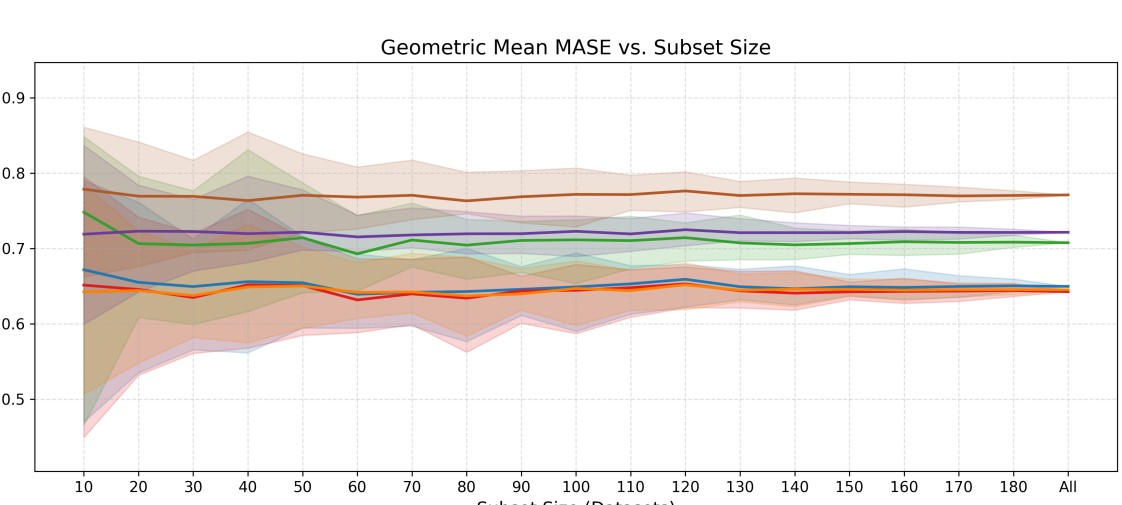

*Figure 34.* Effect of benchmark size on the geometric-mean normalized MASE. For each subset size $K$, we sample 20 subsets of $K$ variables from TimesX, compute the metric for each model on each subset, and plot the mean (solid line) and the 5th–95th percentile band (shaded area). When $K$ is at existing benchmark scale (10–40), the bands are wide and strongly overlapping, indicating unstable rankings; at larger $K$ the bands narrow and the ordering becomes more stable.

```
I have a time series forecasting correction task for you.

Here is some context about the task. Please consider this information when reviewing
    the forecast:
<context>
{background information, constraints, scenario descriptions, holiday information,
    etc.}
</context>

Here is the historical time series in (timestamp, value) format:
<history>
{historical data points}
</history>

An initial forecast has been generated using a deep model. Here it is:
<initial_forecast>
{TimesFM prediction results}
</initial_forecast>

Please review the initial forecast and adjust the values considering the provided
    context. Make reasonable modifications where the context provides relevant
    information that could improve the forecast.

Return your corrected forecast in (timestamp, value) format between <forecast> and
    </forecast> tags.
Do not include any other information (e.g., comments) in the forecast.

Example format:
<forecast>
(2024-01-01 12:00:00, 123.45)
(2024-01-01 13:00:00, 124.67)
</forecast>
```

*Figure 35.* Prompt template for Text Revision method (TextRev).

```
You are an expert time series forecaster with access to forecast adjustment tools.

Task context:
<context>
{contextual information}
</context>

Critic feedback:
<feedback>
{critic analysis and suggestions}
</feedback>

Current forecast:
<current_forecast>
{current prediction data}
</current_forecast>

Available adjustment functions: {function list}

Please analyze the critic feedback and select appropriate functions to adjust the
    forecast. You can:
1. Call a single function for specific adjustments
2. Call multiple functions for combined adjustments
3. Choose not to call any functions if the current forecast is already reasonable

Please explain your adjustment strategy and call the corresponding functions.
```

*Figure 36.* Prompt template for Function Call Revision method (FuncRev).

```
You are an expert time series forecaster with Python programming capabilities.
    Instead of using predefined functions, you should write Python code to adjust
    the forecast based on the critic's feedback.

Here is the context about the task:
<context>
{contextual information}
</context>

The critic's feedback:
<critic_feedback>
{critic feedback}
</critic_feedback>

PROGRAMMING ENVIRONMENT:
You have access to a restricted Python environment with pre-imported libraries and
    variables.

Pre-imported libraries: numpy, pandas, math, datetime, requests, sqlite3, csv, json,
     sympy, statsmodels, networkx

Available variables in your code:
- current_forecast: Dictionary mapping timestamps to forecast values
- timestamps: List of prediction timestamps (strings in "YYYY-MM-DD HH:MM:SS" format
    )
- forecast_values: List of forecast values corresponding to timestamps

INSTRUCTIONS:
1. Write Python code to adjust the forecast based on the critic's feedback
2. DO NOT include any import statements - all libraries are already imported
3. Your code can perform any mathematical operations, transformations, or
    adjustments
4. You must assign the final adjusted forecast to a variable called '
    adjusted_forecast'
5. The 'adjusted_forecast' should be a dictionary mapping timestamps to adjusted
    values
6. You can modify forecast_values list and then reconstruct the dictionary, or work
    directly with current_forecast
7. Be creative with your adjustments - you're not limited to predefined functions

EXAMPLE CODE STRUCTURE:
```python
# Your analysis and adjustment logic here
# DO NOT include import statements - libraries are pre-imported

# Example: Apply some adjustment based on critic feedback
for i, timestamp in enumerate(timestamps):
    # Your logic here
    forecast_values[i] = forecast_values[i] * some_factor # example adjustment

# Final result
adjusted_forecast = {timestamp: value for timestamp, value in zip(timestamps,
    forecast_values)}
```

CRITICAL REQUIREMENTS:
- DO NOT include any import statements (libraries are pre-imported)
- Is syntactically correct Python
- Uses only the pre-imported libraries
- Assigns the final result to 'adjusted_forecast' variable
- Handles the forecast data appropriately

Your Python code (without any import statements):
```

*Figure 37.* Prompt template for Code Revision method (CodeRev).

```
I have a time series forecasting correction task for you.

Here is some context about the task. Please consider this information when reviewing
    the forecast:
<context>
{background information, constraints, scenario descriptions, holiday information,
    etc.}
</context>

Here is the historical time series in (timestamp, value) format:
<history>
{historical data points}
</history>

An initial forecast has been generated using a deep model. Here it is:
<initial_forecast>
{TimesFM prediction results}
</initial_forecast>

# One-shot demonstration (historical instance)
# NOTE: demo_forecast uses the ground-truth future values for the demonstration
    instance.
<demo>
  <demo_context>
  {context for the demonstration instance}
  </demo_context>

  <demo_history>
  {historical data points for the demonstration instance}
  </demo_history>

  <demo_initial_forecast>
  {initial forecast for the demonstration instance}
  </demo_initial_forecast>

  <demo_forecast>
  {ground-truth future values for the demonstration instance}
  </demo_forecast>
</demo>

Please review the initial forecast and adjust the values considering the provided
    context. Make reasonable modifications where the context provides relevant
    information that could improve the forecast.

Return your corrected forecast in (timestamp, value) format between <forecast> and
    </forecast> tags.
Do not include any other information (e.g., comments) in the forecast.

Example format:
<forecast>
(2024-01-01 12:00:00, 123.45)
(2024-01-01 13:00:00, 124.67)
</forecast>
```

*Figure 38.* Prompt template for TEXTREV-ICL (one-shot demonstration).

| Method | affordable_housing | air_conditioner | air_pollution | air_travel | alphabet |
|---|---|---|---|---|---|
| SeasonalNaive | 1.000 | 1.000 | 1.000 | 1.000 | 1.000 |
| Sundial | 0.708 | 0.230 | 0.656 | 0.965 | 0.581 |
| Moirai2.0 | 0.705 | 0.391 | 0.642 | 0.969 | 0.645 |
| TimesFM2.5 | 0.662 | 0.227 | 0.436 | 0.858 | 0.476 |
| AvgEnsemble(TimesFM,Moirai) | 0.678 | 0.249 | 0.466 | 0.907 | 0.530 |
| DeepSeek-V3 | 0.701 | 0.495 | 0.523 | 0.950 | 0.586 |
| Gemini-2.0-Flash | 0.688 | 0.274 | 0.356 | 0.917 | 0.513 |
| GPT-4o | 0.659 | 0.339 | 0.367 | 0.829 | 0.567 |
| FuncRev(TimesFM,Gemini) | 0.730 | 0.398 | 0.416 | 0.895 | 0.483 |
| CodeRev(TimesFM,Gemini) | 0.646 | 0.243 | 0.521 | 0.947 | 0.479 |
| TextRev(TimesFM,Gemini) | 0.666 | 0.230 | 0.421 | 0.842 | 0.465 |
| AvgEnsemble(TimesFM,GPT) | 0.644 | 0.238 | 0.343 | 0.851 | 0.463 |
| AvgEnsemble(TimesFM,Gemini) | 0.641 | 0.243 | 0.340 | 0.844 | 0.495 |

*Table 19.* Detailed MASE results of Table 1. (part 1/38)

| Method | aluminum_usd_t | amazon | animal_migration | animal_rescue | animal_welfare |
|---|---|---|---|---|---|
| SeasonalNaive | 1.000 | 1.000 | 1.000 | 1.000 | 1.000 |
| Sundial | 1.014 | 1.051 | 0.634 | 0.830 | 0.709 |
| Moirai2.0 | 0.785 | 0.875 | 0.628 | 0.820 | 0.739 |
| TimesFM2.5 | 0.873 | 0.635 | 0.617 | 0.713 | 0.516 |
| AvgEnsemble(TimesFM,Moirai) | 0.810 | 0.747 | 0.599 | 0.752 | 0.599 |
| DeepSeek-V3 | 0.782 | 1.241 | 0.794 | 1.166 | 0.695 |
| Gemini-2.0-Flash | 0.795 | 0.570 | 0.696 | 0.956 | 0.647 |
| GPT-4o | 0.784 | 0.996 | 0.664 | 1.048 | 0.618 |
| FuncRev(TimesFM,Gemini) | 1.259 | 0.828 | 0.687 | 0.873 | 0.583 |
| CodeRev(TimesFM,Gemini) | 1.821 | 0.661 | 0.589 | 0.786 | 0.566 |
| TextRev(TimesFM,Gemini) | 0.528 | 0.683 | 0.612 | 0.814 | 0.537 |
| AvgEnsemble(TimesFM,GPT) | 0.805 | 0.544 | 0.644 | 0.797 | 0.549 |
| AvgEnsemble(TimesFM,Gemini) | 0.795 | 0.492 | 0.649 | 0.778 | 0.560 |

*Table 20.* Detailed MASE results of Table 1. (cont'd, part 2/38)

| Method | apple_inc | art_exhibitions | artificial_intelligence | asset_management | autonomous_driving |
|---|---|---|---|---|---|
| SeasonalNaive | 1.000 | 1.000 | 1.000 | 1.000 | 1.000 |
| Sundial | 0.791 | 0.727 | 1.050 | 1.026 | 1.099 |
| Moirai2.0 | 0.772 | 0.806 | 0.921 | 0.827 | 0.949 |
| TimesFM2.5 | 0.732 | 0.816 | 0.902 | 0.830 | 0.888 |
| AvgEnsemble(TimesFM,Moirai) | 0.749 | 0.801 | 0.908 | 0.825 | 0.915 |
| DeepSeek-V3 | 0.714 | 0.752 | 0.825 | 0.877 | 0.980 |
| Gemini-2.0-Flash | 0.690 | 0.767 | 0.967 | 0.887 | 1.056 |
| GPT-4o | 0.656 | 0.674 | 0.761 | 0.759 | 0.855 |
| FuncRev(TimesFM,Gemini) | 0.666 | 0.738 | 1.020 | 0.866 | 0.781 |
| CodeRev(TimesFM,Gemini) | 0.731 | 0.846 | 0.884 | 0.815 | 0.841 |
| TextRev(TimesFM,Gemini) | 0.735 | 0.816 | 0.894 | 0.844 | 0.891 |
| AvgEnsemble(TimesFM,GPT) | 0.679 | 0.822 | 0.891 | 0.830 | 0.935 |
| AvgEnsemble(TimesFM,Gemini) | 0.675 | 0.725 | 0.861 | 0.816 | 0.918 |

*Table 21.* Detailed MASE results of Table 1. (cont'd, part 3/38)

| Method | back_to_school | barley_inr_t | beef_brl_kg | beekeeping | biodiversity |
|---|---|---|---|---|---|
| SeasonalNaive | 1.000 | 1.000 | 1.000 | 1.000 | 1.000 |
| Sundial | 0.544 | 0.939 | 0.522 | 0.477 | 0.687 |
| Moirai2.0 | 0.522 | 0.925 | 0.499 | 0.656 | 0.610 |
| TimesFM2.5 | 0.339 | 0.936 | 0.437 | 0.526 | 0.291 |
| AvgEnsemble(TimesFM,Moirai) | 0.423 | 0.917 | 0.450 | 0.584 | 0.414 |
| DeepSeek-V3 | 0.292 | 0.954 | 0.567 | 0.585 | 0.367 |
| Gemini-2.0-Flash | 0.306 | 0.950 | 0.574 | 0.596 | 0.314 |
| GPT-4o | 0.257 | 0.956 | 0.559 | 0.517 | 0.281 |
| FuncRev(TimesFM,Gemini) | 0.431 | 1.653 | 0.696 | 1.019 | 0.329 |
| CodeRev(TimesFM,Gemini) | 0.341 | 1.180 | 0.617 | 0.658 | 0.327 |
| TextRev(TimesFM,Gemini) | 0.248 | 2.005 | 0.406 | 0.596 | 0.296 |
| AvgEnsemble(TimesFM,GPT) | 0.314 | 0.938 | 0.463 | 0.519 | 0.274 |
| AvgEnsemble(TimesFM,Gemini) | 0.311 | 0.950 | 0.574 | 0.533 | 0.271 |

*Table 22.* Detailed MASE results of Table 1. (cont'd, part 4/38)

| Method | bitumen_cny_t | black_friday_deals | brent_usd_bbl | broadway_shows | butter_eur_t |
|---|---|---|---|---|---|
| SeasonalNaive | 1.000 | 1.000 | 1.000 | 1.000 | 1.000 |
| Sundial | 0.852 | 0.622 | 0.825 | 0.593 | 0.701 |
| Moirai2.0 | 0.748 | 0.584 | 0.851 | 0.653 | 0.694 |
| TimesFM2.5 | 0.744 | 0.552 | 0.767 | 0.561 | 0.777 |
| AvgEnsemble(TimesFM,Moirai) | 0.733 | 0.557 | 0.802 | 0.599 | 0.726 |
| DeepSeek-V3 | 0.727 | 0.129 | 0.820 | 0.732 | 0.762 |
| Gemini-2.0-Flash | 0.733 | 0.571 | 0.818 | 0.666 | 0.773 |
| GPT-4o | 0.728 | 0.457 | 0.830 | 0.649 | 0.763 |
| FuncRev(TimesFM,Gemini) | 1.209 | 1.222 | 0.789 | 0.750 | 0.805 |
| CodeRev(TimesFM,Gemini) | 0.968 | 1.594 | 0.771 | 0.554 | 0.777 |
| TextRev(TimesFM,Gemini) | 0.938 | 0.374 | 0.641 | 0.588 | 0.916 |
| AvgEnsemble(TimesFM,GPT) | 0.729 | 0.433 | 0.783 | 0.561 | 0.753 |
| AvgEnsemble(TimesFM,Gemini) | 0.733 | 0.435 | 0.818 | 0.577 | 0.773 |

*Table 23.* Detailed MASE results of Table 1. (cont'd, part 5/38)

| Method | cancer_research | canola_cad_t | carbon_emissions | cheese_usd_lbs | christmas_gifts |
|---|---|---|---|---|---|
| SeasonalNaive | 1.000 | 1.000 | 1.000 | 1.000 | 1.000 |
| Sundial | 0.846 | 0.994 | 0.693 | 0.873 | 0.535 |
| Moirai2.0 | 0.845 | 0.845 | 0.671 | 0.850 | 0.499 |
| TimesFM2.5 | 0.779 | 0.883 | 0.513 | 0.855 | 0.306 |
| AvgEnsemble(TimesFM,Moirai) | 0.802 | 0.859 | 0.568 | 0.845 | 0.383 |
| DeepSeek-V3 | 0.856 | 0.827 | 0.633 | 0.810 | 0.120 |
| Gemini-2.0-Flash | 0.829 | 0.846 | 0.718 | 0.787 | 0.098 |
| GPT-4o | 0.718 | 0.817 | 0.665 | 0.789 | 0.119 |
| FuncRev(TimesFM,Gemini) | 0.935 | 1.048 | 0.583 | 0.877 | 0.445 |
| CodeRev(TimesFM,Gemini) | 0.736 | 0.880 | 0.501 | 0.908 | 0.426 |
| TextRev(TimesFM,Gemini) | 0.766 | 1.055 | 0.530 | 0.792 | 0.285 |
| AvgEnsemble(TimesFM,GPT) | 0.802 | 0.858 | 0.565 | 0.816 | 0.187 |
| AvgEnsemble(TimesFM,Gemini) | 0.792 | 0.846 | 0.594 | 0.787 | 0.188 |

*Table 24.* Detailed MASE results of Table 1. (cont'd, part 6/38)

| Method | climate_change | coal_usd_t | cocoa_usd_t | coffee_usd_lbs | comic_con |
|---|---|---|---|---|---|
| SeasonalNaive | 1.000 | 1.000 | 1.000 | 1.000 | 1.000 |
| Sundial | 0.795 | 0.900 | 0.984 | 0.957 | 0.551 |
| Moirai2.0 | 0.858 | 0.638 | 0.822 | 0.775 | 0.663 |
| TimesFM2.5 | 0.878 | 0.625 | 0.864 | 0.775 | 0.498 |
| AvgEnsemble(TimesFM,Moirai) | 0.822 | 0.605 | 0.834 | 0.751 | 0.561 |
| DeepSeek-V3 | 1.475 | 0.653 | 0.812 | 0.750 | 0.445 |
| Gemini-2.0-Flash | 1.485 | 0.661 | 0.827 | 0.757 | 0.450 |
| GPT-4o | 1.359 | 0.655 | 0.768 | 0.757 | 0.507 |
| FuncRev(TimesFM,Gemini) | 0.797 | 0.674 | 0.849 | 0.870 | 0.426 |
| CodeRev(TimesFM,Gemini) | 0.952 | 0.617 | 0.750 | 0.819 | 0.439 |
| TextRev(TimesFM,Gemini) | 0.871 | 0.436 | 0.768 | 1.067 | 0.467 |
| AvgEnsemble(TimesFM,GPT) | 1.163 | 0.634 | 0.838 | 0.743 | 0.498 |
| AvgEnsemble(TimesFM,Gemini) | 0.762 | 0.661 | 0.827 | 0.757 | 0.479 |

*Table 25.* Detailed MASE results of Table 1. (cont'd, part 7/38)

| Method | consumer_electronics | copper_usd_lbs | corn_usd_bu | cost_of_living | cotton_usd_lbs |
|---|---|---|---|---|---|
| SeasonalNaive | 1.000 | 1.000 | 1.000 | 1.000 | 1.000 |
| Sundial | 0.727 | 0.757 | 0.985 | 0.676 | 0.891 |
| Moirai2.0 | 0.786 | 0.640 | 0.832 | 0.723 | 0.955 |
| TimesFM2.5 | 0.638 | 0.711 | 0.850 | 0.725 | 0.834 |
| AvgEnsemble(TimesFM,Moirai) | 0.676 | 0.655 | 0.826 | 0.722 | 0.872 |
| DeepSeek-V3 | 0.795 | 0.620 | 0.857 | 0.763 | 1.147 |
| Gemini-2.0-Flash | 0.621 | 0.635 | 0.855 | 0.799 | 1.148 |
| GPT-4o | 0.716 | 0.625 | 0.850 | 0.732 | 1.141 |
| FuncRev(TimesFM,Gemini) | 0.661 | 1.191 | 0.954 | 0.733 | 1.029 |
| CodeRev(TimesFM,Gemini) | 0.648 | 1.004 | 0.871 | 0.734 | 1.164 |
| TextRev(TimesFM,Gemini) | 0.644 | 1.177 | 1.043 | 0.720 | 0.965 |
| AvgEnsemble(TimesFM,GPT) | 0.600 | 0.652 | 0.849 | 0.746 | 0.963 |
| AvgEnsemble(TimesFM,Gemini) | 0.610 | 0.635 | 0.855 | 0.753 | 1.148 |

*Table 26.* Detailed MASE results of Table 1. (cont'd, part 8/38)

| Method | cryptocurrency | cybersecurity | data_breach | data_privacy | deforestation |
|---|---|---|---|---|---|
| SeasonalNaive | 1.000 | 1.000 | 1.000 | 1.000 | 1.000 |
| Sundial | 0.811 | 1.102 | 0.762 | 0.948 | 0.739 |
| Moirai2.0 | 0.814 | 0.916 | 0.727 | 0.774 | 0.621 |
| TimesFM2.5 | 0.772 | 0.879 | 0.706 | 0.709 | 0.376 |
| AvgEnsemble(TimesFM,Moirai) | 0.786 | 0.887 | 0.711 | 0.726 | 0.472 |
| DeepSeek-V3 | 0.864 | 0.837 | 0.958 | 0.770 | 0.466 |
| Gemini-2.0-Flash | 0.750 | 0.994 | 0.828 | 0.760 | 0.298 |
| GPT-4o | 0.711 | 0.687 | 1.054 | 0.635 | 0.269 |
| FuncRev(TimesFM,Gemini) | 0.819 | 0.693 | 0.829 | 0.787 | 0.390 |
| CodeRev(TimesFM,Gemini) | 0.775 | 0.867 | 0.825 | 0.706 | 0.374 |
| TextRev(TimesFM,Gemini) | 0.769 | 0.787 | 0.784 | 0.718 | 0.363 |
| AvgEnsemble(TimesFM,GPT) | 0.753 | 0.929 | 0.745 | 0.721 | 0.323 |
| AvgEnsemble(TimesFM,Gemini) | 0.761 | 0.904 | 0.740 | 0.724 | 0.330 |

*Table 27.* Detailed MASE results of Table 1. (cont'd, part 9/38)

| Method | diabetes | diammonium_usd_t | domestic_violence | drones | drought |
|---|---|---|---|---|---|
| SeasonalNaive | 1.000 | 1.000 | 1.000 | 1.000 | 1.000 |
| Sundial | 0.611 | 0.934 | 0.631 | 0.924 | 0.753 |
| Moirai2.0 | 0.647 | 0.768 | 0.693 | 0.861 | 0.884 |
| TimesFM2.5 | 0.570 | 0.818 | 0.605 | 0.917 | 0.757 |
| AvgEnsemble(TimesFM,Moirai) | 0.592 | 0.790 | 0.625 | 0.888 | 0.819 |
| DeepSeek-V3 | 0.637 | 0.794 | 0.644 | 0.993 | 0.905 |
| Gemini-2.0-Flash | 0.572 | 0.711 | 0.725 | 0.850 | 0.670 |
| GPT-4o | 0.575 | 0.799 | 0.738 | 0.988 | 0.655 |
| FuncRev(TimesFM,Gemini) | 0.689 | 1.681 | 0.666 | 0.817 | 0.844 |
| CodeRev(TimesFM,Gemini) | 0.599 | 3.869 | 0.622 | 0.911 | 0.849 |
| TextRev(TimesFM,Gemini) | 0.578 | 0.609 | 0.605 | 0.914 | 0.751 |
| AvgEnsemble(TimesFM,GPT) | 0.555 | 0.764 | 0.628 | 0.877 | 0.685 |
| AvgEnsemble(TimesFM,Gemini) | 0.568 | 0.711 | 0.632 | 0.881 | 0.719 |

*Table 28.* Detailed MASE results of Table 1. (cont'd, part 10/38)

| Method | drug_overdose | earthquake | electric_vehicle | endangered_species | esports |
|---|---|---|---|---|---|
| SeasonalNaive | 1.000 | 1.000 | 1.000 | 1.000 | 1.000 |
| Sundial | 0.561 | 0.705 | 0.924 | 0.598 | 0.599 |
| Moirai2.0 | 0.573 | 0.716 | 0.956 | 0.600 | 0.588 |
| TimesFM2.5 | 0.444 | 0.687 | 0.811 | 0.395 | 0.600 |
| AvgEnsemble(TimesFM,Moirai) | 0.471 | 0.694 | 0.875 | 0.462 | 0.572 |
| DeepSeek-V3 | 0.618 | 0.765 | 1.071 | 0.520 | 0.823 |
| Gemini-2.0-Flash | 0.514 | 0.854 | 0.829 | 0.332 | 0.621 |
| GPT-4o | 0.531 | 0.894 | 0.881 | 0.322 | 0.727 |
| FuncRev(TimesFM,Gemini) | 0.628 | 0.728 | 0.759 | 0.449 | 0.685 |
| CodeRev(TimesFM,Gemini) | 0.596 | 0.696 | 0.796 | 0.416 | 0.618 |
| TextRev(TimesFM,Gemini) | 0.432 | 0.726 | 0.805 | 0.411 | 0.615 |
| AvgEnsemble(TimesFM,GPT) | 0.459 | 0.752 | 0.793 | 0.338 | 0.600 |
| AvgEnsemble(TimesFM,Gemini) | 0.461 | 0.727 | 0.789 | 0.337 | 0.576 |

*Table 29.* Detailed MASE results of Table 1. (cont'd, part 11/38)

| Method | ethanol_usd_gal | fashion_week | federal_budget_deficit | federal_reserve | film_festivals |
|---|---|---|---|---|---|
| SeasonalNaive | 1.000 | 1.000 | 1.000 | 1.000 | 1.000 |
| Sundial | 0.861 | 0.532 | 0.566 | 0.646 | 0.701 |
| Moirai2.0 | 0.778 | 0.604 | 0.710 | 0.675 | 0.661 |
| TimesFM2.5 | 0.808 | 0.322 | 0.433 | 0.581 | 0.610 |
| AvgEnsemble(TimesFM,Moirai) | 0.778 | 0.442 | 0.539 | 0.611 | 0.623 |
| DeepSeek-V3 | 0.799 | 0.399 | 0.625 | 0.671 | 0.522 |
| Gemini-2.0-Flash | 0.805 | 0.250 | 0.473 | 0.646 | 0.689 |
| GPT-4o | 0.808 | 0.331 | 0.455 | 0.606 | 0.577 |
| FuncRev(TimesFM,Gemini) | 0.746 | 0.277 | 0.446 | 0.582 | 0.628 |
| CodeRev(TimesFM,Gemini) | 0.846 | 0.279 | 0.500 | 0.583 | 0.600 |
| TextRev(TimesFM,Gemini) | 0.995 | 0.302 | 0.425 | 0.583 | 0.599 |
| AvgEnsemble(TimesFM,GPT) | 0.789 | 0.236 | 0.400 | 0.580 | 0.610 |
| AvgEnsemble(TimesFM,Gemini) | 0.805 | 0.237 | 0.398 | 0.582 | 0.624 |

*Table 30.* Detailed MASE results of Table 1. (cont'd, part 12/38)

| Method | financial_regulation | flooding | flu_shot | food_recall | food_safety |
|---|---|---|---|---|---|
| SeasonalNaive | 1.000 | 1.000 | 1.000 | 1.000 | 1.000 |
| Sundial | 0.940 | 0.693 | 0.341 | 0.761 | 0.950 |
| Moirai2.0 | 0.792 | 0.659 | 0.362 | 0.727 | 0.787 |
| TimesFM2.5 | 0.626 | 0.664 | 0.223 | 0.698 | 0.774 |
| AvgEnsemble(TimesFM,Moirai) | 0.668 | 0.657 | 0.274 | 0.701 | 0.772 |
| DeepSeek-V3 | 0.677 | 0.696 | 0.365 | 1.138 | 0.812 |
| Gemini-2.0-Flash | 0.741 | 0.772 | 0.272 | 1.118 | 0.757 |
| GPT-4o | 0.531 | 0.783 | 0.315 | 1.287 | 0.565 |
| FuncRev(TimesFM,Gemini) | 0.551 | 0.844 | 0.325 | 0.640 | 1.014 |
| CodeRev(TimesFM,Gemini) | 0.616 | 0.780 | 0.287 | 0.776 | 0.762 |
| TextRev(TimesFM,Gemini) | 0.651 | 0.687 | 0.218 | 0.754 | 0.744 |
| AvgEnsemble(TimesFM,GPT) | 0.652 | 0.699 | 0.212 | 0.852 | 0.756 |
| AvgEnsemble(TimesFM,Gemini) | 0.637 | 0.695 | 0.199 | 0.816 | 0.728 |

*Table 31.* Detailed MASE results of Table 1. (cont'd, part 13/38)

| Method | food_wine_festivals | formula_1 | gallium_cny_kg | gas_prices | gasoline_usd_gal |
|---|---|---|---|---|---|
| SeasonalNaive | 1.000 | 1.000 | 1.000 | 1.000 | 1.000 |
| Sundial | 0.811 | 0.678 | 0.875 | 1.006 | 0.840 |
| Moirai2.0 | 0.882 | 0.823 | 0.673 | 0.912 | 0.693 |
| TimesFM2.5 | 0.782 | 0.575 | 0.731 | 0.966 | 0.773 |
| AvgEnsemble(TimesFM,Moirai) | 0.830 | 0.647 | 0.668 | 0.939 | 0.717 |
| DeepSeek-V3 | 0.949 | 0.767 | 0.577 | 1.250 | 0.691 |
| Gemini-2.0-Flash | 0.935 | 0.717 | 0.581 | 1.217 | 0.698 |
| GPT-4o | 0.860 | 0.717 | 0.574 | 1.352 | 0.679 |
| FuncRev(TimesFM,Gemini) | 0.738 | 0.560 | 2.508 | 0.847 | 1.983 |
| CodeRev(TimesFM,Gemini) | 0.765 | 0.621 | 3.055 | 1.072 | 2.390 |
| TextRev(TimesFM,Gemini) | 0.798 | 0.566 | 1.809 | 0.954 | 0.727 |
| AvgEnsemble(TimesFM,GPT) | 0.782 | 0.620 | 0.650 | 1.050 | 0.715 |
| AvgEnsemble(TimesFM,Gemini) | 0.806 | 0.604 | 0.581 | 0.924 | 0.698 |

*Table 32.* Detailed MASE results of Table 1. (cont'd, part 14/38)

| Method | gender_equality | gene_editing | germanium_cny_kg | global_warming | gold_usd_t_oz |
|---|---|---|---|---|---|
| SeasonalNaive | 1.000 | 1.000 | 1.000 | 1.000 | 1.000 |
| Sundial | 0.483 | 0.772 | 0.991 | 0.673 | 1.127 |
| Moirai2.0 | 0.508 | 0.866 | 0.655 | 0.772 | 0.846 |
| TimesFM2.5 | 0.314 | 0.610 | 0.795 | 0.526 | 1.058 |
| AvgEnsemble(TimesFM,Moirai) | 0.377 | 0.708 | 0.660 | 0.633 | 0.944 |
| DeepSeek-V3 | 0.477 | 0.870 | 0.622 | 0.967 | 0.917 |
| Gemini-2.0-Flash | 0.333 | 0.692 | 0.732 | 0.576 | 0.927 |
| GPT-4o | 0.332 | 0.601 | 0.619 | 0.663 | 0.921 |
| FuncRev(TimesFM,Gemini) | 0.446 | 0.685 | 5.698 | 0.547 | 1.239 |
| CodeRev(TimesFM,Gemini) | 0.335 | 0.644 | 4.188 | 0.635 | 1.058 |
| TextRev(TimesFM,Gemini) | 0.327 | 0.639 | 1.322 | 0.534 | 1.172 |
| AvgEnsemble(TimesFM,GPT) | 0.299 | 0.630 | 0.730 | 0.515 | 0.986 |
| AvgEnsemble(TimesFM,Gemini) | 0.299 | 0.609 | 0.732 | 0.530 | 0.927 |

*Table 33.* Detailed MASE results of Table 1. (cont'd, part 15/38)

| Method | goldman_sachs | government_spending | halloween_costumes | healthcare_costs | healthcare_policy |
|---|---|---|---|---|---|
| SeasonalNaive | 1.000 | 1.000 | 1.000 | 1.000 | 1.000 |
| Sundial | 0.592 | 0.723 | 0.564 | 0.897 | 0.972 |
| Moirai2.0 | 0.661 | 0.663 | 0.542 | 0.768 | 0.860 |
| TimesFM2.5 | 0.595 | 0.525 | 0.321 | 0.544 | 0.704 |
| AvgEnsemble(TimesFM,Moirai) | 0.624 | 0.557 | 0.412 | 0.620 | 0.760 |
| DeepSeek-V3 | 0.651 | 0.652 | 0.131 | 0.628 | 0.753 |
| Gemini-2.0-Flash | 0.667 | 0.505 | 0.112 | 0.613 | 0.841 |
| GPT-4o | 0.683 | 0.547 | 0.131 | 0.513 | 0.593 |
| FuncRev(TimesFM,Gemini) | 0.585 | 0.637 | 0.349 | 0.541 | 0.688 |
| CodeRev(TimesFM,Gemini) | 0.597 | 0.539 | 0.770 | 0.517 | 0.659 |
| TextRev(TimesFM,Gemini) | 0.586 | 0.518 | 0.220 | 0.541 | 0.699 |
| AvgEnsemble(TimesFM,GPT) | 0.605 | 0.493 | 0.173 | 0.562 | 0.738 |
| AvgEnsemble(TimesFM,Gemini) | 0.581 | 0.486 | 0.185 | 0.545 | 0.700 |

*Table 34.* Detailed MASE results of Table 1. (cont'd, part 16/38)

| Method | heatwave | heavy_rainfall | hedge_funds | hiv_aids | homelessness |
|---|---|---|---|---|---|
| SeasonalNaive | 1.000 | 1.000 | 1.000 | 1.000 | 1.000 |
| Sundial | 0.465 | 0.813 | 0.845 | 0.688 | 0.673 |
| Moirai2.0 | 0.527 | 0.736 | 0.818 | 0.776 | 0.746 |
| TimesFM2.5 | 0.494 | 0.590 | 0.838 | 0.638 | 0.499 |
| AvgEnsemble(TimesFM,Moirai) | 0.477 | 0.648 | 0.821 | 0.693 | 0.584 |
| DeepSeek-V3 | 0.727 | 0.640 | 0.928 | 0.815 | 1.035 |
| Gemini-2.0-Flash | 0.543 | 0.626 | 0.842 | 0.674 | 0.630 |
| GPT-4o | 0.695 | 0.525 | 0.844 | 0.593 | 0.647 |
| FuncRev(TimesFM,Gemini) | 0.627 | 0.672 | 0.824 | 0.581 | 0.497 |
| CodeRev(TimesFM,Gemini) | 0.516 | 0.597 | 0.815 | 0.652 | 0.622 |
| TextRev(TimesFM,Gemini) | 0.518 | 0.594 | 0.818 | 0.628 | 0.494 |
| AvgEnsemble(TimesFM,GPT) | 0.512 | 0.587 | 0.798 | 0.628 | 0.527 |
| AvgEnsemble(TimesFM,Gemini) | 0.517 | 0.580 | 0.796 | 0.621 | 0.536 |

*Table 35.* Detailed MASE results of Table 1. (cont'd, part 17/38)

| Method | human_rights | immigration_reform | income_inequality | indium_cny_kg | infectious_disease |
|---|---|---|---|---|---|
| SeasonalNaive | 1.000 | 1.000 | 1.000 | 1.000 | 1.000 |
| Sundial | 0.632 | 0.844 | 0.686 | 0.826 | 0.676 |
| Moirai2.0 | 0.627 | 0.824 | 0.549 | 0.846 | 0.780 |
| TimesFM2.5 | 0.429 | 0.670 | 0.285 | 0.787 | 0.588 |
| AvgEnsemble(TimesFM,Moirai) | 0.498 | 0.712 | 0.384 | 0.807 | 0.674 |
| DeepSeek-V3 | 0.705 | 0.901 | 0.593 | 0.804 | 0.738 |
| Gemini-2.0-Flash | 0.433 | 0.734 | 0.311 | 0.812 | 0.558 |
| GPT-4o | 0.527 | 0.779 | 0.324 | 0.786 | 0.515 |
| FuncRev(TimesFM,Gemini) | 0.485 | 0.753 | 0.350 | 1.224 | 0.728 |
| CodeRev(TimesFM,Gemini) | 0.428 | 0.816 | 0.298 | 0.852 | 0.594 |
| TextRev(TimesFM,Gemini) | 0.416 | 0.709 | 0.278 | 0.441 | 0.584 |
| AvgEnsemble(TimesFM,GPT) | 0.407 | 0.682 | 0.283 | 0.769 | 0.557 |
| AvgEnsemble(TimesFM,Gemini) | 0.407 | 0.676 | 0.279 | 0.812 | 0.571 |

*Table 36.* Detailed MASE results of Table 1. (cont'd, part 18/38)

| Method | inflation | infrastructure_spending | international_trade | invasive_species | investment_banking |
|---|---|---|---|---|---|
| SeasonalNaive | 1.000 | 1.000 | 1.000 | 1.000 | 1.000 |
| Sundial | 0.664 | 0.833 | 0.827 | 0.667 | 0.907 |
| Moirai2.0 | 0.682 | 0.692 | 0.706 | 0.527 | 0.779 |
| TimesFM2.5 | 0.667 | 0.652 | 0.422 | 0.321 | 0.648 |
| AvgEnsemble(TimesFM,Moirai) | 0.668 | 0.665 | 0.529 | 0.392 | 0.704 |
| DeepSeek-V3 | 0.718 | 0.661 | 0.657 | 0.367 | 0.841 |
| Gemini-2.0-Flash | 0.600 | 0.759 | 0.441 | 0.298 | 0.741 |
| GPT-4o | 0.661 | 0.526 | 0.390 | 0.275 | 0.684 |
| FuncRev(TimesFM,Gemini) | 0.838 | 0.671 | 0.510 | 0.368 | 0.710 |
| CodeRev(TimesFM,Gemini) | 0.688 | 0.735 | 0.428 | 0.334 | 0.629 |
| TextRev(TimesFM,Gemini) | 0.689 | 0.690 | 0.461 | 0.317 | 0.654 |
| AvgEnsemble(TimesFM,GPT) | 0.603 | 0.684 | 0.407 | 0.287 | 0.659 |
| AvgEnsemble(TimesFM,Gemini) | 0.610 | 0.687 | 0.401 | 0.279 | 0.651 |

*Table 37.* Detailed MASE results of Table 1. (cont'd, part 19/38)

| Method | lead_usd_t | lithium_cny_t | lumber_usd_1000_board_feet | magnesium_cny_t | major_league_baseball |
|---|---|---|---|---|---|
| SeasonalNaive | 1.000 | 1.000 | 1.000 | 1.000 | 1.000 |
| Sundial | 0.775 | 0.815 | 1.015 | 0.960 | 0.536 |
| Moirai2.0 | 0.793 | 0.554 | 0.907 | 0.756 | 0.562 |
| TimesFM2.5 | 0.859 | 0.584 | 1.059 | 0.821 | 0.332 |
| AvgEnsemble(TimesFM,Moirai) | 0.808 | 0.564 | 0.980 | 0.781 | 0.431 |
| DeepSeek-V3 | 0.834 | 0.589 | 0.886 | 0.771 | 0.421 |
| Gemini-2.0-Flash | 0.838 | 0.578 | 0.882 | 0.781 | 0.241 |
| GPT-4o | 0.829 | 0.589 | 0.888 | 0.764 | 0.278 |
| FuncRev(TimesFM,Gemini) | 1.144 | 0.518 | 1.083 | 1.201 | 0.345 |
| CodeRev(TimesFM,Gemini) | 0.916 | 0.482 | 0.987 | 1.023 | 0.366 |
| TextRev(TimesFM,Gemini) | 1.105 | 0.368 | 1.040 | 1.355 | 0.338 |
| AvgEnsemble(TimesFM,GPT) | 0.832 | 0.576 | 0.968 | 0.797 | 0.332 |
| AvgEnsemble(TimesFM,Gemini) | 0.838 | 0.578 | 0.882 | 0.781 | 0.274 |

*Table 38.* Detailed MASE results of Table 1. (cont'd, part 20/38)

| Method | manganese_cny_mtu | marine_life | marine_pollution | mental_health | meta_platforms |
|---|---|---|---|---|---|
| SeasonalNaive | 1.000 | 1.000 | 1.000 | 1.000 | 1.000 |
| Sundial | 0.724 | 0.866 | 0.529 | 0.651 | 0.638 |
| Moirai2.0 | 0.574 | 0.693 | 0.535 | 0.586 | 0.696 |
| TimesFM2.5 | 0.658 | 0.564 | 0.359 | 0.444 | 0.647 |
| AvgEnsemble(TimesFM,Moirai) | 0.598 | 0.592 | 0.397 | 0.491 | 0.669 |
| DeepSeek-V3 | 0.499 | 0.749 | 0.397 | 0.584 | 0.944 |
| Gemini-2.0-Flash | 0.519 | 0.569 | 0.337 | 0.459 | 0.652 |
| GPT-4o | 0.519 | 0.443 | 0.315 | 0.423 | 0.823 |
| FuncRev(TimesFM,Gemini) | 0.442 | 0.497 | 0.343 | 0.457 | 0.311 |
| CodeRev(TimesFM,Gemini) | 0.765 | 0.486 | 0.347 | 0.448 | 0.645 |
| TextRev(TimesFM,Gemini) | 1.089 | 0.537 | 0.353 | 0.425 | 0.651 |
| AvgEnsemble(TimesFM,GPT) | 0.589 | 0.555 | 0.328 | 0.431 | 0.631 |
| AvgEnsemble(TimesFM,Gemini) | 0.519 | 0.537 | 0.326 | 0.440 | 0.640 |

*Table 39.* Detailed MASE results of Table 1. (cont'd, part 21/38)

| Method | meteor_shower | methanol_cny_t | microsoft | milk_usd_cwt | minimum_wage |
|---|---|---|---|---|---|
| SeasonalNaive | 1.000 | 1.000 | 1.000 | 1.000 | 1.000 |
| Sundial | 0.666 | 1.060 | 0.689 | 0.681 | 0.817 |
| Moirai2.0 | 0.597 | 0.941 | 0.668 | 0.605 | 0.716 |
| TimesFM2.5 | 0.533 | 0.898 | 0.607 | 0.607 | 0.660 |
| AvgEnsemble(TimesFM,Moirai) | 0.555 | 0.914 | 0.629 | 0.598 | 0.673 |
| DeepSeek-V3 | 0.532 | 0.892 | 0.802 | 0.577 | 0.987 |
| Gemini-2.0-Flash | 0.329 | 0.877 | 0.649 | 0.585 | 1.051 |
| GPT-4o | 0.514 | 0.902 | 0.650 | 0.575 | 1.249 |
| FuncRev(TimesFM,Gemini) | 0.863 | 0.743 | 0.667 | 1.120 | 0.819 |
| CodeRev(TimesFM,Gemini) | 0.677 | 0.861 | 0.615 | 1.508 | 0.885 |
| TextRev(TimesFM,Gemini) | 0.560 | 0.969 | 0.599 | 0.609 | 0.750 |
| AvgEnsemble(TimesFM,GPT) | 0.346 | 0.882 | 0.621 | 0.591 | 0.747 |
| AvgEnsemble(TimesFM,Gemini) | 0.350 | 0.877 | 0.607 | 0.585 | 0.723 |

*Table 40.* Detailed MASE results of Table 1. (cont'd, part 22/38)

| Method | molybdenum_cny_kg | mortgage_rates | music_festivals | naphtha_usd_t | national_basketball_association |
|---|---|---|---|---|---|
| SeasonalNaive | 1.000 | 1.000 | 1.000 | 1.000 | 1.000 |
| Sundial | 0.853 | 0.685 | 0.532 | 1.118 | 0.589 |
| Moirai2.0 | 0.613 | 0.691 | 0.593 | 0.845 | 0.573 |
| TimesFM2.5 | 0.724 | 0.685 | 0.467 | 0.800 | 0.521 |
| AvgEnsemble(TimesFM,Moirai) | 0.602 | 0.688 | 0.488 | 0.817 | 0.535 |
| DeepSeek-V3 | 0.544 | 0.848 | 0.633 | 0.815 | 0.619 |
| Gemini-2.0-Flash | 0.560 | 0.716 | 0.460 | 0.827 | 0.687 |
| GPT-4o | 0.558 | 0.809 | 0.572 | 0.822 | 0.594 |
| FuncRev(TimesFM,Gemini) | 0.769 | 0.810 | 0.503 | 1.062 | 0.578 |
| CodeRev(TimesFM,Gemini) | 0.685 | 0.700 | 0.453 | 0.933 | 0.546 |
| TextRev(TimesFM,Gemini) | 1.183 | 0.682 | 0.447 | 1.061 | 0.519 |
| AvgEnsemble(TimesFM,GPT) | 0.627 | 0.695 | 0.467 | 0.787 | 0.521 |
| AvgEnsemble(TimesFM,Gemini) | 0.560 | 0.694 | 0.426 | 0.827 | 0.565 |

*Table 41.* Detailed MASE results of Table 1. (cont'd, part 23/38)

| Method | national_debt | national_football_league | neodymium_cny_t | nickel_usd_t | nobel_prize |
|---|---|---|---|---|---|
| SeasonalNaive | 1.000 | 1.000 | 1.000 | 1.000 | 1.000 |
| Sundial | 0.809 | 0.672 | 0.534 | 0.810 | 0.663 |
| Moirai2.0 | 0.815 | 0.657 | 0.614 | 0.775 | 0.634 |
| TimesFM2.5 | 0.841 | 0.610 | 0.549 | 0.741 | 0.573 |
| AvgEnsemble(TimesFM,Moirai) | 0.821 | 0.625 | 0.519 | 0.751 | 0.592 |
| DeepSeek-V3 | 0.801 | 0.710 | 0.542 | 0.778 | 0.609 |
| Gemini-2.0-Flash | 0.804 | 0.561 | 0.552 | 0.773 | 0.499 |
| GPT-4o | 0.812 | 0.648 | 0.543 | 0.779 | 0.475 |
| FuncRev(TimesFM,Gemini) | 0.850 | 0.694 | 0.803 | 0.806 | 0.562 |
| CodeRev(TimesFM,Gemini) | 0.856 | 0.620 | 0.661 | 0.692 | 0.500 |
| TextRev(TimesFM,Gemini) | 0.824 | 0.626 | 0.457 | 0.689 | 0.489 |
| AvgEnsemble(TimesFM,GPT) | 0.806 | 0.610 | 0.548 | 0.739 | 0.519 |
| AvgEnsemble(TimesFM,Gemini) | 0.803 | 0.572 | 0.552 | 0.773 | 0.479 |

*Table 42.* Detailed MASE results of Table 1. (cont'd, part 24/38)

| Method | nvidia | oat_usd_bu | obesity | opioid_crisis | organic_food |
|---|---|---|---|---|---|
| SeasonalNaive | 1.000 | 1.000 | 1.000 | 1.000 | 1.000 |
| Sundial | 0.694 | 0.840 | 0.574 | 0.554 | 0.669 |
| Moirai2.0 | 0.849 | 0.861 | 0.728 | 0.665 | 0.639 |
| TimesFM2.5 | 0.761 | 0.810 | 0.453 | 0.543 | 0.564 |
| AvgEnsemble(TimesFM,Moirai) | 0.803 | 0.827 | 0.577 | 0.562 | 0.565 |
| DeepSeek-V3 | 1.121 | 0.888 | 1.111 | 0.570 | 0.627 |
| Gemini-2.0-Flash | 0.691 | 0.877 | 0.701 | 0.554 | 0.469 |
| GPT-4o | 0.854 | 0.880 | 0.728 | 0.611 | 0.471 |
| FuncRev(TimesFM,Gemini) | 0.769 | 0.811 | 0.883 | 0.563 | 0.548 |
| CodeRev(TimesFM,Gemini) | 0.701 | 0.805 | 0.577 | 0.526 | 0.561 |
| TextRev(TimesFM,Gemini) | 0.649 | 0.715 | 0.461 | 0.533 | 0.547 |
| AvgEnsemble(TimesFM,GPT) | 0.684 | 0.837 | 0.544 | 0.504 | 0.480 |
| AvgEnsemble(TimesFM,Gemini) | 0.685 | 0.877 | 0.456 | 0.495 | 0.479 |

*Table 43.* Detailed MASE results of Table 1. (cont'd, part 25/38)

| Method | palladium_usd_t_oz | pest_control | pet_adoption | pet_health | platinum_usd_t_oz |
|---|---|---|---|---|---|
| SeasonalNaive | 1.000 | 1.000 | 1.000 | 1.000 | 1.000 |
| Sundial | 0.812 | 0.435 | 0.815 | 1.123 | 0.865 |
| Moirai2.0 | 0.749 | 0.394 | 0.780 | 0.887 | 0.703 |
| TimesFM2.5 | 0.800 | 0.349 | 0.730 | 0.870 | 0.766 |
| AvgEnsemble(TimesFM,Moirai) | 0.761 | 0.367 | 0.752 | 0.874 | 0.729 |
| DeepSeek-V3 | 0.773 | 0.438 | 0.965 | 1.039 | 0.667 |
| Gemini-2.0-Flash | 0.759 | 0.387 | 0.862 | 1.111 | 0.673 |
| GPT-4o | 0.763 | 0.320 | 1.256 | 0.794 | 0.671 |
| FuncRev(TimesFM,Gemini) | 0.711 | 0.242 | 1.051 | 0.813 | 0.782 |
| CodeRev(TimesFM,Gemini) | 0.911 | 0.337 | 0.904 | 0.898 | 0.734 |
| TextRev(TimesFM,Gemini) | 0.732 | 0.316 | 0.791 | 0.862 | 0.840 |
| AvgEnsemble(TimesFM,GPT) | 0.739 | 0.364 | 0.753 | 0.964 | 0.700 |
| AvgEnsemble(TimesFM,Gemini) | 0.759 | 0.351 | 0.748 | 0.951 | 0.673 |

*Table 44.* Detailed MASE results of Table 1. (cont'd, part 26/38)

| Method | polyethylene_cny_t | polypropylene_cny_t | polyvinyl_cny_t | potatoes_eur_100kg | poultry_brl_kgs |
|---|---|---|---|---|---|
| SeasonalNaive | 1.000 | 1.000 | 1.000 | 1.000 | 1.000 |
| Sundial | 0.810 | 1.122 | 0.869 | 0.823 | 1.052 |
| Moirai2.0 | 0.746 | 0.945 | 0.600 | 0.677 | 0.834 |
| TimesFM2.5 | 0.793 | 0.880 | 0.749 | 0.752 | 0.928 |
| AvgEnsemble(TimesFM,Moirai) | 0.765 | 0.907 | 0.669 | 0.708 | 0.877 |
| DeepSeek-V3 | 0.677 | 0.950 | 0.606 | 0.672 | 0.771 |
| Gemini-2.0-Flash | 0.673 | 0.936 | 0.586 | 0.679 | 0.783 |
| GPT-4o | 0.680 | 0.930 | 0.584 | 0.667 | 0.783 |
| FuncRev(TimesFM,Gemini) | 0.749 | 0.852 | 0.734 | 0.779 | 1.041 |
| CodeRev(TimesFM,Gemini) | 0.786 | 0.879 | 0.698 | 0.749 | 0.943 |
| TextRev(TimesFM,Gemini) | 0.699 | 0.866 | 0.927 | 0.781 | 1.307 |
| AvgEnsemble(TimesFM,GPT) | 0.727 | 0.897 | 0.649 | 0.693 | 0.847 |
| AvgEnsemble(TimesFM,Gemini) | 0.673 | 0.936 | 0.586 | 0.679 | 0.783 |

*Table 45.* Detailed MASE results of Table 1. (cont'd, part 27/38)

| Method | presidential_election | private_equity | propane_usd_gal | protest | quantum_computing |
|---|---|---|---|---|---|
| SeasonalNaive | 1.000 | 1.000 | 1.000 | 1.000 | 1.000 |
| Sundial | 0.847 | 1.059 | 0.840 | 0.910 | 0.870 |
| Moirai2.0 | 0.839 | 0.820 | 0.712 | 0.907 | 0.888 |
| TimesFM2.5 | 0.847 | 0.809 | 0.732 | 0.859 | 0.908 |
| AvgEnsemble(TimesFM,Moirai) | 0.843 | 0.807 | 0.717 | 0.878 | 0.898 |
| DeepSeek-V3 | 0.824 | 0.932 | 0.756 | 0.879 | 0.787 |
| Gemini-2.0-Flash | 0.827 | 0.878 | 0.756 | 0.985 | 0.930 |
| GPT-4o | 0.831 | 0.698 | 0.759 | 1.025 | 0.843 |
| FuncRev(TimesFM,Gemini) | 0.733 | 0.690 | 0.842 | 1.132 | 0.693 |
| CodeRev(TimesFM,Gemini) | 0.857 | 0.787 | 0.751 | 0.977 | 0.863 |
| TextRev(TimesFM,Gemini) | 0.847 | 0.817 | 1.014 | 0.878 | 0.881 |
| AvgEnsemble(TimesFM,GPT) | 0.837 | 0.835 | 0.728 | 0.886 | 0.907 |
| AvgEnsemble(TimesFM,Gemini) | 0.838 | 0.800 | 0.756 | 0.857 | 0.899 |

*Table 46.* Detailed MASE results of Table 1. (cont'd, part 28/38)

| Method | rapeseed_eur_t | refugee_support | renewable_energy | rhodium_usd_t_oz | rice_usd_cwt |
|---|---|---|---|---|---|
| SeasonalNaive | 1.000 | 1.000 | 1.000 | 1.000 | 1.000 |
| Sundial | 0.856 | 0.629 | 0.638 | 1.052 | 0.792 |
| Moirai2.0 | 0.664 | 0.707 | 0.674 | 0.979 | 0.715 |
| TimesFM2.5 | 0.722 | 0.491 | 0.436 | 0.987 | 0.740 |
| AvgEnsemble(TimesFM,Moirai) | 0.674 | 0.591 | 0.515 | 0.973 | 0.723 |
| DeepSeek-V3 | 0.777 | 0.665 | 0.499 | 0.917 | 0.653 |
| Gemini-2.0-Flash | 0.773 | 0.531 | 0.420 | 0.914 | 0.671 |
| GPT-4o | 0.778 | 0.575 | 0.346 | 0.907 | 0.647 |
| FuncRev(TimesFM,Gemini) | 0.683 | 0.644 | 0.464 | 1.070 | 0.941 |
| CodeRev(TimesFM,Gemini) | 0.710 | 0.624 | 0.391 | 0.786 | 0.868 |
| TextRev(TimesFM,Gemini) | 0.862 | 0.584 | 0.435 | 0.876 | 0.923 |
| AvgEnsemble(TimesFM,GPT) | 0.686 | 0.498 | 0.404 | 0.945 | 0.688 |
| AvgEnsemble(TimesFM,Gemini) | 0.773 | 0.488 | 0.385 | 0.914 | 0.671 |

*Table 47.* Detailed MASE results of Table 1. (cont'd, part 29/38)

| Method | robotics | rocket_launch | rubber_usd_cents_kg | salmon_nok_kg | silver_usd_t_oz |
|---|---|---|---|---|---|
| SeasonalNaive | 1.000 | 1.000 | 1.000 | 1.000 | 1.000 |
| Sundial | 1.014 | 0.834 | 1.010 | 0.786 | 0.946 |
| Moirai2.0 | 0.832 | 0.768 | 0.818 | 1.013 | 0.847 |
| TimesFM2.5 | 0.786 | 0.797 | 0.904 | 0.841 | 0.954 |
| AvgEnsemble(TimesFM,Moirai) | 0.796 | 0.778 | 0.855 | 0.919 | 0.887 |
| DeepSeek-V3 | 1.014 | 1.087 | 0.814 | 1.357 | 0.822 |
| Gemini-2.0-Flash | 0.962 | 1.030 | 0.806 | 1.326 | 0.818 |
| GPT-4o | 0.788 | 1.210 | 0.822 | 1.317 | 0.828 |
| FuncRev(TimesFM,Gemini) | 0.810 | 0.852 | 0.955 | – | 0.981 |
| CodeRev(TimesFM,Gemini) | 0.760 | 0.899 | 0.944 | 0.860 | 0.941 |
| TextRev(TimesFM,Gemini) | 0.744 | 0.848 | 1.126 | 0.790 | 0.729 |
| AvgEnsemble(TimesFM,GPT) | 0.850 | 0.891 | 0.849 | 1.059 | 0.862 |
| AvgEnsemble(TimesFM,Gemini) | 0.868 | 0.858 | 0.806 | 1.326 | 0.818 |

*Table 48.* Detailed MASE results of Table 1. (cont'd, part 30/38)

| Method | ski_gear | soybeans_usd_bu | space_exploration | steel_cny_t | stock_market |
|---|---|---|---|---|---|
| SeasonalNaive | 1.000 | 1.000 | 1.000 | 1.000 | 1.000 |
| Sundial | 0.293 | 0.900 | 0.642 | 1.025 | 0.747 |
| Moirai2.0 | 0.344 | 0.850 | 0.577 | 0.918 | 0.750 |
| TimesFM2.5 | 0.231 | 0.801 | 0.405 | 0.958 | 0.729 |
| AvgEnsemble(TimesFM,Moirai) | 0.275 | 0.803 | 0.453 | 0.932 | 0.738 |
| DeepSeek-V3 | 0.247 | 0.866 | 0.499 | 0.877 | 1.045 |
| Gemini-2.0-Flash | 0.225 | 0.908 | 0.418 | 0.866 | 0.758 |
| GPT-4o | 0.245 | 0.864 | 0.455 | 0.873 | 0.832 |
| FuncRev(TimesFM,Gemini) | 0.276 | 0.846 | 0.409 | 1.113 | 0.651 |
| CodeRev(TimesFM,Gemini) | 0.237 | 0.876 | 0.399 | 0.964 | 0.729 |
| TextRev(TimesFM,Gemini) | 0.216 | 0.821 | 0.409 | 1.056 | 0.721 |
| AvgEnsemble(TimesFM,GPT) | 0.223 | 0.822 | 0.397 | 0.892 | 0.739 |
| AvgEnsemble(TimesFM,Gemini) | 0.225 | 0.908 | 0.393 | 0.866 | 0.730 |

*Table 49.* Detailed MASE results of Table 1. (cont'd, part 31/38)

| Method | student_loans | sugar_usd_lbs | sustainable_fashion | tax_software | taxes |
|---|---|---|---|---|---|
| SeasonalNaive | 1.000 | 1.000 | 1.000 | 1.000 | 1.000 |
| Sundial | 0.831 | 1.218 | 0.621 | 0.805 | 0.696 |
| Moirai2.0 | 0.847 | 1.056 | 0.632 | 0.659 | 0.545 |
| TimesFM2.5 | 0.854 | 0.934 | 0.537 | 0.487 | 0.346 |
| AvgEnsemble(TimesFM,Moirai) | 0.846 | 0.986 | 0.568 | 0.538 | 0.423 |
| DeepSeek-V3 | 0.908 | 0.943 | 0.622 | 0.437 | 0.526 |
| Gemini-2.0-Flash | 0.844 | 0.946 | 0.648 | 0.379 | 0.191 |
| GPT-4o | 0.982 | 0.953 | 0.526 | 0.363 | 0.180 |
| FuncRev(TimesFM,Gemini) | 1.052 | 0.959 | 0.567 | 0.462 | 0.375 |
| CodeRev(TimesFM,Gemini) | 0.864 | 1.043 | 0.529 | 0.484 | 0.376 |
| TextRev(TimesFM,Gemini) | 0.852 | 1.085 | 0.515 | 0.466 | 0.274 |
| AvgEnsemble(TimesFM,GPT) | 0.830 | 0.901 | 0.559 | 0.404 | 0.250 |
| AvgEnsemble(TimesFM,Gemini) | 0.819 | 0.946 | 0.563 | 0.396 | 0.245 |

*Table 50.* Detailed MASE results of Table 1. (cont'd, part 32/38)

| Method | tellurium_cny_kg | tesla | tin_usd_t | titanium_cny_kg | tour_de_france |
|---|---|---|---|---|---|
| SeasonalNaive | 1.000 | 1.000 | 1.000 | 1.000 | 1.000 |
| Sundial | 0.765 | 0.714 | 0.835 | 0.781 | 0.557 |
| Moirai2.0 | 0.647 | 0.720 | 0.791 | 0.692 | 0.395 |
| TimesFM2.5 | 0.675 | 0.720 | 0.783 | 0.718 | 0.372 |
| AvgEnsemble(TimesFM,Moirai) | 0.652 | 0.717 | 0.779 | 0.694 | 0.355 |
| DeepSeek-V3 | 0.709 | 0.679 | 0.738 | 0.614 | 0.354 |
| Gemini-2.0-Flash | 0.711 | 0.743 | 0.740 | 0.604 | 0.285 |
| GPT-4o | 0.699 | 0.652 | 0.747 | 0.610 | 0.257 |
| FuncRev(TimesFM,Gemini) | 1.352 | 0.591 | 1.029 | 0.795 | 0.536 |
| CodeRev(TimesFM,Gemini) | 0.680 | 0.709 | 0.757 | 0.745 | 1.438 |
| TextRev(TimesFM,Gemini) | 1.379 | 0.695 | 0.765 | 0.319 | 0.358 |
| AvgEnsemble(TimesFM,GPT) | 0.692 | 0.723 | 0.744 | 0.659 | 0.284 |
| AvgEnsemble(TimesFM,Gemini) | 0.711 | 0.703 | 0.740 | 0.604 | 0.299 |

*Table 51.* Detailed MASE results of Table 1. (cont'd, part 33/38)

| Method | traffic_insurance | unemployment_rate | uranium_usd_lbs | urea_usd_t | usdtoaud_exchangerate |
|---|---|---|---|---|---|
| SeasonalNaive | 1.000 | 1.000 | 1.000 | 1.000 | 1.000 |
| Sundial | 1.056 | 0.651 | 1.050 | 0.729 | 1.019 |
| Moirai2.0 | 0.935 | 0.621 | 0.992 | 0.500 | 0.869 |
| TimesFM2.5 | 0.950 | 0.497 | 1.100 | 0.646 | 0.866 |
| AvgEnsemble(TimesFM,Moirai) | 0.942 | 0.527 | 1.037 | 0.559 | 0.858 |
| DeepSeek-V3 | 0.887 | 0.731 | 0.863 | 0.492 | 0.841 |
| Gemini-2.0-Flash | 1.048 | 0.529 | 0.869 | 0.507 | 0.807 |
| GPT-4o | 0.774 | 0.625 | 0.871 | 0.495 | 0.833 |
| FuncRev(TimesFM,Gemini) | 0.729 | 0.835 | 1.074 | 0.601 | 0.764 |
| CodeRev(TimesFM,Gemini) | 0.904 | 0.505 | 1.119 | 0.610 | 0.815 |
| TextRev(TimesFM,Gemini) | 0.947 | 0.504 | 0.937 | 0.626 | 0.742 |
| AvgEnsemble(TimesFM,GPT) | 0.970 | 0.488 | 0.975 | 0.571 | 0.828 |
| AvgEnsemble(TimesFM,Gemini) | 0.952 | 0.495 | 0.869 | 0.507 | 0.828 |

*Table 52.* Detailed MASE results of Table 1. (cont'd, part 34/38)

| Method | usdtobrl_exchangerate | usdtocad_exchangerate | usdtochf_exchangerate | usdtogbp_exchangerate | usdtohkd_exchangerate |
|---|---|---|---|---|---|
| SeasonalNaive | 1.000 | 1.000 | 1.000 | 1.000 | 1.000 |
| Sundial | 0.955 | 1.013 | 1.135 | 1.321 | 0.820 |
| Moirai2.0 | 0.774 | 0.726 | 0.900 | 0.828 | 0.778 |
| TimesFM2.5 | 0.703 | 0.714 | 1.028 | 0.801 | 0.748 |
| AvgEnsemble(TimesFM,Moirai) | 0.727 | 0.701 | 0.962 | 0.812 | 0.761 |
| DeepSeek-V3 | 0.695 | 0.629 | 0.852 | 0.773 | 0.781 |
| Gemini-2.0-Flash | 0.697 | 0.636 | 0.878 | 0.778 | 0.810 |
| GPT-4o | 0.696 | 0.612 | 0.852 | 0.779 | 0.784 |
| FuncRev(TimesFM,Gemini) | 0.772 | 0.692 | 1.125 | 0.689 | 0.754 |
| CodeRev(TimesFM,Gemini) | 0.727 | 0.708 | 1.020 | 0.817 | 0.750 |
| TextRev(TimesFM,Gemini) | 0.892 | 0.761 | 0.588 | 0.687 | 0.797 |
| AvgEnsemble(TimesFM,GPT) | 0.676 | 0.654 | 0.970 | 0.781 | 0.776 |
| AvgEnsemble(TimesFM,Gemini) | 0.676 | 0.654 | 0.970 | 0.781 | 0.776 |

*Table 53.* Detailed MASE results of Table 1. (cont'd, part 35/38)

| Method | usdtoinr_exchangerate | usdtokrw_exchangerate | usdtomxn_exchangerate | usdtosgd_exchangerate | used_car |
|---|---|---|---|---|---|
| SeasonalNaive | 1.000 | 1.000 | 1.000 | 1.000 | 1.000 |
| Sundial | 0.906 | 1.198 | 1.123 | 1.337 | 0.719 |
| Moirai2.0 | 0.736 | 0.922 | 0.804 | 0.983 | 0.775 |
| TimesFM2.5 | 0.828 | 0.954 | 0.914 | 0.975 | 0.663 |
| AvgEnsemble(TimesFM,Moirai) | 0.779 | 0.929 | 0.852 | 0.970 | 0.703 |
| DeepSeek-V3 | 0.720 | 0.858 | 0.767 | 0.933 | 0.797 |
| Gemini-2.0-Flash | 0.737 | 0.866 | 0.766 | 0.910 | 0.644 |
| GPT-4o | 0.733 | 0.859 | 0.749 | 0.909 | 0.738 |
| FuncRev(TimesFM,Gemini) | 0.790 | 0.941 | 1.073 | 0.836 | 0.741 |
| CodeRev(TimesFM,Gemini) | 0.843 | 0.947 | 1.305 | 0.950 | 0.642 |
| TextRev(TimesFM,Gemini) | 0.641 | 1.028 | 0.824 | 0.886 | 0.635 |
| AvgEnsemble(TimesFM,GPT) | 0.780 | 0.898 | 0.822 | 0.938 | 0.644 |
| AvgEnsemble(TimesFM,Gemini) | 0.780 | 0.898 | 0.822 | 0.938 | 0.658 |

*Table 54.* Detailed MASE results of Table 1. (cont'd, part 36/38)

| Method | vaccine_research | venture_capital | volcanic_eruption | water_scarcity | wheat_usd_bu |
|---|---|---|---|---|---|
| SeasonalNaive | 1.000 | 1.000 | 1.000 | 1.000 | 1.000 |
| Sundial | 0.855 | 0.966 | 0.592 | 0.688 | 0.832 |
| Moirai2.0 | 0.832 | 0.876 | 0.716 | 0.672 | 0.824 |
| TimesFM2.5 | 0.716 | 0.769 | 0.532 | 0.414 | 0.884 |
| AvgEnsemble(TimesFM,Moirai) | 0.762 | 0.816 | 0.583 | 0.518 | 0.844 |
| DeepSeek-V3 | 0.831 | 0.868 | 0.749 | 0.589 | 0.991 |
| Gemini-2.0-Flash | 0.838 | 0.823 | 0.642 | 0.437 | 0.966 |
| GPT-4o | 0.691 | 0.811 | 0.594 | 0.434 | 0.972 |
| FuncRev(TimesFM,Gemini) | 0.490 | 0.757 | 0.527 | 0.390 | 1.053 |
| CodeRev(TimesFM,Gemini) | 0.677 | 0.797 | 0.600 | 0.440 | 1.010 |
| TextRev(TimesFM,Gemini) | 0.706 | 0.791 | 0.523 | 0.413 | 0.946 |
| AvgEnsemble(TimesFM,GPT) | 0.761 | 0.764 | 0.549 | 0.412 | 0.895 |
| AvgEnsemble(TimesFM,Gemini) | 0.746 | 0.755 | 0.486 | 0.415 | 0.966 |

*Table 55.* Detailed MASE results of Table 1. (cont'd, part 37/38)

| Method | wildfires | wildlife_conservation | wool_aud_100kg | zinc_usd_t |
|---|---|---|---|---|
| SeasonalNaive | 1.000 | 1.000 | 1.000 | 1.000 |
| Sundial | 0.622 | 0.911 | 0.864 | 0.987 |
| Moirai2.0 | 0.618 | 0.809 | 0.767 | 0.681 |
| TimesFM2.5 | 0.617 | 0.690 | 0.794 | 0.724 |
| AvgEnsemble(TimesFM,Moirai) | 0.613 | 0.736 | 0.772 | 0.692 |
| DeepSeek-V3 | 0.665 | 0.863 | 0.744 | 0.691 |
| Gemini-2.0-Flash | 0.650 | 0.830 | 0.742 | 0.695 |
| GPT-4o | 0.752 | 0.614 | 0.742 | 0.697 |
| FuncRev(TimesFM,Gemini) | 0.738 | 0.640 | 0.571 | 0.939 |
| CodeRev(TimesFM,Gemini) | 1.458 | 0.664 | 0.809 | 0.789 |
| TextRev(TimesFM,Gemini) | 0.700 | 0.666 | 0.831 | 0.641 |
| AvgEnsemble(TimesFM,GPT) | 0.630 | 0.741 | 0.764 | 0.698 |
| AvgEnsemble(TimesFM,Gemini) | 0.610 | 0.724 | 0.742 | 0.695 |

*Table 56.* Detailed MASE results of Table 1. (cont'd, part 38/38)

| Method | affordable_housing | air_conditioner | air_pollution | air_travel | alphabet |
|---|---|---|---|---|---|
| SeasonalNaive | 1.000 | 1.000 | 1.000 | 1.000 | 1.000 |
| DeepSeek-V3 (Event) | 0.613 | 0.480 | 0.489 | 0.986 | 0.650 |
| DeepSeek-V3 (AllContext) | 0.701 | 0.495 | 0.523 | 0.950 | 0.586 |
| Gemini-2.0-Flash (Event) | 0.668 | 0.287 | 0.349 | 0.909 | 0.570 |
| Gemini-2.0-Flash (AllContext) | 0.688 | 0.274 | 0.356 | 0.917 | 0.513 |
| GPT-4o (Event) | 0.634 | 0.326 | 0.337 | 0.852 | 0.521 |
| GPT-4o (AllContext) | 0.659 | 0.339 | 0.367 | 0.829 | 0.567 |

*Table 57.* Detailed MASE Results of Event Type Attribution Analysis. (part 1/38)

| Method | aluminum_usd_t | amazon | animal_migration | animal_rescue | animal_welfare |
|---|---|---|---|---|---|
| SeasonalNaive | 1.000 | 1.000 | 1.000 | 1.000 | 1.000 |
| DeepSeek-V3 (Event) | 0.924 | 1.213 | 0.764 | 1.072 | 0.734 |
| DeepSeek-V3 (AllContext) | 0.782 | 1.241 | 0.794 | 1.166 | 0.695 |
| Gemini-2.0-Flash (Event) | 1.095 | 0.521 | 0.708 | 0.924 | 0.666 |
| Gemini-2.0-Flash (AllContext) | 0.795 | 0.570 | 0.696 | 0.956 | 0.647 |
| GPT-4o (Event) | 0.881 | 0.922 | 0.662 | 0.965 | 0.595 |
| GPT-4o (AllContext) | 0.784 | 0.996 | 0.664 | 1.048 | 0.618 |

*Table 58.* Detailed MASE Results of Event Type Attribution Analysis. (cont'd, part 2/38)

| Method | apple_inc | art_exhibitions | artificial_intelligence | asset_management | autonomous_driving |
|---|---|---|---|---|---|
| SeasonalNaive | 1.000 | 1.000 | 1.000 | 1.000 | 1.000 |
| DeepSeek-V3 (Event) | 0.710 | 0.769 | 0.906 | 0.815 | 0.941 |
| DeepSeek-V3 (AllContext) | 0.714 | 0.752 | 0.825 | 0.877 | 0.980 |
| Gemini-2.0-Flash (Event) | 0.682 | 0.769 | 0.872 | 0.841 | 0.985 |
| Gemini-2.0-Flash (AllContext) | 0.690 | 0.767 | 0.967 | 0.887 | 1.056 |
| GPT-4o (Event) | 0.626 | 0.668 | 0.759 | 0.803 | 0.821 |
| GPT-4o (AllContext) | 0.656 | 0.674 | 0.761 | 0.759 | 0.855 |

*Table 59.* Detailed MASE Results of Event Type Attribution Analysis. (cont'd, part 3/38)

| Method | back_to_school | barley_inr_t | beef_brl_kg | beekeeping | biodiversity |
|---|---|---|---|---|---|
| SeasonalNaive | 1.000 | 1.000 | 1.000 | 1.000 | 1.000 |
| DeepSeek-V3 (Event) | 0.291 | 0.976 | 0.576 | 0.620 | 0.406 |
| DeepSeek-V3 (AllContext) | 0.292 | 0.954 | 0.567 | 0.585 | 0.367 |
| Gemini-2.0-Flash (Event) | 0.300 | 0.889 | 0.649 | 0.629 | 0.310 |
| Gemini-2.0-Flash (AllContext) | 0.306 | 0.950 | 0.574 | 0.596 | 0.314 |
| GPT-4o (Event) | 0.263 | 0.912 | 0.457 | 0.533 | 0.239 |
| GPT-4o (AllContext) | 0.257 | 0.956 | 0.559 | 0.517 | 0.281 |

*Table 60.* Detailed MASE Results of Event Type Attribution Analysis. (cont'd, part 4/38)

| Method | bitumen_cny_t | black_friday_deals | brent_usd_bbl | broadway_shows | butter_eur_t |
|---|---|---|---|---|---|
| SeasonalNaive | 1.000 | 1.000 | 1.000 | 1.000 | 1.000 |
| DeepSeek-V3 (Event) | 0.896 | 0.343 | 0.823 | 0.727 | 0.758 |
| DeepSeek-V3 (AllContext) | 0.727 | 0.129 | 0.820 | 0.732 | 0.762 |
| Gemini-2.0-Flash (Event) | 0.841 | 0.552 | 0.824 | 0.627 | 0.894 |
| Gemini-2.0-Flash (AllContext) | 0.733 | 0.571 | 0.818 | 0.666 | 0.773 |
| GPT-4o (Event) | 0.977 | 0.449 | 0.853 | 0.715 | 0.745 |
| GPT-4o (AllContext) | 0.728 | 0.457 | 0.830 | 0.649 | 0.763 |

*Table 61.* Detailed MASE Results of Event Type Attribution Analysis. (cont'd, part 5/38)

| Method | cancer_research | canola_cad_t | carbon_emissions | cheese_usd_lbs | christmas_gifts |
|---|---|---|---|---|---|
| SeasonalNaive | 1.000 | 1.000 | 1.000 | 1.000 | 1.000 |
| DeepSeek-V3 (Event) | 0.815 | 0.794 | 0.916 | 1.055 | 0.116 |
| DeepSeek-V3 (AllContext) | 0.856 | 0.827 | 0.633 | 0.810 | 0.120 |
| Gemini-2.0-Flash (Event) | 0.815 | 0.998 | 0.760 | 0.868 | 0.094 |
| Gemini-2.0-Flash (AllContext) | 0.829 | 0.846 | 0.718 | 0.787 | 0.098 |
| GPT-4o (Event) | 0.756 | 0.850 | 0.669 | 0.867 | 0.088 |
| GPT-4o (AllContext) | 0.718 | 0.817 | 0.665 | 0.789 | 0.119 |

*Table 62.* Detailed MASE Results of Event Type Attribution Analysis. (cont'd, part 6/38)

| Method | climate_change | coal_usd_t | cocoa_usd_t | coffee_usd_lbs | comic_con |
|---|---|---|---|---|---|
| SeasonalNaive | 1.000 | 1.000 | 1.000 | 1.000 | 1.000 |
| DeepSeek-V3 (Event) | 1.771 | 0.625 | 0.797 | 0.819 | 0.408 |
| DeepSeek-V3 (AllContext) | 1.475 | 0.653 | 0.812 | 0.750 | 0.445 |
| Gemini-2.0-Flash (Event) | 0.696 | 0.704 | 0.935 | 0.955 | 0.492 |
| Gemini-2.0-Flash (AllContext) | 1.485 | 0.661 | 0.827 | 0.757 | 0.450 |
| GPT-4o (Event) | 1.950 | 0.731 | 0.767 | 0.895 | 0.518 |
| GPT-4o (AllContext) | 1.359 | 0.655 | 0.768 | 0.757 | 0.507 |

*Table 63.* Detailed MASE Results of Event Type Attribution Analysis. (cont'd, part 7/38)

| Method | consumer_electronics | copper_usd_lbs | corn_usd_bu | cost_of_living | cotton_usd_lbs |
|---|---|---|---|---|---|
| SeasonalNaive | 1.000 | 1.000 | 1.000 | 1.000 | 1.000 |
| DeepSeek-V3 (Event) | 0.753 | 0.732 | 0.865 | 0.778 | 1.311 |
| DeepSeek-V3 (AllContext) | 0.795 | 0.620 | 0.857 | 0.763 | 1.147 |
| Gemini-2.0-Flash (Event) | 0.640 | 0.856 | 0.899 | 0.809 | 1.347 |
| Gemini-2.0-Flash (AllContext) | 0.621 | 0.635 | 0.855 | 0.799 | 1.148 |
| GPT-4o (Event) | 0.649 | 0.740 | 0.911 | 0.712 | 1.196 |
| GPT-4o (AllContext) | 0.716 | 0.625 | 0.850 | 0.732 | 1.141 |

*Table 64.* Detailed MASE Results of Event Type Attribution Analysis. (cont'd, part 8/38)

| Method | cryptocurrency | cybersecurity | data_breach | data_privacy | deforestation |
|---|---|---|---|---|---|
| SeasonalNaive | 1.000 | 1.000 | 1.000 | 1.000 | 1.000 |
| DeepSeek-V3 (Event) | 0.815 | 0.896 | 1.236 | 0.680 | 0.407 |
| DeepSeek-V3 (AllContext) | 0.864 | 0.837 | 0.958 | 0.770 | 0.466 |
| Gemini-2.0-Flash (Event) | 0.764 | 0.942 | 0.800 | 0.764 | 0.302 |
| Gemini-2.0-Flash (AllContext) | 0.750 | 0.994 | 0.828 | 0.760 | 0.298 |
| GPT-4o (Event) | 0.742 | 0.695 | 1.057 | 0.648 | 0.267 |
| GPT-4o (AllContext) | 0.711 | 0.687 | 1.054 | 0.635 | 0.269 |

*Table 65.* Detailed MASE Results of Event Type Attribution Analysis. (cont'd, part 9/38)

| Method | diabetes | diammonium_usd_t | domestic_violence | drones | drought |
|---|---|---|---|---|---|
| SeasonalNaive | 1.000 | 1.000 | 1.000 | 1.000 | 1.000 |
| DeepSeek-V3 (Event) | 0.663 | 0.782 | 0.593 | 0.999 | 0.754 |
| DeepSeek-V3 (AllContext) | 0.637 | 0.794 | 0.644 | 0.993 | 0.905 |
| Gemini-2.0-Flash (Event) | 0.591 | 0.743 | 0.737 | 0.854 | 0.726 |
| Gemini-2.0-Flash (AllContext) | 0.572 | 0.711 | 0.725 | 0.850 | 0.670 |
| GPT-4o (Event) | 0.551 | 0.754 | 0.684 | 0.900 | 0.701 |
| GPT-4o (AllContext) | 0.575 | 0.799 | 0.738 | 0.988 | 0.655 |

*Table 66.* Detailed MASE Results of Event Type Attribution Analysis. (cont'd, part 10/38)

| Method | drug_overdose | earthquake | electric_vehicle | endangered_species | esports |
|---|---|---|---|---|---|
| SeasonalNaive | 1.000 | 1.000 | 1.000 | 1.000 | 1.000 |
| DeepSeek-V3 (Event) | 0.543 | 0.751 | 1.136 | 0.412 | 0.825 |
| DeepSeek-V3 (AllContext) | 0.618 | 0.765 | 1.071 | 0.520 | 0.823 |
| Gemini-2.0-Flash (Event) | 0.521 | 0.806 | 0.820 | 0.328 | 0.609 |
| Gemini-2.0-Flash (AllContext) | 0.514 | 0.854 | 0.829 | 0.332 | 0.621 |
| GPT-4o (Event) | 0.532 | 0.893 | 0.854 | 0.323 | 0.806 |
| GPT-4o (AllContext) | 0.531 | 0.894 | 0.881 | 0.322 | 0.727 |

*Table 67.* Detailed MASE Results of Event Type Attribution Analysis. (cont'd, part 11/38)

| Method | ethanol_usd_gal | fashion_week | federal_budget_deficit | federal_reserve | film_festivals |
|---|---|---|---|---|---|
| SeasonalNaive | 1.000 | 1.000 | 1.000 | 1.000 | 1.000 |
| DeepSeek-V3 (Event) | 1.068 | 0.382 | 0.585 | 0.699 | 0.564 |
| DeepSeek-V3 (AllContext) | 0.799 | 0.399 | 0.625 | 0.671 | 0.522 |
| Gemini-2.0-Flash (Event) | 0.989 | 0.248 | 0.463 | 0.640 | 0.700 |
| Gemini-2.0-Flash (AllContext) | 0.805 | 0.250 | 0.473 | 0.646 | 0.689 |
| GPT-4o (Event) | 0.985 | 0.332 | 0.441 | 0.603 | 0.541 |
| GPT-4o (AllContext) | 0.808 | 0.331 | 0.455 | 0.606 | 0.577 |

*Table 68.* Detailed MASE Results of Event Type Attribution Analysis. (cont'd, part 12/38)

| Method | financial_regulation | flooding | flu_shot | food_recall | food_safety |
|---|---|---|---|---|---|
| SeasonalNaive | 1.000 | 1.000 | 1.000 | 1.000 | 1.000 |
| DeepSeek-V3 (Event) | 0.691 | 0.732 | 0.369 | 1.442 | 0.819 |
| DeepSeek-V3 (AllContext) | 0.677 | 0.696 | 0.365 | 1.138 | 0.812 |
| Gemini-2.0-Flash (Event) | 0.721 | 0.752 | 0.237 | 1.032 | 0.698 |
| Gemini-2.0-Flash (AllContext) | 0.741 | 0.772 | 0.272 | 1.118 | 0.757 |
| GPT-4o (Event) | 0.556 | 0.761 | 0.292 | 1.308 | 0.609 |
| GPT-4o (AllContext) | 0.531 | 0.783 | 0.315 | 1.287 | 0.565 |

*Table 69.* Detailed MASE Results of Event Type Attribution Analysis. (cont'd, part 13/38)

| Method | food_wine_festivals | formula_1 | gallium_cny_kg | gas_prices | gasoline_usd_gal |
|---|---|---|---|---|---|
| SeasonalNaive | 1.000 | 1.000 | 1.000 | 1.000 | 1.000 |
| DeepSeek-V3 (Event) | 0.941 | 0.678 | 0.710 | 1.145 | 0.929 |
| DeepSeek-V3 (AllContext) | 0.949 | 0.767 | 0.577 | 1.250 | 0.691 |
| Gemini-2.0-Flash (Event) | 0.912 | 0.674 | 0.864 | 0.956 | 0.884 |
| Gemini-2.0-Flash (AllContext) | 0.935 | 0.717 | 0.581 | 1.217 | 0.698 |
| GPT-4o (Event) | 0.885 | 0.733 | 0.651 | 1.408 | 0.718 |
| GPT-4o (AllContext) | 0.860 | 0.717 | 0.574 | 1.352 | 0.679 |

*Table 70.* Detailed MASE Results of Event Type Attribution Analysis. (cont'd, part 14/38)

| Method | gender_equality | gene_editing | germanium_cny_kg | global_warming | gold_usd_t_oz |
|---|---|---|---|---|---|
| SeasonalNaive | 1.000 | 1.000 | 1.000 | 1.000 | 1.000 |
| DeepSeek-V3 (Event) | 0.538 | 0.988 | 0.677 | 0.843 | 0.979 |
| DeepSeek-V3 (AllContext) | 0.477 | 0.870 | 0.622 | 0.967 | 0.917 |
| Gemini-2.0-Flash (Event) | 0.329 | 0.637 | 1.466 | 0.597 | 0.812 |
| Gemini-2.0-Flash (AllContext) | 0.333 | 0.692 | 0.732 | 0.576 | 0.927 |
| GPT-4o (Event) | 0.306 | 0.623 | 0.541 | 0.579 | 1.066 |
| GPT-4o (AllContext) | 0.332 | 0.601 | 0.619 | 0.663 | 0.921 |

*Table 71.* Detailed MASE Results of Event Type Attribution Analysis. (cont'd, part 15/38)

| Method | goldman_sachs | government_spending | halloween_costumes | healthcare_costs | healthcare_policy |
|---|---|---|---|---|---|
| SeasonalNaive | 1.000 | 1.000 | 1.000 | 1.000 | 1.000 |
| DeepSeek-V3 (Event) | 0.664 | 0.640 | 0.144 | 0.726 | 0.880 |
| DeepSeek-V3 (AllContext) | 0.651 | 0.652 | 0.131 | 0.628 | 0.753 |
| Gemini-2.0-Flash (Event) | 0.658 | 0.495 | 0.131 | 0.577 | 0.759 |
| Gemini-2.0-Flash (AllContext) | 0.667 | 0.505 | 0.112 | 0.613 | 0.841 |
| GPT-4o (Event) | 0.610 | 0.509 | 0.123 | 0.529 | 0.617 |
| GPT-4o (AllContext) | 0.683 | 0.547 | 0.131 | 0.513 | 0.593 |

*Table 72.* Detailed MASE Results of Event Type Attribution Analysis. (cont'd, part 16/38)

| Method | heatwave | heavy_rainfall | hedge_funds | hiv_aids | homelessness |
|---|---|---|---|---|---|
| SeasonalNaive | 1.000 | 1.000 | 1.000 | 1.000 | 1.000 |
| DeepSeek-V3 (Event) | 0.703 | 0.679 | 0.878 | 0.681 | 0.953 |
| DeepSeek-V3 (AllContext) | 0.727 | 0.640 | 0.928 | 0.815 | 1.035 |
| Gemini-2.0-Flash (Event) | 0.559 | 0.614 | 0.834 | 0.674 | 0.654 |
| Gemini-2.0-Flash (AllContext) | 0.543 | 0.626 | 0.842 | 0.674 | 0.630 |
| GPT-4o (Event) | 0.654 | 0.545 | 0.851 | 0.526 | 0.604 |
| GPT-4o (AllContext) | 0.695 | 0.525 | 0.844 | 0.593 | 0.647 |

*Table 73.* Detailed MASE Results of Event Type Attribution Analysis. (cont'd, part 17/38)

| Method | human_rights | immigration_reform | income_inequality | indium_cny_kg | infectious_disease |
|---|---|---|---|---|---|
| SeasonalNaive | 1.000 | 1.000 | 1.000 | 1.000 | 1.000 |
| DeepSeek-V3 (Event) | 0.675 | 1.008 | 0.642 | 1.056 | 0.835 |
| DeepSeek-V3 (AllContext) | 0.705 | 0.901 | 0.593 | 0.804 | 0.738 |
| Gemini-2.0-Flash (Event) | 0.424 | 0.750 | 0.311 | 1.259 | 0.581 |
| Gemini-2.0-Flash (AllContext) | 0.433 | 0.734 | 0.311 | 0.812 | 0.558 |
| GPT-4o (Event) | 0.445 | 0.778 | 0.303 | 1.204 | 0.518 |
| GPT-4o (AllContext) | 0.527 | 0.779 | 0.324 | 0.786 | 0.515 |

*Table 74.* Detailed MASE Results of Event Type Attribution Analysis. (cont'd, part 18/38)

| Method | inflation | infrastructure_spending | international_trade | invasive_species | investment_banking |
|---|---|---|---|---|---|
| SeasonalNaive | 1.000 | 1.000 | 1.000 | 1.000 | 1.000 |
| DeepSeek-V3 (Event) | 0.781 | 0.723 | 0.605 | 0.375 | 0.697 |
| DeepSeek-V3 (AllContext) | 0.718 | 0.661 | 0.657 | 0.367 | 0.841 |
| Gemini-2.0-Flash (Event) | 0.622 | 0.767 | 0.434 | 0.287 | 0.731 |
| Gemini-2.0-Flash (AllContext) | 0.600 | 0.759 | 0.441 | 0.298 | 0.741 |
| GPT-4o (Event) | 0.658 | 0.570 | 0.371 | 0.259 | 0.636 |
| GPT-4o (AllContext) | 0.661 | 0.526 | 0.390 | 0.275 | 0.684 |

*Table 75.* Detailed MASE Results of Event Type Attribution Analysis. (cont'd, part 19/38)

| Method | lead_usd_t | lithium_cny_t | lumber_usd_1000_board_feet | magnesium_cny_t | major_league_baseball |
|---|---|---|---|---|---|
| SeasonalNaive | 1.000 | 1.000 | 1.000 | 1.000 | 1.000 |
| DeepSeek-V3 (Event) | 1.120 | 0.468 | 1.075 | 0.903 | 0.323 |
| DeepSeek-V3 (AllContext) | 0.834 | 0.589 | 0.886 | 0.771 | 0.421 |
| Gemini-2.0-Flash (Event) | 1.129 | 0.415 | 0.850 | 0.939 | 0.235 |
| Gemini-2.0-Flash (AllContext) | 0.838 | 0.578 | 0.882 | 0.781 | 0.241 |
| GPT-4o (Event) | 1.041 | 0.470 | 0.933 | 0.821 | 0.245 |
| GPT-4o (AllContext) | 0.829 | 0.589 | 0.888 | 0.764 | 0.278 |

*Table 76.* Detailed MASE Results of Event Type Attribution Analysis. (cont'd, part 20/38)

| Method | manganese_cny_mtu | marine_life | marine_pollution | mental_health | meta_platforms |
|---|---|---|---|---|---|
| SeasonalNaive | 1.000 | 1.000 | 1.000 | 1.000 | 1.000 |
| DeepSeek-V3 (Event) | 0.533 | 0.615 | 0.538 | 0.553 | 0.960 |
| DeepSeek-V3 (AllContext) | 0.499 | 0.749 | 0.397 | 0.584 | 0.944 |
| Gemini-2.0-Flash (Event) | 0.498 | 0.536 | 0.337 | 0.476 | 0.666 |
| Gemini-2.0-Flash (AllContext) | 0.519 | 0.569 | 0.337 | 0.459 | 0.652 |
| GPT-4o (Event) | 0.608 | 0.470 | 0.308 | 0.441 | 0.749 |
| GPT-4o (AllContext) | 0.519 | 0.443 | 0.315 | 0.423 | 0.823 |

*Table 77.* Detailed MASE Results of Event Type Attribution Analysis. (cont'd, part 21/38)

| Method | meteor_shower | methanol_cny_t | microsoft | milk_usd_cwt | minimum_wage |
|---|---|---|---|---|---|
| SeasonalNaive | 1.000 | 1.000 | 1.000 | 1.000 | 1.000 |
| DeepSeek-V3 (Event) | 0.528 | 0.998 | 0.691 | 0.689 | 1.005 |
| DeepSeek-V3 (AllContext) | 0.532 | 0.892 | 0.802 | 0.577 | 0.987 |
| Gemini-2.0-Flash (Event) | 0.336 | 0.816 | 0.616 | 0.686 | 1.017 |
| Gemini-2.0-Flash (AllContext) | 0.329 | 0.877 | 0.649 | 0.585 | 1.051 |
| GPT-4o (Event) | 0.562 | 1.028 | 0.649 | 0.630 | 1.334 |
| GPT-4o (AllContext) | 0.514 | 0.902 | 0.650 | 0.575 | 1.249 |

*Table 78.* Detailed MASE Results of Event Type Attribution Analysis. (cont'd, part 22/38)

| Method | molybdenum_cny_kg | mortgage_rates | music_festivals | naphtha_usd_t | national_basketball_association |
|---|---|---|---|---|---|
| SeasonalNaive | 1.000 | 1.000 | 1.000 | 1.000 | 1.000 |
| DeepSeek-V3 (Event) | 0.706 | 0.787 | 0.743 | 0.997 | 0.637 |
| DeepSeek-V3 (AllContext) | 0.544 | 0.848 | 0.633 | 0.815 | 0.619 |
| Gemini-2.0-Flash (Event) | 0.684 | 0.716 | 0.460 | 1.097 | 0.665 |
| Gemini-2.0-Flash (AllContext) | 0.560 | 0.716 | 0.460 | 0.827 | 0.687 |
| GPT-4o (Event) | 0.696 | 0.788 | 0.524 | 1.014 | 0.539 |
| GPT-4o (AllContext) | 0.558 | 0.809 | 0.572 | 0.822 | 0.594 |

*Table 79.* Detailed MASE Results of Event Type Attribution Analysis. (cont'd, part 23/38)

| Method | national_debt | national_football_league | neodymium_cny_t | nickel_usd_t | nobel_prize |
|---|---|---|---|---|---|
| SeasonalNaive | 1.000 | 1.000 | 1.000 | 1.000 | 1.000 |
| DeepSeek-V3 (Event) | 0.808 | 0.759 | 0.704 | 0.821 | 0.543 |
| DeepSeek-V3 (AllContext) | 0.801 | 0.710 | 0.542 | 0.778 | 0.609 |
| Gemini-2.0-Flash (Event) | 0.791 | 0.554 | 0.513 | 0.910 | 0.414 |
| Gemini-2.0-Flash (AllContext) | 0.804 | 0.561 | 0.552 | 0.773 | 0.499 |
| GPT-4o (Event) | 0.818 | 0.632 | 0.535 | 0.801 | 0.550 |
| GPT-4o (AllContext) | 0.812 | 0.648 | 0.543 | 0.779 | 0.475 |

*Table 80.* Detailed MASE Results of Event Type Attribution Analysis. (cont'd, part 24/38)

| Method | nvidia | oat_usd_bu | obesity | opioid_crisis | organic_food |
|---|---|---|---|---|---|
| SeasonalNaive | 1.000 | 1.000 | 1.000 | 1.000 | 1.000 |
| DeepSeek-V3 (Event) | 1.052 | 1.024 | 1.026 | 0.567 | 0.621 |
| DeepSeek-V3 (AllContext) | 1.121 | 0.888 | 1.111 | 0.570 | 0.627 |
| Gemini-2.0-Flash (Event) | 0.730 | 1.027 | 0.520 | 0.557 | 0.476 |
| Gemini-2.0-Flash (AllContext) | 0.691 | 0.877 | 0.701 | 0.554 | 0.469 |
| GPT-4o (Event) | 0.872 | 0.999 | 0.684 | 0.541 | 0.477 |
| GPT-4o (AllContext) | 0.854 | 0.880 | 0.728 | 0.611 | 0.471 |

*Table 81.* Detailed MASE Results of Event Type Attribution Analysis. (cont'd, part 25/38)

| Method | palladium_usd_t_oz | pest_control | pet_adoption | pet_health | platinum_usd_t_oz |
|---|---|---|---|---|---|
| SeasonalNaive | 1.000 | 1.000 | 1.000 | 1.000 | 1.000 |
| DeepSeek-V3 (Event) | 0.916 | 0.407 | 0.991 | 0.953 | 0.752 |
| DeepSeek-V3 (AllContext) | 0.773 | 0.438 | 0.965 | 1.039 | 0.667 |
| Gemini-2.0-Flash (Event) | 0.975 | 0.368 | 0.866 | 1.064 | 0.772 |
| Gemini-2.0-Flash (AllContext) | 0.759 | 0.387 | 0.862 | 1.111 | 0.673 |
| GPT-4o (Event) | 0.869 | 0.303 | 1.039 | 0.820 | 0.701 |
| GPT-4o (AllContext) | 0.763 | 0.320 | 1.256 | 0.794 | 0.671 |

*Table 82.* Detailed MASE Results of Event Type Attribution Analysis. (cont'd, part 26/38)

| Method | polyethylene_cny_t | polypropylene_cny_t | polyvinyl_cny_t | potatoes_eur_100kg | poultry_brl_kgs |
|---|---|---|---|---|---|
| SeasonalNaive | 1.000 | 1.000 | 1.000 | 1.000 | 1.000 |
| DeepSeek-V3 (Event) | 0.769 | 1.059 | 0.708 | 0.731 | 0.828 |
| DeepSeek-V3 (AllContext) | 0.677 | 0.950 | 0.606 | 0.672 | 0.771 |
| Gemini-2.0-Flash (Event) | 0.730 | 0.848 | 0.853 | 0.742 | 0.921 |
| Gemini-2.0-Flash (AllContext) | 0.673 | 0.936 | 0.586 | 0.679 | 0.783 |
| GPT-4o (Event) | 0.817 | 0.892 | 0.658 | 0.612 | 0.888 |
| GPT-4o (AllContext) | 0.680 | 0.930 | 0.584 | 0.667 | 0.783 |

*Table 83.* Detailed MASE Results of Event Type Attribution Analysis. (cont'd, part 27/38)

| Method | presidential_election | private_equity | propane_usd_gal | protest | quantum_computing |
|---|---|---|---|---|---|
| SeasonalNaive | 1.000 | 1.000 | 1.000 | 1.000 | 1.000 |
| DeepSeek-V3 (Event) | 0.829 | 0.721 | 0.950 | 0.922 | 0.908 |
| DeepSeek-V3 (AllContext) | 0.824 | 0.932 | 0.756 | 0.879 | 0.787 |
| Gemini-2.0-Flash (Event) | 0.829 | 0.829 | 0.844 | 0.933 | 0.895 |
| Gemini-2.0-Flash (AllContext) | 0.827 | 0.878 | 0.756 | 0.985 | 0.930 |
| GPT-4o (Event) | 0.817 | 0.702 | 0.937 | 1.024 | 0.858 |
| GPT-4o (AllContext) | 0.831 | 0.698 | 0.759 | 1.025 | 0.843 |

*Table 84.* Detailed MASE Results of Event Type Attribution Analysis. (cont'd, part 28/38)

| Method | rapeseed_eur_t | refugee_support | renewable_energy | rhodium_usd_t_oz | rice_usd_cwt |
|---|---|---|---|---|---|
| SeasonalNaive | 1.000 | 1.000 | 1.000 | 1.000 | 1.000 |
| DeepSeek-V3 (Event) | 1.070 | 0.696 | 0.504 | 0.835 | 0.872 |
| DeepSeek-V3 (AllContext) | 0.777 | 0.665 | 0.499 | 0.917 | 0.653 |
| Gemini-2.0-Flash (Event) | 0.853 | 0.522 | 0.388 | 0.779 | 0.736 |
| Gemini-2.0-Flash (AllContext) | 0.773 | 0.531 | 0.420 | 0.914 | 0.671 |
| GPT-4o (Event) | 1.095 | 0.541 | 0.352 | 0.723 | 0.775 |
| GPT-4o (AllContext) | 0.778 | 0.575 | 0.346 | 0.907 | 0.647 |

*Table 85.* Detailed MASE Results of Event Type Attribution Analysis. (cont'd, part 29/38)

| Method | robotics | rocket_launch | rubber_usd_cents_kg | salmon_nok_kg | silver_usd_t_oz |
|---|---|---|---|---|---|
| SeasonalNaive | 1.000 | 1.000 | 1.000 | 1.000 | 1.000 |
| DeepSeek-V3 (Event) | 0.937 | 1.125 | 1.243 | 1.326 | 1.105 |
| DeepSeek-V3 (AllContext) | 1.014 | 1.087 | 0.814 | 1.357 | 0.822 |
| Gemini-2.0-Flash (Event) | 1.001 | 0.971 | 0.974 | 1.329 | 0.936 |
| Gemini-2.0-Flash (AllContext) | 0.962 | 1.030 | 0.806 | 1.326 | 0.818 |
| GPT-4o (Event) | 0.826 | 1.160 | 1.151 | 1.182 | 0.984 |
| GPT-4o (AllContext) | 0.788 | 1.210 | 0.822 | 1.317 | 0.828 |

*Table 86.* Detailed MASE Results of Event Type Attribution Analysis. (cont'd, part 30/38)

| Method | ski_gear | soybeans_usd_bu | space_exploration | steel_cny_t | stock_market |
|---|---|---|---|---|---|
| SeasonalNaive | 1.000 | 1.000 | 1.000 | 1.000 | 1.000 |
| DeepSeek-V3 (Event) | 0.275 | 1.035 | 0.471 | 1.215 | 1.059 |
| DeepSeek-V3 (AllContext) | 0.247 | 0.866 | 0.499 | 0.877 | 1.045 |
| Gemini-2.0-Flash (Event) | 0.233 | 1.260 | 0.417 | 1.023 | 0.735 |
| Gemini-2.0-Flash (AllContext) | 0.225 | 0.908 | 0.418 | 0.866 | 0.758 |
| GPT-4o (Event) | 0.240 | 1.086 | 0.393 | 1.004 | 0.813 |
| GPT-4o (AllContext) | 0.245 | 0.864 | 0.455 | 0.873 | 0.832 |

*Table 87.* Detailed MASE Results of Event Type Attribution Analysis. (cont'd, part 31/38)

| Method | student_loans | sugar_usd_lbs | sustainable_fashion | tax_software | taxes |
|---|---|---|---|---|---|
| SeasonalNaive | 1.000 | 1.000 | 1.000 | 1.000 | 1.000 |
| DeepSeek-V3 (Event) | 0.940 | 1.188 | 0.606 | 0.404 | 0.336 |
| DeepSeek-V3 (AllContext) | 0.908 | 0.943 | 0.622 | 0.437 | 0.526 |
| Gemini-2.0-Flash (Event) | 0.823 | 1.129 | 0.649 | 0.365 | 0.182 |
| Gemini-2.0-Flash (AllContext) | 0.844 | 0.946 | 0.648 | 0.379 | 0.191 |
| GPT-4o (Event) | 0.968 | 1.145 | 0.534 | 0.362 | 0.170 |
| GPT-4o (AllContext) | 0.982 | 0.953 | 0.526 | 0.363 | 0.180 |

*Table 88.* Detailed MASE Results of Event Type Attribution Analysis. (cont'd, part 32/38)

| Method | tellurium_cny_kg | tesla | tin_usd_t | titanium_cny_kg | tour_de_france |
|---|---|---|---|---|---|
| SeasonalNaive | 1.000 | 1.000 | 1.000 | 1.000 | 1.000 |
| DeepSeek-V3 (Event) | 1.479 | 0.667 | 0.855 | 0.675 | 0.305 |
| DeepSeek-V3 (AllContext) | 0.709 | 0.679 | 0.738 | 0.614 | 0.354 |
| Gemini-2.0-Flash (Event) | 0.787 | 0.711 | 1.080 | 0.628 | 0.321 |
| Gemini-2.0-Flash (AllContext) | 0.711 | 0.743 | 0.740 | 0.604 | 0.285 |
| GPT-4o (Event) | 0.704 | 0.629 | 0.686 | 0.703 | 0.198 |
| GPT-4o (AllContext) | 0.699 | 0.652 | 0.747 | 0.610 | 0.257 |

*Table 89.* Detailed MASE Results of Event Type Attribution Analysis. (cont'd, part 33/38)

| Method | traffic_insurance | unemployment_rate | uranium_usd_lbs | urea_usd_t | usdtoaud_exchangerate |
|---|---|---|---|---|---|
| SeasonalNaive | 1.000 | 1.000 | 1.000 | 1.000 | 1.000 |
| DeepSeek-V3 (Event) | 0.947 | 0.714 | 1.036 | 0.418 | 1.275 |
| DeepSeek-V3 (AllContext) | 0.887 | 0.731 | 0.863 | 0.492 | 0.841 |
| Gemini-2.0-Flash (Event) | 0.997 | 0.544 | 1.142 | 0.593 | 1.072 |
| Gemini-2.0-Flash (AllContext) | 1.048 | 0.529 | 0.869 | 0.507 | 0.807 |
| GPT-4o (Event) | 0.765 | 0.557 | 0.993 | 0.428 | 1.195 |
| GPT-4o (AllContext) | 0.774 | 0.625 | 0.871 | 0.495 | 0.833 |

*Table 90.* Detailed MASE Results of Event Type Attribution Analysis. (cont'd, part 34/38)

| Method | usdtobrl_exchangerate | usdtocad_exchangerate | usdtochf_exchangerate | usdtogbp_exchangerate | usdtohkd_exchangerate |
|---|---|---|---|---|---|
| SeasonalNaive | 1.000 | 1.000 | 1.000 | 1.000 | 1.000 |
| DeepSeek-V3 (Event) | 1.065 | 0.999 | 1.110 | 1.178 | 0.919 |
| DeepSeek-V3 (AllContext) | 0.695 | 0.629 | 0.852 | 0.773 | 0.781 |
| Gemini-2.0-Flash (Event) | 1.099 | 0.795 | 0.890 | 0.964 | 0.970 |
| Gemini-2.0-Flash (AllContext) | 0.697 | 0.636 | 0.878 | 0.778 | 0.810 |
| GPT-4o (Event) | 1.004 | 0.835 | 0.932 | 0.860 | 0.881 |
| GPT-4o (AllContext) | 0.696 | 0.612 | 0.852 | 0.779 | 0.784 |

*Table 91.* Detailed MASE Results of Event Type Attribution Analysis. (cont'd, part 35/38)

| Method | usdtoinr_exchangerate | usdtokrw_exchangerate | usdtomxn_exchangerate | usdtosgd_exchangerate | used_car |
|---|---|---|---|---|---|
| SeasonalNaive | 1.000 | 1.000 | 1.000 | 1.000 | 1.000 |
| DeepSeek-V3 (Event) | 0.658 | 0.927 | 1.166 | 1.282 | 0.805 |
| DeepSeek-V3 (AllContext) | 0.720 | 0.858 | 0.767 | 0.933 | 0.797 |
| Gemini-2.0-Flash (Event) | 0.734 | 0.920 | 0.927 | 1.041 | 0.673 |
| Gemini-2.0-Flash (AllContext) | 0.737 | 0.866 | 0.766 | 0.910 | 0.644 |
| GPT-4o (Event) | 0.647 | 0.855 | 0.920 | 0.943 | 0.760 |
| GPT-4o (AllContext) | 0.733 | 0.859 | 0.749 | 0.909 | 0.738 |

*Table 92.* Detailed MASE Results of Event Type Attribution Analysis. (cont'd, part 36/38)

| Method | vaccine_research | venture_capital | volcanic_eruption | water_scarcity | wheat_usd_bu |
|---|---|---|---|---|---|
| SeasonalNaive | 1.000 | 1.000 | 1.000 | 1.000 | 1.000 |
| DeepSeek-V3 (Event) | 0.893 | 1.073 | 0.905 | 0.553 | 1.251 |
| DeepSeek-V3 (AllContext) | 0.831 | 0.868 | 0.749 | 0.589 | 0.991 |
| Gemini-2.0-Flash (Event) | 0.803 | 0.803 | 0.502 | 0.442 | 1.087 |
| Gemini-2.0-Flash (AllContext) | 0.838 | 0.823 | 0.642 | 0.437 | 0.966 |
| GPT-4o (Event) | 0.688 | 0.790 | 0.552 | 0.418 | 1.194 |
| GPT-4o (AllContext) | 0.691 | 0.811 | 0.594 | 0.434 | 0.972 |

*Table 93.* Detailed MASE Results of Event Type Attribution Analysis. (cont'd, part 37/38)

| Method | wildfires | wildlife_conservation | wool_aud_100kg | zinc_usd_t |
|---|---|---|---|---|
| SeasonalNaive | 1.000 | 1.000 | 1.000 | 1.000 |
| DeepSeek-V3 (Event) | 0.699 | 0.764 | 0.765 | 1.013 |
| DeepSeek-V3 (AllContext) | 0.665 | 0.863 | 0.744 | 0.691 |
| Gemini-2.0-Flash (Event) | 0.626 | 0.799 | 0.728 | 0.927 |
| Gemini-2.0-Flash (AllContext) | 0.650 | 0.830 | 0.742 | 0.695 |
| GPT-4o (Event) | 0.750 | 0.653 | 0.768 | 0.899 |
| GPT-4o (AllContext) | 0.752 | 0.614 | 0.742 | 0.697 |

*Table 94.* Detailed MASE Results of Event Type Attribution Analysis. (cont'd, part 38/38)

| Method | affordable_housing | air_conditioner | air_pollution | air_travel | alphabet |
|---|---|---|---|---|---|
| SeasonalNaive | 1.000 | 1.000 | 1.000 | 1.000 | 1.000 |
| Meta | 0.740 | 0.504 | 0.827 | 1.086 | 0.747 |
| Meta+Date | 0.701 | 0.287 | 0.362 | 0.950 | 0.616 |
| Meta+Date+Cov | 0.673 | 0.271 | 0.355 | 0.943 | 0.514 |
| Meta+Date+Event | 0.668 | 0.287 | 0.349 | 0.909 | 0.570 |
| AllContext | 0.688 | 0.274 | 0.356 | 0.917 | 0.513 |

*Table 95.* Detailed MASE Results of Event Type Deep Analysis Using Gemini-2.0-Flash (part 1/38)

| Method | aluminum_usd_t | amazon | animal_migration | animal_rescue | animal_welfare |
|---|---|---|---|---|---|
| SeasonalNaive | 1.000 | 1.000 | 1.000 | 1.000 | 1.000 |
| Meta | 0.876 | 0.836 | 0.977 | 0.753 | 0.808 |
| Meta+Date | 0.921 | 0.581 | 0.718 | 0.921 | 0.641 |
| Meta+Date+Cov | 1.191 | 0.597 | 0.727 | 0.871 | 0.633 |
| Meta+Date+Event | 1.095 | 0.521 | 0.708 | 0.924 | 0.666 |
| AllContext | 0.795 | 0.570 | 0.696 | 0.956 | 0.647 |

*Table 96.* Detailed MASE Results of Event Type Deep Analysis Using Gemini-2.0-Flash (cont'd, part 2/38)

| Method | apple_inc | art_exhibitions | artificial_intelligence | asset_management | autonomous_driving |
|---|---|---|---|---|---|
| SeasonalNaive | 1.000 | 1.000 | 1.000 | 1.000 | 1.000 |
| Meta | 0.826 | 0.832 | 1.017 | 0.846 | 0.873 |
| Meta+Date | 0.680 | 0.813 | 0.904 | 0.838 | 1.027 |
| Meta+Date+Cov | 0.684 | 0.775 | 0.946 | 0.874 | 1.035 |
| Meta+Date+Event | 0.682 | 0.769 | 0.872 | 0.841 | 0.985 |
| AllContext | 0.690 | 0.767 | 0.967 | 0.887 | 1.056 |

*Table 97.* Detailed MASE Results of Event Type Deep Analysis Using Gemini-2.0-Flash (cont'd, part 3/38)

| Method | back_to_school | barley_inr_t | beef_brl_kg | beekeeping | biodiversity |
|---|---|---|---|---|---|
| SeasonalNaive | 1.000 | 1.000 | 1.000 | 1.000 | 1.000 |
| Meta | 0.570 | 0.969 | 0.524 | 1.019 | 0.648 |
| Meta+Date | 0.288 | 0.944 | 0.494 | 0.625 | 0.324 |
| Meta+Date+Cov | 0.302 | 1.018 | 0.659 | 0.531 | 0.300 |
| Meta+Date+Event | 0.300 | 0.889 | 0.649 | 0.629 | 0.310 |
| AllContext | 0.306 | 0.950 | 0.574 | 0.596 | 0.314 |

*Table 98.* Detailed MASE Results of Event Type Deep Analysis Using Gemini-2.0-Flash (cont'd, part 4/38)

| Method | bitumen_cny_t | black_friday_deals | brent_usd_bbl | broadway_shows | butter_eur_t |
|---|---|---|---|---|---|
| SeasonalNaive | 1.000 | 1.000 | 1.000 | 1.000 | 1.000 |
| Meta | 0.737 | 1.262 | 0.931 | 0.748 | 0.768 |
| Meta+Date | 0.734 | 0.563 | 0.830 | 0.637 | 0.837 |
| Meta+Date+Cov | 0.846 | 0.543 | 0.785 | 0.692 | 0.873 |
| Meta+Date+Event | 0.841 | 0.552 | 0.824 | 0.627 | 0.894 |
| AllContext | 0.733 | 0.571 | 0.818 | 0.666 | 0.773 |

*Table 99.* Detailed MASE Results of Event Type Deep Analysis Using Gemini-2.0-Flash (cont'd, part 5/38)

| Method | cancer_research | canola_cad_t | carbon_emissions | cheese_usd_lbs | christmas_gifts |
|---|---|---|---|---|---|
| SeasonalNaive | 1.000 | 1.000 | 1.000 | 1.000 | 1.000 |
| Meta | 0.883 | 0.772 | 0.838 | 0.771 | 0.225 |
| Meta+Date | 0.832 | 0.868 | 0.774 | 0.861 | 0.093 |
| Meta+Date+Cov | 0.857 | 0.940 | 0.687 | 0.932 | 0.090 |
| Meta+Date+Event | 0.815 | 0.998 | 0.760 | 0.868 | 0.094 |
| AllContext | 0.829 | 0.846 | 0.718 | 0.787 | 0.098 |

*Table 100.* Detailed MASE Results of Event Type Deep Analysis Using Gemini-2.0-Flash (cont'd, part 6/38)

| Method | climate_change | coal_usd_t | cocoa_usd_t | coffee_usd_lbs | comic_con |
|---|---|---|---|---|---|
| SeasonalNaive | 1.000 | 1.000 | 1.000 | 1.000 | 1.000 |
| Meta | 0.893 | 0.736 | 0.884 | 0.788 | 0.812 |
| Meta+Date | 1.213 | 0.712 | 0.885 | 0.745 | 0.465 |
| Meta+Date+Cov | 1.262 | 0.610 | 1.000 | 0.850 | 0.490 |
| Meta+Date+Event | 0.696 | 0.704 | 0.935 | 0.955 | 0.492 |
| AllContext | 1.485 | 0.661 | 0.827 | 0.757 | 0.450 |

*Table 101.* Detailed MASE Results of Event Type Deep Analysis Using Gemini-2.0-Flash (cont'd, part 7/38)

| Method | consumer_electronics | copper_usd_lbs | corn_usd_bu | cost_of_living | cotton_usd_lbs |
|---|---|---|---|---|---|
| SeasonalNaive | 1.000 | 1.000 | 1.000 | 1.000 | 1.000 |
| Meta | 0.857 | 0.733 | 0.822 | 0.699 | 0.844 |
| Meta+Date | 0.634 | 0.816 | 0.966 | 0.801 | 1.274 |
| Meta+Date+Cov | 0.583 | 0.794 | 1.037 | 0.710 | 1.263 |
| Meta+Date+Event | 0.640 | 0.856 | 0.899 | 0.809 | 1.347 |
| AllContext | 0.621 | 0.635 | 0.855 | 0.799 | 1.148 |

*Table 102.* Detailed MASE Results of Event Type Deep Analysis Using Gemini-2.0-Flash (cont'd, part 8/38)

| Method | cryptocurrency | cybersecurity | data_breach | data_privacy | deforestation |
|---|---|---|---|---|---|
| SeasonalNaive | 1.000 | 1.000 | 1.000 | 1.000 | 1.000 |
| Meta | 0.809 | 0.946 | 0.922 | 0.879 | 0.634 |
| Meta+Date | 0.793 | 0.878 | 0.873 | 0.754 | 0.304 |
| Meta+Date+Cov | 0.769 | 0.956 | 0.798 | 0.756 | 0.315 |
| Meta+Date+Event | 0.764 | 0.942 | 0.800 | 0.764 | 0.302 |
| AllContext | 0.750 | 0.994 | 0.828 | 0.760 | 0.298 |

*Table 103.* Detailed MASE Results of Event Type Deep Analysis Using Gemini-2.0-Flash (cont'd, part 9/38)

| Method | diabetes | diammonium_usd_t | domestic_violence | drones | drought |
|---|---|---|---|---|---|
| SeasonalNaive | 1.000 | 1.000 | 1.000 | 1.000 | 1.000 |
| Meta | 0.789 | 0.797 | 0.749 | 0.827 | 0.789 |
| Meta+Date | 0.611 | 0.799 | 0.742 | 0.879 | 0.676 |
| Meta+Date+Cov | 0.568 | 0.785 | 0.710 | 0.853 | 0.651 |
| Meta+Date+Event | 0.591 | 0.743 | 0.737 | 0.854 | 0.726 |
| AllContext | 0.572 | 0.711 | 0.725 | 0.850 | 0.670 |

*Table 104.* Detailed MASE Results of Event Type Deep Analysis Using Gemini-2.0-Flash (cont'd, part 10/38)

| Method | drug_overdose | earthquake | electric_vehicle | endangered_species | esports |
|---|---|---|---|---|---|
| SeasonalNaive | 1.000 | 1.000 | 1.000 | 1.000 | 1.000 |
| Meta | 0.815 | 0.689 | 0.843 | 0.703 | 0.671 |
| Meta+Date | 0.514 | 0.942 | 0.818 | 0.329 | 0.642 |
| Meta+Date+Cov | 0.527 | 0.899 | 0.861 | 0.320 | 0.646 |
| Meta+Date+Event | 0.521 | 0.806 | 0.820 | 0.328 | 0.609 |
| AllContext | 0.514 | 0.854 | 0.829 | 0.332 | 0.621 |

*Table 105.* Detailed MASE Results of Event Type Deep Analysis Using Gemini-2.0-Flash (cont'd, part 11/38)

| Method | ethanol_usd_gal | fashion_week | federal_budget_deficit | federal_reserve | film_festivals |
|---|---|---|---|---|---|
| SeasonalNaive | 1.000 | 1.000 | 1.000 | 1.000 | 1.000 |
| Meta | 0.838 | 0.636 | 0.809 | 0.710 | 0.758 |
| Meta+Date | 0.913 | 0.276 | 0.454 | 0.627 | 0.667 |
| Meta+Date+Cov | 1.134 | 0.248 | 0.425 | 0.610 | 0.663 |
| Meta+Date+Event | 0.989 | 0.248 | 0.463 | 0.640 | 0.700 |
| AllContext | 0.805 | 0.250 | 0.473 | 0.646 | 0.689 |

*Table 106.* Detailed MASE Results of Event Type Deep Analysis Using Gemini-2.0-Flash (cont'd, part 12/38)

| Method | financial_regulation | flooding | flu_shot | food_recall | food_safety |
|---|---|---|---|---|---|
| SeasonalNaive | 1.000 | 1.000 | 1.000 | 1.000 | 1.000 |
| Meta | 0.900 | 0.703 | 0.622 | 0.740 | 0.845 |
| Meta+Date | 0.731 | 0.758 | 0.237 | 1.211 | 0.751 |
| Meta+Date+Cov | 0.713 | 0.753 | 0.261 | 1.100 | 0.716 |
| Meta+Date+Event | – | 0.752 | 0.237 | 1.032 | 0.698 |
| AllContext | 0.741 | 0.772 | 0.272 | 1.118 | 0.757 |

*Table 107.* Detailed MASE Results of Event Type Deep Analysis Using Gemini-2.0-Flash (cont'd, part 13/38)

| Method | food_wine_festivals | formula_1 | gallium_cny_kg | gas_prices | gasoline_usd_gal |
|---|---|---|---|---|---|
| SeasonalNaive | 1.000 | 1.000 | 1.000 | 1.000 | 1.000 |
| Meta | 0.851 | 0.908 | 0.579 | 0.908 | 0.686 |
| Meta+Date | 0.929 | 0.664 | 0.579 | 1.276 | 1.172 |
| Meta+Date+Cov | 0.925 | 0.685 | 0.638 | 1.180 | 0.910 |
| Meta+Date+Event | 0.912 | 0.674 | 0.864 | 0.956 | 0.884 |
| AllContext | 0.935 | 0.717 | 0.581 | 1.217 | 0.698 |

*Table 108.* Detailed MASE Results of Event Type Deep Analysis Using Gemini-2.0-Flash (cont'd, part 14/38)

| Method | gender_equality | gene_editing | germanium_cny_kg | global_warming | gold_usd_t_oz |
|---|---|---|---|---|---|
| SeasonalNaive | 1.000 | 1.000 | 1.000 | 1.000 | 1.000 |
| Meta | 0.814 | 0.894 | 0.606 | 0.814 | 0.797 |
| Meta+Date | 0.340 | 0.692 | 0.620 | 0.569 | 1.005 |
| Meta+Date+Cov | 0.319 | 0.637 | 0.615 | 0.578 | 1.076 |
| Meta+Date+Event | 0.329 | 0.637 | 1.466 | 0.597 | 0.812 |
| AllContext | 0.333 | 0.692 | 0.732 | 0.576 | 0.927 |

*Table 109.* Detailed MASE Results of Event Type Deep Analysis Using Gemini-2.0-Flash (cont'd, part 15/38)

| Method | goldman_sachs | government_spending | halloween_costumes | healthcare_costs | healthcare_policy |
|---|---|---|---|---|---|
| SeasonalNaive | 1.000 | 1.000 | 1.000 | 1.000 | 1.000 |
| Meta | 0.686 | 0.905 | 0.322 | 0.869 | 0.846 |
| Meta+Date | 0.714 | 0.555 | 0.144 | 0.605 | 0.777 |
| Meta+Date+Cov | 0.646 | 0.505 | 0.127 | 0.621 | 0.778 |
| Meta+Date+Event | 0.658 | 0.495 | 0.131 | 0.577 | 0.759 |
| AllContext | 0.667 | 0.505 | 0.112 | 0.613 | 0.841 |

*Table 110.* Detailed MASE Results of Event Type Deep Analysis Using Gemini-2.0-Flash (cont'd, part 16/38)

| Method | heatwave | heavy_rainfall | hedge_funds | hiv_aids | homelessness |
|---|---|---|---|---|---|
| SeasonalNaive | 1.000 | 1.000 | 1.000 | 1.000 | 1.000 |
| Meta | 0.667 | 0.928 | 0.820 | 1.220 | 0.721 |
| Meta+Date | 0.548 | 0.634 | 0.848 | 0.645 | 0.709 |
| Meta+Date+Cov | 0.529 | 0.615 | 0.824 | 0.626 | 0.649 |
| Meta+Date+Event | 0.559 | 0.614 | 0.834 | 0.674 | 0.654 |
| AllContext | 0.543 | 0.626 | 0.842 | 0.674 | 0.630 |

*Table 111.* Detailed MASE Results of Event Type Deep Analysis Using Gemini-2.0-Flash (cont'd, part 17/38)

| Method | human_rights | immigration_reform | income_inequality | indium_cny_kg | infectious_disease |
|---|---|---|---|---|---|
| SeasonalNaive | 1.000 | 1.000 | 1.000 | 1.000 | 1.000 |
| Meta | 0.825 | 0.879 | 0.734 | 0.792 | 0.794 |
| Meta+Date | 0.441 | 0.773 | 0.323 | 0.815 | 0.558 |
| Meta+Date+Cov | 0.427 | 0.724 | 0.290 | 0.785 | 0.555 |
| Meta+Date+Event | 0.424 | 0.750 | 0.311 | 1.259 | 0.581 |
| AllContext | 0.433 | 0.734 | 0.311 | 0.812 | 0.558 |

*Table 112.* Detailed MASE Results of Event Type Deep Analysis Using Gemini-2.0-Flash (cont'd, part 18/38)

| Method | inflation | infrastructure_spending | international_trade | invasive_species | investment_banking |
|---|---|---|---|---|---|
| SeasonalNaive | 1.000 | 1.000 | 1.000 | 1.000 | 1.000 |
| Meta | 0.783 | 0.879 | 0.882 | 0.569 | 0.849 |
| Meta+Date | 0.642 | 0.766 | 0.424 | 0.277 | 0.717 |
| Meta+Date+Cov | 0.596 | 0.766 | 0.410 | 0.273 | 0.743 |
| Meta+Date+Event | 0.622 | 0.767 | 0.434 | 0.287 | 0.731 |
| AllContext | 0.600 | 0.759 | 0.441 | 0.298 | 0.741 |

*Table 113.* Detailed MASE Results of Event Type Deep Analysis Using Gemini-2.0-Flash (cont'd, part 19/38)

| Method | lead_usd_t | lithium_cny_t | lumber_usd_1000_board_feet | magnesium_cny_t | major_league_baseball |
|---|---|---|---|---|---|
| SeasonalNaive | 1.000 | 1.000 | 1.000 | 1.000 | 1.000 |
| Meta | 0.883 | 0.731 | 0.987 | 0.748 | 0.612 |
| Meta+Date | 1.036 | 0.504 | 0.996 | 0.846 | 0.248 |
| Meta+Date+Cov | 1.121 | 0.407 | 0.952 | 0.860 | 0.250 |
| Meta+Date+Event | 1.129 | 0.415 | 0.850 | 0.939 | 0.235 |
| AllContext | 0.838 | 0.578 | 0.882 | 0.781 | 0.241 |

*Table 114.* Detailed MASE Results of Event Type Deep Analysis Using Gemini-2.0-Flash (cont'd, part 20/38)

| Method | manganese_cny_mtu | marine_life | marine_pollution | mental_health | meta_platforms |
|---|---|---|---|---|---|
| SeasonalNaive | 1.000 | 1.000 | 1.000 | 1.000 | 1.000 |
| Meta | 0.542 | 0.892 | 0.805 | 0.736 | 0.820 |
| Meta+Date | 0.553 | 0.523 | 0.332 | 0.461 | 0.665 |
| Meta+Date+Cov | 0.625 | 0.545 | 0.348 | 0.469 | 0.663 |
| Meta+Date+Event | 0.498 | 0.536 | 0.337 | 0.476 | 0.666 |
| AllContext | 0.519 | 0.569 | 0.337 | 0.459 | 0.652 |

*Table 115.* Detailed MASE Results of Event Type Deep Analysis Using Gemini-2.0-Flash (cont'd, part 21/38)

| Method | meteor_shower | methanol_cny_t | microsoft | milk_usd_cwt | minimum_wage |
|---|---|---|---|---|---|
| SeasonalNaive | 1.000 | 1.000 | 1.000 | 1.000 | 1.000 |
| Meta | 0.592 | 0.925 | 0.683 | 0.564 | 0.992 |
| Meta+Date | 0.417 | 0.861 | 0.641 | 0.626 | 1.117 |
| Meta+Date+Cov | 0.297 | 1.012 | 0.667 | 0.702 | 0.991 |
| Meta+Date+Event | 0.336 | 0.816 | 0.616 | 0.686 | 1.017 |
| AllContext | 0.329 | 0.877 | 0.649 | 0.585 | 1.051 |

*Table 116.* Detailed MASE Results of Event Type Deep Analysis Using Gemini-2.0-Flash (cont'd, part 22/38)

| Method | molybdenum_cny_kg | mortgage_rates | music_festivals | naphtha_usd_t | national_basketball_association |
|---|---|---|---|---|---|
| SeasonalNaive | 1.000 | 1.000 | 1.000 | 1.000 | 1.000 |
| Meta | 0.570 | 0.747 | 0.935 | 0.776 | 0.766 |
| Meta+Date | 0.558 | 0.747 | 0.490 | 0.984 | 0.688 |
| Meta+Date+Cov | 0.670 | 0.735 | 0.483 | 1.039 | 0.687 |
| Meta+Date+Event | 0.684 | 0.716 | 0.460 | 1.097 | 0.665 |
| AllContext | 0.560 | 0.716 | 0.460 | 0.827 | 0.687 |

*Table 117.* Detailed MASE Results of Event Type Deep Analysis Using Gemini-2.0-Flash (cont'd, part 23/38)

| Method | national_debt | national_football_league | neodymium_cny_t | nickel_usd_t | nobel_prize |
|---|---|---|---|---|---|
| SeasonalNaive | 1.000 | 1.000 | 1.000 | 1.000 | 1.000 |
| Meta | 0.837 | 0.775 | 0.562 | 0.761 | 0.826 |
| Meta+Date | 0.855 | 0.557 | 0.590 | 0.923 | 0.456 |
| Meta+Date+Cov | 0.816 | 0.557 | 0.695 | 1.110 | 0.500 |
| Meta+Date+Event | 0.791 | 0.554 | 0.513 | 0.910 | 0.414 |
| AllContext | 0.804 | 0.561 | 0.552 | 0.773 | 0.499 |

*Table 118.* Detailed MASE Results of Event Type Deep Analysis Using Gemini-2.0-Flash (cont'd, part 24/38)

| Method | nvidia | oat_usd_bu | obesity | opioid_crisis | organic_food |
|---|---|---|---|---|---|
| SeasonalNaive | 1.000 | 1.000 | 1.000 | 1.000 | 1.000 |
| Meta | 0.772 | 0.851 | 0.745 | 0.726 | 0.787 |
| Meta+Date | 0.815 | 0.952 | 0.490 | 0.603 | 0.474 |
| Meta+Date+Cov | 0.734 | 1.106 | 0.546 | 0.541 | 0.463 |
| Meta+Date+Event | 0.730 | 1.027 | 0.520 | 0.557 | 0.476 |
| AllContext | 0.691 | 0.877 | 0.701 | 0.554 | 0.469 |

*Table 119.* Detailed MASE Results of Event Type Deep Analysis Using Gemini-2.0-Flash (cont'd, part 25/38)

| Method | palladium_usd_t_oz | pest_control | pet_adoption | pet_health | platinum_usd_t_oz |
|---|---|---|---|---|---|
| SeasonalNaive | 1.000 | 1.000 | 1.000 | 1.000 | 1.000 |
| Meta | 0.730 | 0.711 | 0.786 | 0.844 | 0.754 |
| Meta+Date | 0.817 | 0.382 | 1.373 | 1.030 | 0.747 |
| Meta+Date+Cov | 0.884 | 0.360 | 0.770 | 1.124 | 0.919 |
| Meta+Date+Event | 0.975 | 0.368 | 0.866 | 1.064 | 0.772 |
| AllContext | 0.759 | 0.387 | 0.862 | 1.111 | 0.673 |

*Table 120.* Detailed MASE Results of Event Type Deep Analysis Using Gemini-2.0-Flash (cont'd, part 26/38)

| Method | polyethylene_cny_t | polypropylene_cny_t | polyvinyl_cny_t | potatoes_eur_100kg | poultry_brl_kgs |
|---|---|---|---|---|---|
| SeasonalNaive | 1.000 | 1.000 | 1.000 | 1.000 | 1.000 |
| Meta | 0.804 | 1.014 | 0.689 | 0.804 | 0.761 |
| Meta+Date | 0.790 | 0.884 | 0.644 | 0.742 | 0.804 |
| Meta+Date+Cov | 0.818 | 0.898 | 0.865 | 0.760 | 0.832 |
| Meta+Date+Event | 0.730 | 0.848 | 0.853 | 0.742 | 0.921 |
| AllContext | 0.673 | 0.936 | 0.586 | 0.679 | 0.783 |

*Table 121.* Detailed MASE Results of Event Type Deep Analysis Using Gemini-2.0-Flash (cont'd, part 27/38)

| Method | presidential_election | private_equity | propane_usd_gal | protest | quantum_computing |
|---|---|---|---|---|---|
| SeasonalNaive | 1.000 | 1.000 | 1.000 | 1.000 | 1.000 |
| Meta | 0.855 | 0.844 | 0.743 | 0.931 | 0.900 |
| Meta+Date | 0.823 | 0.899 | 0.864 | 0.914 | 0.961 |
| Meta+Date+Cov | 0.818 | 0.906 | 1.001 | 0.933 | 0.939 |
| Meta+Date+Event | 0.829 | 0.829 | 0.844 | 0.933 | 0.895 |
| AllContext | 0.827 | 0.878 | 0.756 | 0.985 | 0.930 |

*Table 122.* Detailed MASE Results of Event Type Deep Analysis Using Gemini-2.0-Flash (cont'd, part 28/38)

| Method | rapeseed_eur_t | refugee_support | renewable_energy | rhodium_usd_t_oz | rice_usd_cwt |
|---|---|---|---|---|---|
| SeasonalNaive | 1.000 | 1.000 | 1.000 | 1.000 | 1.000 |
| Meta | 0.814 | 0.762 | 0.930 | 0.915 | 0.747 |
| Meta+Date | 0.856 | 0.541 | 0.413 | 0.945 | 0.714 |
| Meta+Date+Cov | 0.951 | 0.517 | 0.402 | 0.891 | 0.793 |
| Meta+Date+Event | 0.853 | 0.522 | 0.388 | 0.779 | 0.736 |
| AllContext | 0.773 | 0.531 | 0.420 | 0.914 | 0.671 |

*Table 123.* Detailed MASE Results of Event Type Deep Analysis Using Gemini-2.0-Flash (cont'd, part 29/38)

| Method | robotics | rocket_launch | rubber_usd_cents_kg | salmon_nok_kg | silver_usd_t_oz |
|---|---|---|---|---|---|
| SeasonalNaive | 1.000 | 1.000 | 1.000 | 1.000 | 1.000 |
| Meta | 1.019 | 0.837 | 0.877 | 1.109 | 0.887 |
| Meta+Date | 0.961 | 1.051 | 1.008 | 1.174 | 0.846 |
| Meta+Date+Cov | 0.958 | 1.012 | 1.162 | 0.975 | 0.886 |
| Meta+Date+Event | 1.001 | 0.971 | 0.974 | 1.329 | 0.936 |
| AllContext | 0.962 | 1.030 | 0.806 | 1.326 | 0.818 |

*Table 124.* Detailed MASE Results of Event Type Deep Analysis Using Gemini-2.0-Flash (cont'd, part 30/38)

| Method | ski_gear | soybeans_usd_bu | space_exploration | steel_cny_t | stock_market |
|---|---|---|---|---|---|
| SeasonalNaive | 1.000 | 1.000 | 1.000 | 1.000 | 1.000 |
| Meta | 0.643 | 0.819 | 0.898 | 0.904 | 0.789 |
| Meta+Date | 0.215 | 1.183 | 0.407 | 0.926 | 0.749 |
| Meta+Date+Cov | 0.215 | 1.294 | 0.420 | 0.804 | 0.781 |
| Meta+Date+Event | 0.233 | 1.260 | 0.417 | 1.023 | 0.735 |
| AllContext | 0.225 | 0.908 | 0.418 | 0.866 | 0.758 |

*Table 125.* Detailed MASE Results of Event Type Deep Analysis Using Gemini-2.0-Flash (cont'd, part 31/38)

| Method | student_loans | sugar_usd_lbs | sustainable_fashion | tax_software | taxes |
|---|---|---|---|---|---|
| SeasonalNaive | 1.000 | 1.000 | 1.000 | 1.000 | 1.000 |
| Meta | 0.827 | 0.922 | 0.802 | 0.764 | 0.713 |
| Meta+Date | 0.792 | 0.939 | 0.658 | 0.368 | 0.189 |
| Meta+Date+Cov | 0.850 | 1.129 | 0.633 | 0.364 | 0.240 |
| Meta+Date+Event | 0.823 | 1.129 | 0.649 | 0.365 | 0.182 |
| AllContext | 0.844 | 0.946 | 0.648 | 0.379 | 0.191 |

*Table 126.* Detailed MASE Results of Event Type Deep Analysis Using Gemini-2.0-Flash (cont'd, part 32/38)

| Method | tellurium_cny_kg | tesla | tin_usd_t | titanium_cny_kg | tour_de_france |
|---|---|---|---|---|---|
| SeasonalNaive | 1.000 | 1.000 | 1.000 | 1.000 | 1.000 |
| Meta | 0.722 | 0.707 | 0.802 | 0.588 | 0.380 |
| Meta+Date | 0.687 | 0.776 | 0.933 | 0.618 | 0.287 |
| Meta+Date+Cov | 0.656 | 0.747 | 0.871 | 0.600 | 0.198 |
| Meta+Date+Event | 0.787 | 0.711 | 1.080 | 0.628 | 0.321 |
| AllContext | 0.711 | 0.743 | 0.740 | 0.604 | 0.285 |

*Table 127.* Detailed MASE Results of Event Type Deep Analysis Using Gemini-2.0-Flash (cont'd, part 33/38)

| Method | traffic_insurance | unemployment_rate | uranium_usd_lbs | urea_usd_t | usdtoaud_exchangerate |
|---|---|---|---|---|---|
| SeasonalNaive | 1.000 | 1.000 | 1.000 | 1.000 | 1.000 |
| Meta | 0.905 | 0.798 | 0.936 | 0.613 | 0.925 |
| Meta+Date | 0.964 | 0.533 | 0.944 | 0.496 | 0.827 |
| Meta+Date+Cov | 0.997 | 0.535 | 1.105 | 0.494 | 0.929 |
| Meta+Date+Event | 0.997 | 0.544 | 1.142 | 0.593 | 1.072 |
| AllContext | 1.048 | 0.529 | 0.869 | 0.507 | 0.807 |

*Table 128.* Detailed MASE Results of Event Type Deep Analysis Using Gemini-2.0-Flash (cont'd, part 34/38)

| Method | usdtobrl_exchangerate | usdtocad_exchangerate | usdtochf_exchangerate | usdtogbp_exchangerate | usdtohkd_exchangerate |
|---|---|---|---|---|---|
| SeasonalNaive | 1.000 | 1.000 | 1.000 | 1.000 | 1.000 |
| Meta | 0.795 | 0.716 | 0.978 | 0.766 | 0.810 |
| Meta+Date | 0.674 | 0.711 | 0.968 | 0.856 | 0.856 |
| Meta+Date+Cov | 0.822 | 0.915 | 1.139 | 0.897 | 0.985 |
| Meta+Date+Event | 1.099 | 0.795 | 0.890 | 0.964 | 0.970 |
| AllContext | 0.697 | 0.636 | 0.878 | 0.778 | 0.810 |

*Table 129.* Detailed MASE Results of Event Type Deep Analysis Using Gemini-2.0-Flash (cont'd, part 35/38)

| Method | usdtoinr_exchangerate | usdtokrw_exchangerate | usdtomxn_exchangerate | usdtosgd_exchangerate | used_car |
|---|---|---|---|---|---|
| SeasonalNaive | 1.000 | 1.000 | 1.000 | 1.000 | 1.000 |
| Meta | 0.783 | 0.927 | 0.775 | 1.038 | 0.935 |
| Meta+Date | 0.719 | 0.845 | 0.927 | 0.916 | 0.711 |
| Meta+Date+Cov | 0.705 | 0.911 | 0.915 | 1.017 | 0.682 |
| Meta+Date+Event | 0.734 | 0.920 | 0.927 | 1.041 | 0.673 |
| AllContext | 0.737 | 0.866 | 0.766 | 0.910 | 0.644 |

*Table 130.* Detailed MASE Results of Event Type Deep Analysis Using Gemini-2.0-Flash (cont'd, part 36/38)

| Method | vaccine_research | venture_capital | volcanic_eruption | water_scarcity | wheat_usd_bu |
|---|---|---|---|---|---|
| SeasonalNaive | 1.000 | 1.000 | 1.000 | 1.000 | 1.000 |
| Meta | 0.918 | 0.834 | 1.009 | 0.930 | 0.895 |
| Meta+Date | 0.810 | 0.822 | 0.661 | 0.441 | 1.023 |
| Meta+Date+Cov | 0.846 | 0.848 | 0.591 | 0.443 | 1.048 |
| Meta+Date+Event | 0.803 | 0.803 | 0.502 | 0.442 | 1.087 |
| AllContext | 0.838 | 0.823 | 0.642 | 0.437 | 0.966 |

*Table 131.* Detailed MASE Results of Event Type Deep Analysis Using Gemini-2.0-Flash (cont'd, part 37/38)

| Method | wildfires | wildlife_conservation | wool_aud_100kg | zinc_usd_t |
|---|---|---|---|---|
| SeasonalNaive | 1.000 | 1.000 | 1.000 | 1.000 |
| Meta | 0.630 | 0.897 | 0.755 | 0.779 |
| Meta+Date | 0.626 | 0.785 | 0.749 | 0.963 |
| Meta+Date+Cov | 0.649 | 0.807 | 0.770 | 0.967 |
| Meta+Date+Event | 0.626 | 0.799 | 0.728 | 0.927 |
| AllContext | 0.650 | 0.830 | 0.742 | 0.695 |

*Table 132.* Detailed MASE Results of Event Type Deep Analysis Using Gemini-2.0-Flash (cont'd, part 38/38)

| Method | affordable_housing | air_conditioner | air_pollution | air_travel | alphabet |
|---|---|---|---|---|---|
| SeasonalNaive | 1.000 | 1.000 | 1.000 | 1.000 | 1.000 |
| GPT-4o | 0.625 | 0.498 | 0.311 | 0.822 | 0.738 |
| GPT-5 | 0.500 | 0.448 | 0.321 | 0.864 | 0.677 |
| Gemini-2.0-Flash | 0.549 | 0.456 | 0.366 | 0.969 | 0.653 |
| Gemini-2.5-Flash | 0.583 | 0.328 | 0.356 | 0.932 | 0.635 |
| DeepSeek-V3 | 0.516 | 0.351 | 0.297 | 1.068 | 0.628 |
| DeepSeek-R1 | 0.603 | 0.616 | 0.440 | 0.699 | 0.886 |

*Table 133.* Detailed MASE Results of Reasoning Models with Data after 2025 Jan (part 1/38)

| Method | aluminum_usd_t | amazon | animal_migration | animal_rescue | animal_welfare |
|---|---|---|---|---|---|
| SeasonalNaive | 1.000 | 1.000 | 1.000 | 1.000 | 1.000 |
| GPT-4o | 0.727 | 0.887 | 0.777 | 0.807 | 0.666 |
| GPT-5 | 0.618 | 0.674 | 0.777 | 0.911 | 0.658 |
| Gemini-2.0-Flash | 1.164 | 0.408 | 0.844 | 0.773 | 0.711 |
| Gemini-2.5-Flash | 1.525 | 0.418 | 0.750 | 0.884 | 0.687 |
| DeepSeek-V3 | 0.997 | 1.191 | 0.840 | 0.914 | 0.688 |
| DeepSeek-R1 | 1.296 | 1.309 | 0.751 | 1.231 | 0.767 |

*Table 134.* Detailed MASE Results of Reasoning Models with Data after 2025 Jan (cont'd, part 2/38)

| Method | apple_inc | art_exhibitions | artificial_intelligence | asset_management | autonomous_driving |
|---|---|---|---|---|---|
| SeasonalNaive | 1.000 | 1.000 | 1.000 | 1.000 | 1.000 |
| GPT-4o | 0.775 | 0.728 | 0.992 | 0.744 | 0.862 |
| GPT-5 | 0.712 | 0.794 | 0.823 | 0.755 | 1.055 |
| Gemini-2.0-Flash | 0.750 | 0.922 | 0.965 | 0.841 | 1.171 |
| Gemini-2.5-Flash | 0.722 | 1.529 | 1.082 | 0.663 | 1.148 |
| DeepSeek-V3 | 0.824 | 0.809 | 1.056 | 0.840 | 0.946 |
| DeepSeek-R1 | 0.863 | 1.181 | 0.942 | 0.745 | 1.099 |

*Table 135.* Detailed MASE Results of Reasoning Models with Data after 2025 Jan (cont'd, part 3/38)

| Method | back_to_school | barley_inr_t | beef_brl_kg | beekeeping | biodiversity |
|---|---|---|---|---|---|
| SeasonalNaive | 1.000 | 1.000 | 1.000 | 1.000 | 1.000 |
| GPT-4o | 0.353 | 0.659 | 0.646 | 0.439 | 0.297 |
| GPT-5 | 0.475 | 0.877 | 0.706 | 0.477 | 0.396 |
| Gemini-2.0-Flash | 0.445 | 0.741 | 0.693 | 0.465 | 0.423 |
| Gemini-2.5-Flash | 0.492 | 0.780 | 0.934 | 1.660 | 0.414 |
| DeepSeek-V3 | 0.406 | 0.836 | 0.751 | 0.395 | 0.397 |
| DeepSeek-R1 | 0.655 | 0.922 | 0.945 | 0.420 | 0.522 |

*Table 136.* Detailed MASE Results of Reasoning Models with Data after 2025 Jan (cont'd, part 4/38)

| Method | bitumen_cny_t | black_friday_deals | brent_usd_bbl | broadway_shows | butter_eur_t |
|---|---|---|---|---|---|
| SeasonalNaive | 1.000 | 1.000 | 1.000 | 1.000 | 1.000 |
| GPT-4o | 0.901 | 0.007 | 1.085 | 0.600 | 1.126 |
| GPT-5 | 0.898 | 0.004 | 0.966 | 0.666 | 1.149 |
| Gemini-2.0-Flash | 0.857 | 0.004 | 1.108 | 0.451 | 1.051 |
| Gemini-2.5-Flash | 1.304 | 0.004 | 1.016 | 1.635 | 1.135 |
| DeepSeek-V3 | 0.925 | 0.003 | 0.977 | 0.802 | 1.351 |
| DeepSeek-R1 | 1.465 | 0.002 | 1.094 | 1.391 | 1.204 |

*Table 137.* Detailed MASE Results of Reasoning Models with Data after 2025 Jan (cont'd, part 5/38)

| Method | cancer_research | canola_cad_t | carbon_emissions | cheese_usd_lbs | christmas_gifts |
|---|---|---|---|---|---|
| SeasonalNaive | 1.000 | 1.000 | 1.000 | 1.000 | 1.000 |
| GPT-4o | 0.710 | 0.789 | 0.702 | 0.727 | 0.063 |
| GPT-5 | 0.893 | 0.696 | 0.991 | 0.631 | 0.045 |
| Gemini-2.0-Flash | 0.739 | 0.758 | 0.778 | 0.712 | 0.037 |
| Gemini-2.5-Flash | 0.764 | 0.930 | 1.027 | 0.741 | 0.006 |
| DeepSeek-V3 | 0.749 | 0.634 | 0.680 | 0.692 | 0.063 |
| DeepSeek-R1 | 0.653 | 0.847 | 0.820 | 0.623 | 0.028 |

*Table 138.* Detailed MASE Results of Reasoning Models with Data after 2025 Jan (cont'd, part 6/38)

| Method | climate_change | coal_usd_t | cobalt_usd_t | cocoa_usd_t | coffee_usd_lbs |
|---|---|---|---|---|---|
| SeasonalNaive | 1.000 | 1.000 | 1.000 | 1.000 | 1.000 |
| GPT-4o | 6.545 | 0.931 | 1.378 | 0.983 | 1.173 |
| GPT-5 | 9.371 | 1.015 | 0.932 | 0.895 | 0.770 |
| Gemini-2.0-Flash | 6.639 | 1.105 | 1.007 | 1.029 | 1.220 |
| Gemini-2.5-Flash | 18.592 | 1.086 | 0.736 | 0.994 | 1.680 |
| DeepSeek-V3 | 9.381 | 0.762 | 0.783 | 0.994 | 1.116 |
| DeepSeek-R1 | 7.727 | 1.139 | 0.946 | 1.022 | 1.094 |

*Table 139.* Detailed MASE Results of Reasoning Models with Data after 2025 Jan (cont'd, part 7/38)

| Method | comic_con | consumer_electronics | copper_usd_lbs | corn_usd_bu | cost_of_living |
|---|---|---|---|---|---|
| SeasonalNaive | 1.000 | 1.000 | 1.000 | 1.000 | 1.000 |
| GPT-4o | 0.551 | 0.775 | 0.749 | 1.013 | 0.693 |
| GPT-5 | 0.468 | 0.675 | 0.784 | 0.941 | 0.536 |
| Gemini-2.0-Flash | 0.484 | 0.704 | 0.730 | 1.004 | 0.597 |
| Gemini-2.5-Flash | 0.762 | 0.887 | 0.876 | 1.070 | 0.636 |
| DeepSeek-V3 | 0.562 | 0.749 | 0.723 | 0.905 | 0.554 |
| DeepSeek-R1 | 0.745 | 1.144 | 0.966 | 1.206 | 0.847 |

*Table 140.* Detailed MASE Results of Reasoning Models with Data after 2025 Jan (cont'd, part 8/38)

| Method | cotton_usd_lbs | cryptocurrency | cybersecurity | data_breach | data_privacy |
|---|---|---|---|---|---|
| SeasonalNaive | 1.000 | 1.000 | 1.000 | 1.000 | 1.000 |
| GPT-4o | 1.049 | 0.775 | 0.748 | 1.299 | 0.721 |
| GPT-5 | 1.177 | 0.614 | 0.718 | 1.114 | 0.713 |
| Gemini-2.0-Flash | 1.216 | 0.563 | 0.854 | 1.058 | 0.714 |
| Gemini-2.5-Flash | 1.467 | 1.630 | 0.642 | 2.352 | 0.719 |
| DeepSeek-V3 | 1.141 | 0.697 | 0.891 | 1.207 | 0.925 |
| DeepSeek-R1 | 1.469 | 1.314 | 0.792 | 1.434 | 0.721 |

*Table 141.* Detailed MASE Results of Reasoning Models with Data after 2025 Jan (cont'd, part 9/38)

| Method | deforestation | diabetes | diammonium_usd_t | domestic_violence | drones |
|---|---|---|---|---|---|
| SeasonalNaive | 1.000 | 1.000 | 1.000 | 1.000 | 1.000 |
| GPT-4o | 0.327 | 0.472 | 0.483 | 0.758 | 0.619 |
| GPT-5 | 0.262 | 0.382 | 0.512 | 0.764 | 0.668 |
| Gemini-2.0-Flash | 0.364 | 0.485 | 0.950 | 0.860 | 0.568 |
| Gemini-2.5-Flash | 0.428 | 0.564 | 0.340 | 0.601 | 0.811 |
| DeepSeek-V3 | 0.465 | 0.550 | 0.382 | 0.584 | 0.515 |
| DeepSeek-R1 | 0.527 | 0.409 | 0.372 | 0.515 | 0.740 |

*Table 142.* Detailed MASE Results of Reasoning Models with Data after 2025 Jan (cont'd, part 10/38)

| Method | drought | drug_overdose | earthquake | electric_vehicle | endangered_species |
|---|---|---|---|---|---|
| SeasonalNaive | 1.000 | 1.000 | 1.000 | 1.000 | 1.000 |
| GPT-4o | 0.400 | 0.462 | 0.888 | 0.686 | 0.313 |
| GPT-5 | 0.393 | 0.473 | 0.776 | 0.691 | 0.270 |
| Gemini-2.0-Flash | 0.513 | 0.442 | 0.816 | 0.649 | 0.326 |
| Gemini-2.5-Flash | 1.187 | 0.455 | 0.969 | 0.733 | 0.462 |
| DeepSeek-V3 | 0.708 | 0.491 | 0.977 | 0.943 | 0.355 |
| DeepSeek-R1 | 1.225 | 0.797 | 0.884 | 0.990 | 0.618 |

*Table 143.* Detailed MASE Results of Reasoning Models with Data after 2025 Jan (cont'd, part 11/38)

| Method | esports | ethanol_usd_gal | fashion_week | federal_budget_deficit | federal_reserve |
|---|---|---|---|---|---|
| SeasonalNaive | 1.000 | 1.000 | 1.000 | 1.000 | 1.000 |
| GPT-4o | 0.712 | 1.081 | 0.373 | 0.483 | 0.555 |
| GPT-5 | 0.781 | 1.019 | 0.304 | 0.579 | 0.475 |
| Gemini-2.0-Flash | 0.418 | 0.982 | 0.298 | 0.529 | 0.445 |
| Gemini-2.5-Flash | 1.607 | 1.017 | 0.367 | 0.569 | 0.599 |
| DeepSeek-V3 | 0.667 | 1.248 | 0.411 | 0.537 | 0.519 |
| DeepSeek-R1 | 1.164 | 1.065 | 0.411 | 0.677 | 0.678 |

*Table 144.* Detailed MASE Results of Reasoning Models with Data after 2025 Jan (cont'd, part 12/38)

| Method | film_festivals | financial_regulation | flooding | flu_shot | food_recall |
|---|---|---|---|---|---|
| SeasonalNaive | 1.000 | 1.000 | 1.000 | 1.000 | 1.000 |
| GPT-4o | 0.694 | 0.436 | 1.156 | 0.379 | 1.053 |
| GPT-5 | 0.744 | 0.457 | 1.016 | 0.424 | 0.746 |
| Gemini-2.0-Flash | 0.720 | 0.576 | 1.060 | 0.378 | 0.731 |
| Gemini-2.5-Flash | 0.797 | 0.546 | 1.439 | 0.436 | 1.242 |
| DeepSeek-V3 | 0.600 | 0.806 | 1.094 | 0.324 | 1.073 |
| DeepSeek-R1 | 0.646 | 0.637 | 1.379 | 0.436 | 1.019 |

*Table 145.* Detailed MASE Results of Reasoning Models with Data after 2025 Jan (cont'd, part 13/38)

| Method | food_safety | food_wine_festivals | formula_1 | gallium_cny_kg | gas_prices |
|---|---|---|---|---|---|
| SeasonalNaive | 1.000 | 1.000 | 1.000 | 1.000 | 1.000 |
| GPT-4o | 0.648 | 0.768 | 0.513 | 0.876 | 0.830 |
| GPT-5 | 0.677 | 0.772 | 0.465 | 0.902 | 0.982 |
| Gemini-2.0-Flash | 0.770 | 1.026 | 0.531 | 2.649 | 0.795 |
| Gemini-2.5-Flash | 0.732 | 0.775 | 0.586 | 0.932 | 1.579 |
| DeepSeek-V3 | 0.749 | 0.836 | 0.465 | 1.398 | 0.725 |
| DeepSeek-R1 | 0.499 | 0.808 | 0.635 | 3.772 | 1.169 |

*Table 146.* Detailed MASE Results of Reasoning Models with Data after 2025 Jan (cont'd, part 14/38)

| Method | gasoline_usd_gal | gender_equality | gene_editing | germanium_cny_kg | global_warming |
|---|---|---|---|---|---|
| SeasonalNaive | 1.000 | 1.000 | 1.000 | 1.000 | 1.000 |
| GPT-4o | 0.672 | 0.300 | 0.708 | 0.964 | 0.792 |
| GPT-5 | 0.596 | 0.242 | 0.696 | 0.910 | 0.795 |
| Gemini-2.0-Flash | 0.808 | 0.351 | 0.798 | 1.290 | 0.614 |
| Gemini-2.5-Flash | 0.803 | 0.501 | 1.021 | 3.568 | 0.981 |
| DeepSeek-V3 | 0.677 | 0.587 | 0.884 | 1.534 | 0.963 |
| DeepSeek-R1 | 1.046 | 0.531 | 1.249 | 4.826 | 1.196 |

*Table 147.* Detailed MASE Results of Reasoning Models with Data after 2025 Jan (cont'd, part 15/38)

| Method | gold_usd_t_oz | goldman_sachs | government_spending | healthcare_costs | healthcare_policy |
|---|---|---|---|---|---|
| SeasonalNaive | 1.000 | 1.000 | 1.000 | 1.000 | 1.000 |
| GPT-4o | 1.181 | 0.717 | 0.586 | 0.557 | 0.604 |
| GPT-5 | 0.872 | 0.663 | 0.515 | 0.537 | 0.542 |
| Gemini-2.0-Flash | 1.156 | 0.687 | 0.551 | 0.591 | 0.544 |
| Gemini-2.5-Flash | 1.399 | 0.749 | 0.454 | 0.669 | 0.828 |
| DeepSeek-V3 | 1.077 | 0.672 | 0.591 | 0.670 | 0.704 |
| DeepSeek-R1 | 1.668 | 0.862 | 0.561 | 0.726 | 0.558 |

*Table 148.* Detailed MASE Results of Reasoning Models with Data after 2025 Jan (cont'd, part 16/38)

| Method | heatwave | heavy_rainfall | hedge_funds | hiv_aids | homelessness |
|---|---|---|---|---|---|
| SeasonalNaive | 1.000 | 1.000 | 1.000 | 1.000 | 1.000 |
| GPT-4o | 0.895 | 0.494 | 0.668 | 0.656 | 0.772 |
| GPT-5 | 0.900 | 0.563 | 0.583 | 0.694 | 0.844 |
| Gemini-2.0-Flash | 0.782 | 0.637 | 0.782 | 0.814 | 0.936 |
| Gemini-2.5-Flash | 2.677 | 0.678 | 0.646 | 0.784 | 1.657 |
| DeepSeek-V3 | 0.934 | 0.526 | 0.734 | 0.914 | 0.991 |
| DeepSeek-R1 | 2.180 | 0.456 | 0.689 | 0.776 | 1.210 |

*Table 149.* Detailed MASE Results of Reasoning Models with Data after 2025 Jan (cont'd, part 17/38)

| Method | human_rights | immigration_reform | income_inequality | indium_cny_kg | infectious_disease |
|---|---|---|---|---|---|
| SeasonalNaive | 1.000 | 1.000 | 1.000 | 1.000 | 1.000 |
| GPT-4o | 0.472 | 0.857 | 0.369 | 0.641 | 0.452 |
| GPT-5 | 0.453 | 0.903 | 0.404 | 0.551 | 0.450 |
| Gemini-2.0-Flash | 0.482 | 0.673 | 0.359 | 0.790 | 0.504 |
| Gemini-2.5-Flash | 0.464 | 1.558 | 0.442 | 0.882 | 0.502 |
| DeepSeek-V3 | 0.678 | 0.947 | 0.493 | 0.771 | 0.602 |
| DeepSeek-R1 | 0.567 | 0.812 | 0.414 | 1.200 | 0.780 |

*Table 150.* Detailed MASE Results of Reasoning Models with Data after 2025 Jan (cont'd, part 18/38)

| Method | inflation | infrastructure_spending | international_trade | invasive_species | investment_banking |
|---|---|---|---|---|---|
| SeasonalNaive | 1.000 | 1.000 | 1.000 | 1.000 | 1.000 |
| GPT-4o | 0.640 | 0.428 | 0.456 | 0.266 | 0.647 |
| GPT-5 | 0.732 | 0.614 | 0.406 | 0.268 | 0.453 |
| Gemini-2.0-Flash | 0.566 | 0.693 | 0.478 | 0.286 | 0.640 |
| Gemini-2.5-Flash | 0.613 | 0.847 | 1.113 | 0.546 | 0.384 |
| DeepSeek-V3 | 0.824 | 0.579 | 0.596 | 0.385 | 0.759 |
| DeepSeek-R1 | 0.860 | 0.780 | 1.036 | 0.451 | 0.616 |

*Table 151.* Detailed MASE Results of Reasoning Models with Data after 2025 Jan (cont'd, part 19/38)

| Method | lead_usd_t | lithium_cny_t | lumber_usd_1000_board_feet | magnesium_cny_t | major_league_baseball |
|---|---|---|---|---|---|
| SeasonalNaive | 1.000 | 1.000 | 1.000 | 1.000 | 1.000 |
| GPT-4o | 0.766 | 0.238 | 0.979 | 0.810 | 0.223 |
| GPT-5 | 0.735 | 0.372 | 0.997 | 0.806 | 0.222 |
| Gemini-2.0-Flash | 0.705 | 0.221 | 0.891 | 0.967 | 0.228 |
| Gemini-2.5-Flash | 0.911 | 0.223 | 1.539 | 0.819 | 0.347 |
| DeepSeek-V3 | 1.024 | 0.372 | 0.983 | 0.776 | 0.264 |
| DeepSeek-R1 | 1.054 | 0.257 | 1.298 | 0.701 | 0.580 |

*Table 152.* Detailed MASE Results of Reasoning Models with Data after 2025 Jan (cont'd, part 20/38)

| Method | manganese_cny_mtu | marine_life | marine_pollution | mental_health | meta_platforms |
|---|---|---|---|---|---|
| SeasonalNaive | 1.000 | 1.000 | 1.000 | 1.000 | 1.000 |
| GPT-4o | 0.832 | 0.521 | 0.303 | 0.443 | 0.730 |
| GPT-5 | 0.724 | 0.541 | 0.300 | 0.380 | 0.670 |
| Gemini-2.0-Flash | 0.718 | 0.585 | 0.374 | 0.441 | 0.656 |
| Gemini-2.5-Flash | 1.109 | 0.383 | 0.366 | 0.340 | 0.866 |
| DeepSeek-V3 | 0.782 | 0.626 | 0.401 | 0.618 | 0.691 |
| DeepSeek-R1 | 0.718 | 0.486 | 0.490 | 0.517 | 0.928 |

*Table 153.* Detailed MASE Results of Reasoning Models with Data after 2025 Jan (cont'd, part 21/38)

| Method | meteor_shower | methanol_cny_t | microsoft | milk_usd_cwt | minimum_wage |
|---|---|---|---|---|---|
| SeasonalNaive | 1.000 | 1.000 | 1.000 | 1.000 | 1.000 |
| GPT-4o | 0.665 | 0.809 | 0.721 | 0.903 | 1.096 |
| GPT-5 | 0.275 | 0.842 | 0.595 | 0.958 | 0.918 |
| Gemini-2.0-Flash | 0.311 | 0.812 | 0.709 | 0.857 | 1.009 |
| Gemini-2.5-Flash | 0.573 | 0.702 | 1.163 | 1.067 | 1.308 |
| DeepSeek-V3 | 0.363 | 0.922 | 0.872 | 0.848 | 0.882 |
| DeepSeek-R1 | 0.979 | 0.623 | 1.069 | 0.926 | 0.958 |

*Table 154.* Detailed MASE Results of Reasoning Models with Data after 2025 Jan (cont'd, part 22/38)

| Method | molybdenum_cny_kg | mortgage_rates | music_festivals | naphtha_usd_t | national_basketball_association |
|---|---|---|---|---|---|
| SeasonalNaive | 1.000 | 1.000 | 1.000 | 1.000 | 1.000 |
| GPT-4o | 0.794 | 0.770 | 0.349 | 1.235 | 0.587 |
| GPT-5 | 0.803 | 0.740 | 0.305 | 0.942 | 0.681 |
| Gemini-2.0-Flash | 0.747 | 0.839 | 0.331 | 1.142 | 0.629 |
| Gemini-2.5-Flash | 0.850 | 0.827 | 0.738 | 1.110 | 0.729 |
| DeepSeek-V3 | 1.007 | 0.781 | 0.443 | 1.027 | 0.503 |
| DeepSeek-R1 | 1.003 | 0.920 | 0.557 | 1.154 | 0.641 |

*Table 155.* Detailed MASE Results of Reasoning Models with Data after 2025 Jan (cont'd, part 23/38)

| Method | national_debt | national_football_league | neodymium_cny_t | nickel_usd_t | nobel_prize |
|---|---|---|---|---|---|
| SeasonalNaive | 1.000 | 1.000 | 1.000 | 1.000 | 1.000 |
| GPT-4o | 0.737 | 0.615 | 0.942 | 0.913 | 0.566 |
| GPT-5 | 0.610 | 0.586 | 0.956 | 0.763 | 0.385 |
| Gemini-2.0-Flash | 0.663 | 0.542 | 0.908 | 0.980 | 0.511 |
| Gemini-2.5-Flash | 0.620 | 0.829 | 1.080 | 0.866 | 0.526 |
| DeepSeek-V3 | 0.736 | 0.724 | 1.041 | 0.736 | 0.507 |
| DeepSeek-R1 | 0.697 | 1.121 | 0.963 | 0.672 | 0.729 |

*Table 156.* Detailed MASE Results of Reasoning Models with Data after 2025 Jan (cont'd, part 24/38)

| Method | nvidia | oat_usd_bu | obesity | opioid_crisis | organic_food |
|---|---|---|---|---|---|
| SeasonalNaive | 1.000 | 1.000 | 1.000 | 1.000 | 1.000 |
| GPT-4o | 1.315 | 0.944 | 0.608 | 0.729 | 0.407 |
| GPT-5 | 1.130 | 0.872 | 0.422 | 0.383 | 0.317 |
| Gemini-2.0-Flash | 0.826 | 0.894 | 0.669 | 0.449 | 0.426 |
| Gemini-2.5-Flash | 1.692 | 1.033 | 1.499 | 0.387 | 0.369 |
| DeepSeek-V3 | 1.678 | 1.108 | 0.768 | 0.597 | 0.479 |
| DeepSeek-R1 | 1.663 | 1.187 | 0.886 | 0.641 | 0.722 |

*Table 157.* Detailed MASE Results of Reasoning Models with Data after 2025 Jan (cont'd, part 25/38)

| Method | palladium_usd_t_oz | pest_control | pet_adoption | pet_health | platinum_usd_t_oz |
|---|---|---|---|---|---|
| SeasonalNaive | 1.000 | 1.000 | 1.000 | 1.000 | 1.000 |
| GPT-4o | 0.838 | 0.362 | 1.028 | 0.747 | 0.556 |
| GPT-5 | 1.021 | 0.311 | 0.881 | 0.836 | 0.572 |
| Gemini-2.0-Flash | 1.002 | 0.436 | 0.809 | 0.933 | 0.493 |
| Gemini-2.5-Flash | 1.080 | 0.259 | 1.345 | 0.813 | 0.717 |
| DeepSeek-V3 | 1.036 | 0.394 | 0.994 | 0.883 | 0.611 |
| DeepSeek-R1 | 1.250 | 0.250 | 1.848 | 0.857 | 0.717 |

*Table 158.* Detailed MASE Results of Reasoning Models with Data after 2025 Jan (cont'd, part 26/38)

| Method | polyethylene_cny_t | polypropylene_cny_t | polyvinyl_cny_t | potatoes_eur_100kg | poultry_brl_kgs |
|---|---|---|---|---|---|
| SeasonalNaive | 1.000 | 1.000 | 1.000 | 1.000 | 1.000 |
| GPT-4o | 0.711 | 0.718 | 0.878 | 0.706 | 0.654 |
| GPT-5 | 0.776 | 0.610 | 0.954 | 0.730 | 0.399 |
| Gemini-2.0-Flash | 0.724 | 0.711 | 0.833 | 0.738 | 0.565 |
| Gemini-2.5-Flash | 0.818 | 0.765 | 0.974 | 0.760 | 0.447 |
| DeepSeek-V3 | 0.759 | 0.863 | 0.993 | 0.757 | 0.923 |
| DeepSeek-R1 | 0.687 | 0.798 | 0.943 | 1.034 | 0.534 |

*Table 159.* Detailed MASE Results of Reasoning Models with Data after 2025 Jan (cont'd, part 27/38)

| Method | presidential_election | private_equity | propane_usd_gal | protest | quantum_computing |
|---|---|---|---|---|---|
| SeasonalNaive | 1.000 | 1.000 | 1.000 | 1.000 | 1.000 |
| GPT-4o | 0.952 | 0.619 | 1.246 | 0.974 | 0.702 |
| GPT-5 | 0.407 | 0.691 | 0.869 | 0.989 | 0.902 |
| Gemini-2.0-Flash | 0.522 | 0.818 | 1.140 | 1.043 | 0.843 |
| Gemini-2.5-Flash | 0.387 | 0.671 | 0.794 | 0.999 | 0.800 |
| DeepSeek-V3 | 0.406 | 0.817 | 1.330 | 1.017 | 0.750 |
| DeepSeek-R1 | 0.384 | 0.823 | 1.035 | 0.994 | 0.197 |

*Table 160.* Detailed MASE Results of Reasoning Models with Data after 2025 Jan (cont'd, part 28/38)

| Method | rapeseed_eur_t | refugee_support | renewable_energy | rhodium_usd_t_oz | rice_usd_cwt |
|---|---|---|---|---|---|
| SeasonalNaive | 1.000 | 1.000 | 1.000 | 1.000 | 1.000 |
| GPT-4o | 0.807 | 0.576 | 0.365 | 0.875 | 1.084 |
| GPT-5 | 0.761 | 0.599 | 0.368 | 0.570 | 1.061 |
| Gemini-2.0-Flash | 0.858 | 0.549 | 0.476 | 0.776 | 1.482 |
| Gemini-2.5-Flash | 0.817 | 0.667 | 0.554 | 0.535 | 1.070 |
| DeepSeek-V3 | 0.808 | 0.689 | 0.499 | 1.087 | 1.139 |
| DeepSeek-R1 | 0.900 | 0.751 | 0.537 | 0.630 | 1.474 |

*Table 161.* Detailed MASE Results of Reasoning Models with Data after 2025 Jan (cont'd, part 29/38)

| Method | robotics | rocket_launch | rubber_usd_cents_kg | silver_usd_t_oz | ski_gear |
|---|---|---|---|---|---|
| SeasonalNaive | 1.000 | 1.000 | 1.000 | 1.000 | 1.000 |
| GPT-4o | 1.049 | 0.870 | 1.063 | 1.049 | 0.205 |
| GPT-5 | 0.893 | 0.769 | 0.445 | 1.023 | 0.216 |
| Gemini-2.0-Flash | 1.058 | 0.889 | 1.012 | 1.043 | 0.150 |
| Gemini-2.5-Flash | 1.093 | 0.711 | 0.965 | 1.297 | 0.150 |
| DeepSeek-V3 | 1.110 | 0.815 | 1.017 | 1.035 | 0.248 |
| DeepSeek-R1 | 1.157 | 0.680 | 1.126 | 1.426 | 0.218 |

*Table 162.* Detailed MASE Results of Reasoning Models with Data after 2025 Jan (cont'd, part 30/38)

| Method | soybeans_usd_bu | space_exploration | steel_cny_t | stock_market | student_loans |
|---|---|---|---|---|---|
| SeasonalNaive | 1.000 | 1.000 | 1.000 | 1.000 | 1.000 |
| GPT-4o | 1.198 | 0.433 | 0.883 | 0.925 | 0.879 |
| GPT-5 | 1.121 | 0.440 | 0.898 | 1.041 | 0.759 |
| Gemini-2.0-Flash | 1.569 | 0.393 | 1.097 | 1.027 | 0.872 |
| Gemini-2.5-Flash | 1.470 | 0.334 | 0.966 | 0.848 | 0.897 |
| DeepSeek-V3 | 1.195 | 0.612 | 1.005 | 0.910 | 0.909 |
| DeepSeek-R1 | 1.610 | 0.434 | 1.592 | 0.779 | 0.820 |

*Table 163.* Detailed MASE Results of Reasoning Models with Data after 2025 Jan (cont'd, part 31/38)

| Method | sugar_usd_lbs | sustainable_fashion | tax_software | taxes | tellurium_cny_kg |
|---|---|---|---|---|---|
| SeasonalNaive | 1.000 | 1.000 | 1.000 | 1.000 | 1.000 |
| GPT-4o | 0.613 | 0.514 | 0.290 | 0.168 | 1.085 |
| GPT-5 | 0.640 | 0.549 | 0.324 | 0.168 | 0.975 |
| Gemini-2.0-Flash | 0.722 | 0.581 | 0.327 | 0.170 | 0.893 |
| Gemini-2.5-Flash | 0.603 | 0.635 | 0.347 | 0.170 | 1.240 |
| DeepSeek-V3 | 0.933 | 0.632 | 0.362 | 0.628 | 1.343 |
| DeepSeek-R1 | 0.682 | 0.732 | 0.359 | 0.346 | 2.312 |

*Table 164.* Detailed MASE Results of Reasoning Models with Data after 2025 Jan (cont'd, part 32/38)

| Method | tesla | tin_usd_t | titanium_cny_kg | tour_de_france | traffic_insurance |
|---|---|---|---|---|---|
| SeasonalNaive | 1.000 | 1.000 | 1.000 | 1.000 | 1.000 |
| GPT-4o | 0.706 | 0.665 | 1.184 | 1.729 | 0.777 |
| GPT-5 | 0.642 | 0.504 | 0.520 | 0.949 | 0.757 |
| Gemini-2.0-Flash | 0.751 | 0.697 | 0.831 | 1.438 | 1.079 |
| Gemini-2.5-Flash | 0.627 | 0.501 | 1.094 | 0.891 | 0.758 |
| DeepSeek-V3 | 0.738 | 0.576 | 0.884 | 1.483 | 0.850 |
| DeepSeek-R1 | 0.700 | 0.601 | 0.852 | 2.873 | 0.700 |

*Table 165.* Detailed MASE Results of Reasoning Models with Data after 2025 Jan (cont'd, part 33/38)

| Method | unemployment_rate | uranium_usd_lbs | urea_usd_t | usdtoaud_exchangerate | usdtobrl_exchangerate |
|---|---|---|---|---|---|
| SeasonalNaive | 1.000 | 1.000 | 1.000 | 1.000 | 1.000 |
| GPT-4o | 0.572 | 0.723 | 0.492 | 1.076 | 0.658 |
| GPT-5 | 0.356 | 0.904 | 0.393 | 1.116 | 0.659 |
| Gemini-2.0-Flash | 0.406 | 0.807 | 0.495 | 1.084 | 0.691 |
| Gemini-2.5-Flash | 0.383 | 1.300 | 0.858 | 1.124 | 0.660 |
| DeepSeek-V3 | 0.838 | 0.904 | 0.497 | 1.161 | 0.663 |
| DeepSeek-R1 | 0.720 | 1.091 | 0.669 | 1.360 | 0.620 |

*Table 166.* Detailed MASE Results of Reasoning Models with Data after 2025 Jan (cont'd, part 34/38)

| Method | usdtocad_exchangerate | usdtochf_exchangerate | usdtogbp_exchangerate | usdtohkd_exchangerate | usdtoinr_exchangerate |
|---|---|---|---|---|---|
| SeasonalNaive | 1.000 | 1.000 | 1.000 | 1.000 | 1.000 |
| GPT-4o | 0.873 | 0.615 | 0.981 | 0.621 | 0.601 |
| GPT-5 | 0.861 | 0.663 | 1.003 | 0.629 | 0.608 |
| Gemini-2.0-Flash | 0.868 | 0.643 | 1.012 | 0.632 | 0.626 |
| Gemini-2.5-Flash | 0.886 | 0.636 | 1.019 | 0.623 | 0.615 |
| DeepSeek-V3 | 0.906 | 0.633 | 1.041 | 0.595 | 0.570 |
| DeepSeek-R1 | 0.912 | 0.614 | 0.929 | 0.439 | 0.666 |

*Table 167.* Detailed MASE Results of Reasoning Models with Data after 2025 Jan (cont'd, part 35/38)

| Method | usdtokrw_exchangerate | usdtomxn_exchangerate | usdtosgd_exchangerate | used_car | vaccine_research |
|---|---|---|---|---|---|
| SeasonalNaive | 1.000 | 1.000 | 1.000 | 1.000 | 1.000 |
| GPT-4o | 1.175 | 0.939 | 1.057 | 0.545 | 0.715 |
| GPT-5 | 1.163 | 0.940 | 1.082 | 0.271 | 0.815 |
| Gemini-2.0-Flash | 1.152 | 0.956 | 1.063 | 0.475 | 0.942 |
| Gemini-2.5-Flash | 1.193 | 0.989 | 1.088 | 0.414 | 0.784 |
| DeepSeek-V3 | 1.201 | 0.996 | 1.117 | 0.565 | 0.876 |
| DeepSeek-R1 | 1.257 | 1.001 | 1.000 | 0.618 | 0.923 |

*Table 168.* Detailed MASE Results of Reasoning Models with Data after 2025 Jan (cont'd, part 36/38)

| Method | venture_capital | volcanic_eruption | water_scarcity | wheat_usd_bu | wildfires |
|---|---|---|---|---|---|
| SeasonalNaive | 1.000 | 1.000 | 1.000 | 1.000 | 1.000 |
| GPT-4o | 0.634 | 0.490 | 0.401 | 1.080 | 0.557 |
| GPT-5 | 0.357 | 0.519 | 0.469 | 1.195 | 0.568 |
| Gemini-2.0-Flash | 0.835 | 0.683 | 0.522 | 1.106 | 0.546 |
| Gemini-2.5-Flash | 0.759 | 0.921 | 0.439 | 1.362 | 0.989 |
| DeepSeek-V3 | 0.782 | 1.044 | 0.512 | 1.099 | 0.555 |
| DeepSeek-R1 | 0.874 | 0.827 | 0.515 | 1.525 | 0.834 |

*Table 169.* Detailed MASE Results of Reasoning Models with Data after 2025 Jan (cont'd, part 37/38)

| Method | wildlife_conservation | wool_aud_100kg | zinc_usd_t |
|---|---|---|---|
| SeasonalNaive | 1.000 | 1.000 | 1.000 |
| GPT-4o | 0.631 | 0.441 | 0.911 |
| GPT-5 | 0.686 | 0.468 | 0.803 |
| Gemini-2.0-Flash | 0.911 | 0.426 | 0.989 |
| Gemini-2.5-Flash | 0.481 | 0.484 | 1.214 |
| DeepSeek-V3 | 0.865 | 0.693 | 1.072 |
| DeepSeek-R1 | 0.514 | 0.846 | 1.158 |

*Table 170.* Detailed MASE Results of Reasoning Models with Data after 2025 Jan (cont'd, part 38/38)

| Method | A&E | C&E | Econ | E. Tech | Fin | P&A | Pub. H. | PPG | Sci | Shop | SSSG | Traf | Crops | Energy | Lvstk. | RMC | SAM | SHVM | Curr |
|---|---|---|---|---|---|---|---|---|---|---|---|---|---|---|---|---|---|---|---|
| SeasonalNaive | 1.000 | 1.000 | 1.000 | 1.000 | 1.000 | 1.000 | 1.000 | 1.000 | 1.000 | 1.000 | 1.000 | 1.000 | 1.000 | 1.000 | 1.000 | 1.000 | 1.000 | 1.000 | 1.000 |
| Sundial | 0.625 | 0.649 | 0.728 | 0.799 | 0.845 | 0.700 | 0.663 | 0.742 | 0.744 | 0.482 | 0.740 | 0.837 | 0.899 | 0.915 | 0.817 | 0.927 | 0.862 | 0.837 | 1.070 |
| Moirai2.0 | 0.657 | 0.656 | 0.694 | 0.785 | 0.780 | 0.658 | 0.698 | 0.728 | 0.749 | 0.502 | 0.699 | 0.798 | 0.800 | 0.778 | 0.759 | 0.797 | 0.681 | 0.726 | 0.828 |
| TimesFM2.5 | 0.563 | 0.497 | 0.548 | 0.698 | 0.725 | 0.518 | 0.561 | 0.581 | 0.665 | 0.337 | 0.574 | 0.735 | 0.823 | 0.789 | 0.752 | 0.849 | 0.749 | 0.770 | 0.846 |
| "AvgEnsemble(TimesFM,Moirai)" | 0.595 | 0.553 | 0.600 | 0.731 | 0.744 | 0.570 | 0.601 | 0.633 | 0.694 | 0.396 | 0.619 | 0.752 | 0.803 | 0.771 | 0.743 | 0.812 | 0.689 | 0.731 | 0.830 |
| DeepSeek-V3 | 0.641 | 0.631 | 0.708 | 0.884 | 0.846 | 0.664 | 0.748 | 0.698 | 0.793 | 0.279 | 0.711 | 0.841 | 0.814 | 0.776 | 0.784 | 0.780 | 0.650 | 0.707 | 0.780 |
| Gemini-2.0-Flash | 0.575 | 0.499 | 0.555 | 0.702 | 0.777 | 0.589 | 0.616 | 0.632 | 0.720 | 0.260 | 0.642 | 0.797 | 0.811 | 0.779 | 0.788 | 0.781 | 0.664 | 0.709 | 0.784 |
| GPT-4o | 0.587 | 0.488 | 0.565 | 0.768 | 0.730 | 0.548 | 0.588 | 0.601 | 0.732 | 0.278 | 0.599 | 0.767 | 0.808 | 0.779 | 0.779 | 0.782 | 0.644 | 0.710 | 0.776 |
| "FuncRev(TimesFM,Gemini)" | 0.591 | 0.542 | 0.648 | 0.673 | 0.719 | 0.582 | 0.652 | 0.607 | 0.681 | 0.428 | 0.640 | 0.719 | 0.992 | 0.926 | 0.849 | 1.020 | 1.375 | 0.780 | 0.832 |
| "CodeRev(TimesFM,Gemini)" | 0.564 | 0.526 | 0.581 | 0.693 | 0.721 | 0.549 | 0.610 | 0.593 | 0.692 | 0.431 | 0.661 | 0.861 | 1.015 | 0.922 | 0.877 | 0.953 | 1.132 | 0.764 | 0.874 |
| "TextRev(TimesFM,Gemini)" | 0.564 | 0.499 | 0.547 | 0.686 | 0.728 | 0.529 | 0.552 | 0.591 | 0.672 | 0.294 | 0.580 | 0.727 | 0.901 | 0.832 | 0.822 | 0.871 | 0.955 | 0.729 | 0.775 |
| "AvgEnsemble(TimesFM,GPT)" | 0.564 | 0.477 | 0.523 | 0.670 | 0.729 | 0.528 | 0.563 | 0.584 | 0.663 | 0.273 | 0.584 | 0.736 | 0.804 | 0.765 | 0.754 | 0.801 | 0.694 | 0.728 | 0.806 |
| "AvgEnsemble(TimesFM,Gemini)" | 0.542 | 0.482 | 0.518 | 0.668 | 0.718 | 0.523 | 0.530 | 0.578 | 0.640 | 0.273 | 0.579 | 0.722 | 0.811 | 0.779 | 0.788 | 0.781 | 0.664 | 0.709 | 0.806 |

*Table 171.* Breakdown of MASE of all methods on each of the 19 domains. Acronyms are used to shorten domain names for clarity: see Appendix I.

