# OpenReview forum: "Rethinking Multimodal Time-Series Forecasting Evaluation"
_ICML.cc/2026/Conference — ICML 2026 regular_

### Official Review · Reviewer_JRTc · 2026-02-20

**Soundness:** 3
**Presentation:** 3
**Significance:** 3
**Originality:** 3
**Overall Recommendation:** 5
**Confidence:** 4

**Summary:**

This paper introduces TimesX, a novel multimodal time-series forecasting benchmark featuring large-scale, cross-domain real-world data and rich textual contexts to address existing limitations like poor generalization and data leakage. Through extensive empirical studies, the authors reveal that many forecasting methods performing well on prior benchmarks struggle under this new standard. Ultimately, the results demonstrate that simple ensemble methods capable of effectively leveraging rich textual background information outperform many strong baseline models in real-world scenarios.

**Compliance With Llm Reviewing Policy:**

Affirmed.

**Key Questions For Authors:**

Please refer to Weaknesses.

**Limitations:**

Please refer to Weaknesses.

**Strengths And Weaknesses:**

**Strengths**

- The motivation of this paper is highly clear and compelling, directly addressing the critical pain points of existing multimodal time-series forecasting benchmarks, such as poor generalization, limited textual contexts, and data leakage.

- The experimental evaluation is comprehensive and thorough. The authors conducted extensive empirical studies on various zero-shot multimodal forecasting approaches using their proposed large-scale dataset, TimesX.

- This research is highly meaningful and significantly advances the field. The introduction of the novel benchmark provides an invaluable new standard for evaluating time-series forecasting models in complex, real-world scenarios.

- The paper is exceptionally well-written, structurally clear, and logically rigorous, making it effortless for readers to comprehend the research background and core contributions.

**Weaknesses**

- Regarding dataset construction, incorporating finer temporal granularities, such as hourly data, would provide even greater practical value for highly time-sensitive applications like traffic flow forecasting.

- The current appendix is overly lengthy, which somewhat detracts from the overall reading experience. It is recommended to scale down the visualizations and logically reorganize the layout by prioritizing visual results, prompts, and detailed quantitative results. Adding a dedicated table of contents for the appendix would also greatly improve its navigability.

- The solutions proposed in Section 4.1 are highly insightful and serve as a valuable reference. Building on this, incorporating citations to relevant advanced works, such as ChronoSteer [1], could further enhance the completeness and academic depth of the paper.


**References**
[1] ChronoSteer: Bridging Large Language Model and Time Series Foundation Model via Synthetic Data, 2025

---

> ### Author Rebuttal · Authors · 2026-03-31
>
> We thank Reviewer JRTc for the positive assessment of our submission. We especially appreciate the reviewer’s recognition of the clear and compelling motivation, the comprehensive empirical evaluation, the significance of the proposed benchmark to the multimodal time-series forecasting community, and the overall writing quality and organization of the paper. We are also grateful for the reviewer’s thoughtful suggestions on finer temporal granularity, appendix organization, and related-work completeness.
>
> ### For W1: On adding hourly data such as traffic series
>
> We thank the reviewer for this insightful suggestion. Finer-grained data such as hourly traffic flow indeed have high practical value, especially for highly time-sensitive forecasting scenarios. The current data selection of TimesX is guided by **continuous refreshability** together with **strict temporal isolation** for leakage-aware evaluation. Under these constraints, high-frequency multimodal traffic data sources are currently quite limited. For example, widely used multimodal traffic datasets such as *MSPG (2021--2022), LEU (2012--2013), and PTF (2012) are not directly suitable** for the core TimesX benchmark because they are not designed for refreshability over time.
>
> We **newly incorporated** these high-frequency traffic datasets into the **TimesX finetuning split** to facilitate future research (*see the repo in Line 107, Page 2*). In the final version, we will discuss support for hourly data explicitly as an important direction for future versions.
>
> ### For W2: On improving the appendix
>
> We thank the reviewer for the concrete suggestion. We agree that the current appendix is lengthy, partly because a large-scale benchmark built from scratch, together with its extensive evaluation, requires substantial supporting information. In the final version, we will reorganize the appendix and further improve its layout and structure, including: appropriately reducing the size of some visualizations, reordering the content following the logic of visual results → prompts/details → detailed quantitative results, and adding a **clear appendix roadmap** to help readers navigate different parts more easily.
>
> ### For W3: On the missing related-work discussion
>
> We thank the reviewer for pointing out this important recent related work. **ChronoSteer** model adopts a revision-style paradigm via pretraining stage, which is **a further step** of the **TextRev / FuncRev / CodeRev** series evaluated in our work.
>
> We plan to include ChronoSteer in our evaluation as supported methods once this pretrained model is fully open-sourced. We will also add a discussion of this work in the final version.
>
> We again thank the reviewer for these specific and valuable suggestions.

---

> > ### Author Rebuttal · Reviewer_JRTc · 2026-04-03
> >
> > Thank you for the authors' response. I will maintain my previous positive score.

---

> > > ### Author Response · Authors · 2026-04-07
> > >
> > > We are glad our rebuttal addressed Reviewer JRTc’s concerns, and would like to thank Reviewer JRTc again for the positive assessment and insightful suggestions.

---

### Official Review · Reviewer_RL8E · 2026-03-06

**Soundness:** 2
**Presentation:** 2
**Significance:** 2
**Originality:** 3
**Overall Recommendation:** 4
**Confidence:** 3

**Summary:**

This paper introduces TimesX, a benchmark for multimodal time series forecasting. The benchmark extends traditional numerical time series by incorporating textual contextual information, such as event descriptions, calendar information, and metadata. In addition, the authors propose a data construction pipeline that enables real-time dataset generation, aiming to mitigate data leakage. The authors argue that existing time series forecasting benchmarks often rely on synthetic contextual information or datasets with limited scale, which may lead to an overestimation of the capabilities of large language model (LLM)-based forecasting methods. To address this issue, the paper proposes a new dataset construction framework and evaluates multiple forecasting models on the constructed benchmark.

**Compliance With Llm Reviewing Policy:**

Affirmed.

**Final Justification:**

This paper introduces TimesX, a multimodal time series forecasting benchmark that incorporates textual context. The work addresses an important and underexplored problem, and the scale and coverage of the dataset are commendable strengths.

The rebuttal partially addressed my concerns. The clarification on ChatTS evaluation and pipeline transparency is helpful. However, my concern on W1-2 remains: the reliance on synthetic data for core conclusions leaves the real-world generalizability of the findings an open question. For W3, while the authors explain that online access was disabled during evaluation, the systemic contamination risk for API-accessible models is not fully resolved.

Overall, I appreciate the authors' effort in the rebuttal. Given the partial resolution of my concerns and the value of the dataset contribution to the community, I have raised my score by one point, though concerns on data generalizability and contamination remain.

**Key Questions For Authors:**

1. On the suitability of CodeRev. CodeRev generates time series in the form of executable code, which is reasonable for generative tasks (e.g., CiK). However, for forecasting tasks, the representational capacity of code-generated sequences may be limited. Is it appropriate to directly apply such a mechanism to forecasting scenarios?
2. On the general effectiveness of holidays and covariates. The dataset includes holiday information and other covariates. Do these factors consistently contribute to forecasting performance across different domains? Are there domain-specific differences in their effectiveness?
3. On the reliability of the agent-based pipeline. The dataset is constructed using an automated agent-based pipeline. Is there a risk of hallucination in this process? Are there quality control mechanisms to ensure the factual correctness and consistency of the collected data?

**Limitations:**

Some core aspects of the paper may lack sufficient transparency, such as the specific roles of each agent in the pipeline, the rationale behind the selection of baselines, and the empirical validation of the effectiveness of the dataset’s contextual information.

**Strengths And Weaknesses:**

**Strengths**

1. Significance of the research direction. Multimodal time series forecasting is an important and promising research area, and there is currently a lack of well-established, high-quality datasets. This benchmark helps standardize evaluation protocols and may facilitate future research in this direction.
2. Scale and coverage. As a large-scale benchmark covering multiple domains, TimesX is relatively comprehensive and valuable. The dataset is also designed to mitigate data leakage, which is particularly important in the current research landscape.
3. Insightful findings. Through experiments on this benchmark, the paper reveals that the forecasting capabilities of current LLM-based methods may have been overestimated, providing meaningful insights to the community.

**Weakness**
1. Insufficient experimental validation.
  - In terms of baseline selection, the authors do not appear to consider methods that jointly model time series and contextual information, such as ChatTS.
  - Some claims are not sufficiently supported by empirical evidence. For example, the argument regarding the importance of real-world data lacks strong justification. The authors only compare three models on the CiK dataset, whose time series are relatively simple and code-generated. Therefore, the results may not convincingly demonstrate that real-world datasets are inherently more valuable or more challenging than synthetic ones.
2. Limited transparency in dataset construction. The paper proposes a multi-agent pipeline for dataset construction, but the specific roles of each agent are not clearly described. In particular, the procedures for collecting, filtering, and validating textual contextual information are insufficiently detailed, making it difficult to assess the reliability and quality of the generated context.
3. Concerns about data leakage. Although the authors claim that TimesX avoids data leakage, the dataset is collected via APIs. In principle, other agent-based approaches (e.g., models such as Gemini that can access similar APIs) may retrieve the same information to assist forecasting. Therefore, whether the dataset truly prevents data leakage remains debatable.

---

> ### Author Rebuttal · Authors · 2026-03-31
>
> We thank Reviewer RL8E for recognizing the research direction's significance, TimesX's scale and coverage and insightful findings.
> ### W1-1: Missing evaluation of ChatTS
>
> We thank the reviewer for suggesting TS-LLM models such as ChatTS.
>
> We would like to **clarify our evaluation considerations first**. This work focuses on zero-shot multimodal TS forecasting evaluation. Accordingly, we evaluate strong zero-shot TSFMs (4), LLMs (6), and their compositions (ensembles and agents). ChatTS is related, but its primary focus (TS understanding & reasoning) differs from our forecasting setting.
>
> We **newly evaluated ChatTS** (the latest Qwen3-8B version) on TimesX. Under the Tab. 4 setting, its aggregated MASE is 0.8933 (anonymous.4open.science/r/TimesX_UnderReview-387D/chatts_results.csv), improving over Seasonal Naive.
>
> We will **add discussion** in the final version. We also view these models, such as ChatTS, as *promising bases for future fine-tuning works*. We **further provided the ChatTS-ready version of TimesX** in the repo (Line 107, Page 2).
>
> ### W1-2: Concerns about the claim on "real-world data more valuable than syn data" supported based on only CiK
>
> We would first like to **clarify our claim**. As stated in **Remark 3.1 (The value of synthetic data)**: “We highly recommend using both synthetic and real-world benchmarks for a more robust evaluation.” **Our point is real-world evaluation remains necessary**.
>
> We **then explain why only use all 5 CiK subsets as the syn data**. Multimodal TSF is still early-stage, and to the best of our knowledge, CiK (ICML 2025) is the most representative peer-reviewed synthetic benchmark for multimodal forecasting; most others are designed mainly for training.
>
> Constructing high-fidelity synthetic multimodal TSF benchmarks **remains an important open problem**. We are happy to address further concerns.
>
> ### W2: Transparency in dataset construction
>
> We would like to clarify that **these details have been provided**: Sec. 3.2.2 and Fig. 2 describe the responsibilities of four agents and the overall workflow. App. E / Fig. 11 presents the detailed workflow with illustrations. App. F provides an execution log demo.
>
> We will present these details more centrally.
>
> ### W3: Concerns about test-time access to similar data APIs
>
> We thank the reviewer for this thoughtful concern. In our evaluation, we **disable online access by implementation**:
>
> - For LLMs: search, tools, and function calling are disabled, so plain LLMs cannot access APIs or the web during evaluation.
>
> - For agents such as FuncRev and CodeRev: all generated code runs in an OS-level no-network sandbox, so these methods also cannot access APIs or the web at runtime.
>
> We will clarify this explicitly.
>
> ### Q1: The suitability of CodeRev for forecasting
>
> We thank the reviewer. We **first clarify what CodeRev does**: it does *not generate forecasts from scratch, but revises an initial forecast using code*. Thus, CodeRev for forecasting is similar to CiK, which the reviewer considers reasonable. Moreover, **recent works suggest the potential** of coding for forecasting [1][2]. Besides, **our evaluation is broader than CodeRev** alone, covering TSFMs, LLMs, ensembles, TextRev, and FuncRev.
>
> We will clarify the motivation more explicitly.
>
> [1] Beyond Naïve Prompting: Strategies for Improved Zero-shot Context-aided Forecasting with LLMs
>
> [2] TS2Code: Enhancing Time Series Understanding via Learning to Code
>
> ### Q2: On the effectiveness of holidays and covariates across domains
>
> We have provided the ablations in **Tab. 5 (overall) and App. Y.3 (variable-level)**.
>
> **Further domain-level analysis** is:
>
> - For holiday. We compare Meta+Date with Meta and count a domain as effective if over half of its variables improve. Under this criterion, holidays are **effective in 12/19 domains**, with an average intra-domain win rate of 65.2%. The largest gains appear in Shopping, Climate & Environment, Public Health, and Public Policy & Governance, where calendar effects are stronger.
>
> - For covariates. Using the same criterion, covariates are **effective in 12/19 domains**, with an average intra-domain win rate of of 57.9%. The largest gains appear in Specialty & Advanced Materials, Raw Materials & Construction, Livestock & Food Products, and Currency, where exogenous drivers are likely more important.
>
> We will add these analysis.
>
> ### Q3: On the quality control
>
> We clarify this from both **mechanism design and evaluation evidence**: First, the multi-agent workflow has **multi-stage verification (adversarial) and iterative refinement (collaboration) (Sec. 3.2.2, Fig. 2, App. E/F)**. The Hypothesizer proposes candidates; the Verifier retrieves URLs, checks facts and dates, and identifies missing or unverified information; the Enricher fills unresolved details through strict time-bounded search; and the Synthesizer integrates verified evidence and **discards unverified events (~32.20%)**. We also provide a **manual audit in Sec. 3.2.4**.

---

> > ### Author Rebuttal · Reviewer_RL8E · 2026-04-01
> >
> > The rebuttal partially addresses my concerns. For W1-2, the authors clarify the rationale for using synthetic data, but the generalizability of the conclusions to real-world settings remains an open question. For W3, the explanation of disabling online access is helpful, but the systemic data contamination concern for models with access to similar APIs is not fully resolved. Overall, the rebuttal demonstrates reasonable effort, and I am willing to raise my score accordingly.

---

> > > ### Author Response · Authors · 2026-04-03
> > >
> > > We thank Reviewer RL8E for the positive follow-up. We are encouraged that many concerns, including W1-1, W2, Q1, Q2, and Q3, are resolved
> > >
> > > Please allow us to give **W1-2** and **W3** another shot:
> > >
> > > For **W1-2**, we understand the reviewer’s concern to be that "our point" that *"real-world data are more valuable for evaluation than synthetic data" may still be debatable*. We want to clarify that **we recommend using both synthetic and real-world data for evaluation, stated in Remark 3.1**, and **real-world evaluation data (such as our TimesX) remains necessary** (demonstrated in section 3.1.1). We also agree with the reviewer that how to construct high-fidelity synthetic evaluation data for forecasting, which can generalize well to real-world settings remains an open question. We will make sure to make this discussion more explicit in the final version.
> > >
> > > For **W3: test-time access to similar data APIs**, we thank the reviewer for this thoughtful concern. We would like to first **clarify the necessary conditions** for test-time leakage through API access:
> > >
> > > **Request: LLM can call APIs \& Response: the environment has web access**
> > >
> > >
> > > $\rightarrow$ **the LLM can access similar data APIs during evaluation.**
> > >
> > > In our evaluation, **this path is blocked by implementation**.
> > >
> > > - For **plain LLMs**, web searching and function calling are disabled by configs; that is, they are **not allowed to call APIs or even perform alternative web searches**.
> > >
> > > - For **agents**, all generated code is executed in a no-network sandbox, so **the execution environment has no web access**. In addition, as clarified in Appendix Q.2., the allowed function list does not include API-access functions.
> > >
> > >
> > > We further *manually checked the execution logs* and did not observe such test-time leakage.
> > >
> > > There is also no pretraining leakage: we evaluate on data collected after the knowledge cutoff of these LLMs (Section 3.1.2).
> > >
> > > We sincerely thank the reviewer again for the positive feedback, careful review, and insightful questions. We would be very happy to further address any concerns the reviewer may still have.

---

### Official Review · Reviewer_45TG · 2026-03-08

**Soundness:** 4
**Presentation:** 4
**Significance:** 3
**Originality:** 3
**Overall Recommendation:** 5
**Confidence:** 3

**Summary:**

This paper introduces TimesX, a real-world, large-scale, cross-domain multimodal time series forecasting benchmark with 190 variables from 19 domains, together with a data-generation pipeline designed to mitigate leakage and provide multiple forms of textual context, including metadata, calendar information, covariates, and time-stamped events. Overall, the submission's principal finding concerns the empirical observation that conclusions from earlier multimodal benchmarks can be misleading, and that on TimesX, simple ensemble methods combining TFMs and LLMs can outperform more complex agentic revision methods in the zero-shot setting.

**Compliance With Llm Reviewing Policy:**

Affirmed.

**Final Justification:**

I maintain my positive recommendation for this paper. The work addresses an important problem, and I found the overall contribution meaningful in terms of significance and practical relevance. The method and experimental evaluation are solid, and the paper provides enough evidence to support its main claims.
My main concerns from the initial review have been addressed in the authors’ response. I keep my original positive score 5.

**Key Questions For Authors:**

- **Can the authors summarize more explicitly under what conditions multimodal methods outperform unimodal methods, and when they do not?**
  Table 6 and the surrounding discussion suggest that the advantage is domain-dependent, for example in Shopping and some externally policy-driven domains. A clearer summary of these patterns would help clarify the practical scope of the benchmark’s conclusions. A strong answer would improve the paper’s significance.
- **How exactly are missing values handled in TimesX?**
  Since data quality is critical for a benchmark paper, the manuscript should clearly explain the missing-value treatment in the main text, including whether values are imputed, removed, forward-filled, or otherwise processed.
- **Why did the benchmark construction focus on heterogeneous univariate variables, rather than also including homogeneous multivariate forecasting settings?**
  It would be helpful to understand whether this was a deliberate design choice, a limitation of available sources, or a scope decision for the first release. A convincing explanation would help clarify the intended applicability of TimesX.
- **Can the authors explain the motivation behind the peak-based and event-peak-based search heuristics more clearly?**
  Since the Hypothesizer relies on points of interest such as local maxima, unexpected movements, or peaks in search trends, the paper should better justify why these are the right signals for event discovery and whether other alternatives were considered.

**Limitations:**

Yes

**Strengths And Weaknesses:**

**Strengths**

- The paper addresses an important and timely problem. A high-quality multimodal forecasting benchmark is clearly valuable for evaluating whether textual context genuinely helps time series forecasting, especially under zero-shot settings.
- The benchmark construction is ambitious and practically meaningful. TimesX is built from real-world data, spans 19 domains and 190 variables, and includes multiple types of aligned textual context rather than only one narrow form of auxiliary information.
- The paper makes a serious effort to address leakage, which is a central concern for pretrained LLMs and TFMs. The time-isolation principle, refreshable benchmark design, and verifier-style pipeline are all useful contributions for future benchmark design.
- The empirical study is broad and produces interesting findings. In particular, the paper shows that synthetic benchmarks may overestimate LLM advantages, that richer context improves performance, and that simple ensembles can outperform more complex revision-based approaches on this benchmark.

**Weaknesses**

- The collected data are essentially heterogeneous univariate time series paired with text context. It is not clear why homogeneous multivariate forecasting settings (for example, traffic flow on multiple sensors) were not included in the benchmark construction, since these are also important and common in time series forecasting. This limits how broadly the paper’s conclusions can be interpreted as applying to multimodal TSF in general.
- The treatment of missing values of time series is not described. Since benchmark quality is crucial here, the manuscript should explain much more explicitly how missing values are detected, imputed, filtered, or otherwise handled.
- The motivation for the two heuristic search principles used by the Hypothesizer, especially peak-based selection and event-related peaks, could be explained more clearly. The paper says the agent identifies points of interest such as local maxima, unexpected movements, or peaks in corresponding search trends, but the rationale for why these are the right anchors for event discovery is not fully articulated. This is especially important because these heuristics directly affect the generated event contexts.
- Although the paper presents domain-wise results, the manuscript does not yet summarize clearly enough when multimodal methods help more than unimodal ones, and under what kinds of domains or context conditions this advantage appears. Some discussion is present, for example for Shopping and several materials or currency-related domains, but this insight could be made much more systematic and central to the paper.

---

> ### Author Rebuttal · Authors · 2026-03-31
>
> We thank Reviewer 45TG for the positive assessment and practical suggestions of our submission.
>
> ### For W1 (Q3): Only heterogeneous univariate variables in the current release
>
> We thank the reviewer for the forward-looking suggestion. We agree that extending TimesX to multimodal multivariate settings is meaningful and represents an important direction for future releases.
>
> We would like to clarify why the **current release focuses** on multimodal univariate forecasting. **Following the development of unimodal zero-shot TSF,** univariate forecasting is often a natural first step before moving to multivariate settings. In particular, widely used foundation TSF benchmarks are still mainly univariate (e.g., GIFT-Eval), and **most foundation TSF models** (at the time of submission) **primarily support univariate forecasting**, such as TimesFM-2.5, Sundial, Chronos-Bolt, TiRex, and Toto.
>
> By contrast, multimodal TSF is still at an **earlier stage** of development. Therefore, in this work, we deliberately focus on multimodal univariate forecasting before extending to multivariate versions. In addition, we **have provided covariates as contexts** in the current version.
>
> We will add a discussion of multimodal (homogeneous) multivariate TSF as a future direction in the final version.
>
> ### For W2 (Q2): Missing-value handling in TimesX
>
> We thank the reviewer for this practical suggestion. Our numerical sources are official or high-quality (paid) APIs, which provide a **strong initial basis**. We **handle missing values separately for the input window and the target window, following common practices**, such as implementations of TimesFM and Chronos.
>
> - Target window (metric calculation): We follow common practice by masking missing targets, i.e., we compute evaluation metrics only on observed time indices.
> - Input window (model input): In our data, missingness is typically short. The average maximum contiguous **missing gap per sample is only 0.7962 (<1)**. We further remove anomalous samples (3) whose maximum gap exceeds 6. For the remaining data, we apply **linear interpolation**, which is suitable for short gaps.
>
> Notably, *this imputation is only required for time-series foundation models.* For LLM-based methods, we provide the observed series directly as (time index, value) pairs, so no imputation is needed.
>
> We will add an explicit description and detailed statistics of these missing-value handling steps in the final version.
>
> ### For W3 (Q4): Clearer explanation of the peak-based search heuristics
>
> We thank Reviewer 45TG for the suggestion to clarify the motivation behind these heuristic principles more clearly.
>
> We would like to clarify that:
>
> - (1) **Motivation Clarification**. We use peak-based anchors mainly to improve the **efficiency and cost-effectiveness** of dataset construction, i.e., to provide high-recall anchors for event discovery together with coverage-based early stopping (*see App. E, Lines 715--738; Cost analysis in Remark 3*). Concretely, under a fixed budget, the Hypothesizer prioritizes candidate events around salient change points in numeric series. Once the current candidate events already cover the search-trend peaks in time, the iteration can stop early. To reduce the noise and bias that may be introduced by this heuristic, the subsequent Verifier and Enricher further check facts and iteratively add missing details.
> - (2) **Supports for this motivation**. *Prior works* [1-3] have shown that Search Trend Peaks are often driven by real-world events, and can therefore serve as useful signals for event discovery.
> - (3) **Comparison with alternative strategies**. (a) We further conducted an ablation study (no peak heuristics; unlimited budget till no newly added events) on 10 variables. The peak-based heuristic **reduces dataset construction cost** by ~21%, while **maintaining** ～96% of the **final dataset quality** (accepted event numbers). (b) **Tab. 3** further shows that our construction strategy **leads to better forecasting performance** than a naive alternative.
>
> We will add discussion and analysis.
>
> [1] Predicting the Present with Google Trends.
>
> [2] The Use of Google Trends in Health Care Research: A Systematic Review
>
> [3] Spikes and Variance: Using Google Trends to Detect and Forecast Protests.
>
> ### For W4 (Q1): When multimodal methods help more than unimodal ones
>
> We thank the reviewer for this insightful suggestion.
>
> We have included a **more systematic analysis in App. V (Fig. 33--34),** where we show how the **win rate of multimodal methods over unimodal ones changes with data characteristics**:
>  - multimodal methods tend to show larger advantages when a dataset features **more provided events, richer event details, and stronger calendar-driven patterns**
> - when the series exhibits stronger endogenous temporal properties such as **trend, non-stationarity, or transitions**, unimodal methods tend to be more competitive.
>
> We will move this analysis to main body.

---

> > ### Author Rebuttal · Reviewer_45TG · 2026-04-01
> >
> > Thank you for the detailed response. My concerns have been addressed, and I will maintain my positive score.

---

> > > ### Author Response · Authors · 2026-04-07
> > >
> > > We are glad our rebuttal addressed Reviewer 45TG’s concerns, and would like to thank Reviewer 45TG again for the positive assessment and insightful suggestions.

---

### Official Review · Reviewer_3Tbv · 2026-03-12

**Soundness:** 3
**Presentation:** 3
**Significance:** 3
**Originality:** 3
**Overall Recommendation:** 4
**Confidence:** 2

**Summary:**

This paper introduces TimesX, a multimodal time series forecasting benchmark designed to address three limitations of existing benchmarks: (1) poor generalization due to small-scale or synthetic data, (2) limited types of textual contexts, and (3) inability to mitigate data leakage. TimesX contains 190 real-world variables across 19 domains with four types of textual context (metadata, calendar, covariates, and time-stamped events). A multi-agent pipeline (Hypothesizer-Verifier-Enricher-Synthesizer) is proposed to automatically construct and verify event contexts. The benchmark is designed to be refreshable over time to maintain leakage-free evaluation. Extensive zero-shot experiments on 13 methods reveal that LLMs' advantage over time series foundation models is smaller on real data than on synthetic benchmarks, and that simple ensemble methods (AVGENS) outperform more complex agentic revision approaches.

**Compliance With Llm Reviewing Policy:**

Affirmed.

**Key Questions For Authors:**

See Weakness.

**Limitations:**

Yes

**Strengths And Weaknesses:**

**Strengths**:
* S1. Important and timely contribution. The paper addresses a real and pressing need in the multimodal time series community. The critique of existing benchmarks is well-supported with controlled experiments (Fig. 1, Table 2, Table 3), each clearly demonstrating how synthetic data, data leakage, and low-quality context can distort evaluation conclusions.
* S2. Well-designed data construction pipeline. The multi-agent Hypothesizer-Verifier-Enricher-Synthesizer framework for event context construction is a thoughtful design that balances factual accuracy, temporal alignment, and scalability. The adversarial interaction between Hypothesizer and Verifier is a practical approach to reducing hallucination and leakage.
* S3. Strong experimental methodology. The evaluation is thorough: 13 methods, 312K+ LLM inferences, 10-seed repetition, per-domain breakdowns (Table 6), context-type ablations (Table 5), leakage analysis (Table 2), and cross-benchmark comparisons. The controlled experiment replacing Time-MMD's context with TimesX-generated context (Table 3) is particularly convincing.

**Weaknesses**:
* W1. Despite claiming large-scale and cross-domain coverage, the numerical time series come from only three sources: Google Search Trends (weekly), a commodity price API (daily), and a currency rate API (daily). This means the benchmark heavily skews toward search interest and financial data. Many important time series domains are absent including healthcare, energy, transportation, climate, manufacturing. Can the author provide a more comprehensive discussion on these and analyze the current limitations.

---

> ### Author Rebuttal · Authors · 2026-03-31
>
> We thank Reviewer 3Tbv for the careful reading and the positive assessment of our submission. We particularly appreciate the reviewer’s recognition of (i) the importance and timeliness of the problem, (ii) the design of our Hypothesizer–Verifier–Enricher–Synthesizer pipeline, and (iii) the thorough experimental study, including the controlled analyses on synthetic data, leakage, and context quality.
>
> ### For W1: Numerical Data Sources
>
> We would like to clarify that:
>
> (1) **This is a design trade-off in the current version.** As described in our design principles (Section 3), TimesX prioritizes automatic refreshability for strict temporal isolation, so that the benchmark can continuously mitigate information leakage over time. Many publicly available time series do not meet these requirements, particularly because they are updated irregularly /rarely or not api-accessible.
>
> (2) To examine **whether this source skew makes the current benchmark numerically biased**, we compared three representative unimodal time-series foundation models on TimesX and GIFT-Eval, a widely used zero-shot unimodal TSF benchmark:
>
> | Benchmark | TimesFM | Moirai | Sundial |
> |---|---:|---:|---:|
> | TimesX | 0.645 | 0.722 | 0.758 |
> | GIFT-Eval | 0.705 | 0.728 | 0.771 |
>
> As shown above, **the ranking is identical** on both benchmarks, and **the results (MASE) are also close**. We view this as evidence that the current numerical component of TimesX does not appear to be heavily skewed for unimodal evaluation.
>
> (3) **We have provided datasets and results from nine additional numerical sources beyond these three APIs**, namely, a TimesX version of the 9 Time-MMD datasets (See results in **Section 3.1.3 and Table 3**; See datasets in original repository in Line 107, Page 2).
>
> Following **the reviewer’s suggested five domains**, we **newly extended TimesX with five official/government data sources**, which are expected to be refreshable with relatively limited manual effort. We also provided constructed contexts, covering 2023–2025 *(see the repository in Line 107, Page 2)*.
>
> | Variable | Domain |
> |---|---|
> | Influenza Patients Proportion | Healthcare |
> | Gasoline Prices | Energy |
> | Travel Volume | Transportation |
> | Drought Level | Climate |
> | Manufacturers' Total Inventories | Manufacturing |
>
> We again thank the reviewer for this constructive suggestion. In the final version, we will add a dedicated discussion paragraph on extending our numerical data sources for future versions.

---

> > ### Author Rebuttal · Reviewer_3Tbv · 2026-04-02
> >
> > Thank you for your detailed response. I will keep my original score.

---

> > > ### Author Response · Authors · 2026-04-07
> > >
> > > We are glad our rebuttal addressed Reviewer 3Tbv’s concerns, and would like to thank Reviewer 3Tbv again for the positive assessment and insightful suggestions.

---

### Decision · Program_Chairs · 2026-04-30

**Decision:**

Accept (regular)

**Comment:**

The paper proposes TimesX, a large-scale, context-enriched multimodal time series forecasting benchmark designed to address limitations of existing benchmarks, including poor generalization, limited textual context, and data leakage.

The paper addresses an important and timely problem, and reviewers consistently recognized the value of establishing a high-quality benchmark for multimodal time series forecasting. The proposed dataset is thoughtfully designed, incorporating diverse real-world domains and multiple types of textual context, and the multi-agent data construction pipeline is a well-motivated and practical solution for scalable and leakage-aware dataset generation. The experimental evaluation is comprehensive and carefully executed, covering a wide range of models and providing insightful findings that challenge existing assumptions.

While some limitations remain, such as the current focus on specific data sources, incomplete coverage of certain domains or settings, and the need for clearer documentation of some design choices, these issues are well acknowledged and do not undermine the overall contribution. The authors provided constructive rebuttal responses that addressed most reviewer concerns, leading to a positive consensus.

Therefore, the paper is accepted.